# From Genomic Whispers to Therapeutics: Multi-Resolution Transcriptome-Guided Diffusion Models for Drug Design and Screening

## Abstract

Traditional drug discovery is protracted and extremely expensive. While *Structure-based Drug Design (SBDD)* has advanced AI-driven molecular generation, target-centric models struggle with diseases arising from the dysregulation of complex physiological systems. To bridge this gap, we introduce *Transcriptome-based Drug Design (TBDD)*: designing molecules from a cell's transcriptomic response to perturbations. We present scTrans-Gen, a diffusion model that conditions generation on multi-resolution transcriptomic data (bulk and single-cell). Central to our approach is a transcriptome-centric condition extractor that aligns perturbation signals across domains into a function-oriented chemical space, avoiding the ill-posed reconstruction of microscopic structures from macroscopic signals. To exploit single-cell data, we propose a Transcriptome Pseudoimage mechanism for robust high-resolution conditioning. Across diverse benchmarks, scTrans-Gen outperforms strong baselines on multiple metrics. We further demonstrate novel inhibitor design for specified gene knockouts and an efficient generate-then-search screening workflow suitable for time-sensitive clinical scenarios. Altogether, scTrans-Gen offers a practical route to function-oriented drug discovery and personalized precision medicine. The code is available at: https://anonymous.4open.science/r/scTrans-Gen.

## 1 Introduction

Drug discovery is a long journey marked by high cost and high failure rates (Sadybekov & Katritch, 2023). For decades, computational methods have sought to accelerate this process, spanning from virtual screening to *de novo* drug design (Sadybekov & Katritch, 2023; Tang et al., 2024). Virtual screening is inherently limited to retrieving molecules from existing libraries and cannot create compounds with novel scaffolds. Generative approaches, especially *Structure-based Drug Design (SBDD)*, have therefore gained traction for their ability to create entirely new molecules by modeling protein 3D structures (Bai et al., 2024) and following the *lock-and-key* principle to yield high-affinity ligands (Saini et al., 2025; Huang et al., 2024). *However, SBDD's target-centric view struggles with systemic diseases driven by dysregulated, multi-pathway networks, and its effectiveness hinges on obtaining high-quality protein structures at scale* (Munson et al., 2024).

We posit that a drug's systemic cellular effects can be captured precisely by its post-perturbation transcriptome (Bunne et al., 2024). Changes in Transcriptome expression reflect the global functional response of cells treated as complex systems (Ji et al., 2021). Yet current work on transcriptomics largely focuses on predicting cellular responses to a given drug rather than exploiting the perturbation signal to reverse-engineer and design new molecules (Hsieh et al., 2023; Rampášek et al., 2019; Wei et al., 2022). We therefore introduce the problem: *Transcriptome-based Drug Design (TBDD)*. Formally, we seek to learn a conditional generator $p(\mathbf{M} \mid \mathbf{z})$, where the functional signal $\mathbf{z}$ is defined by a pair of pre- and post-perturbation transcriptomes $(\mathbf{T}_{\text{pre}}, \mathbf{T}_{\text{post}})$. The generator samples a novel molecule $\mathbf{M}$ that drives the cellular state from $\mathbf{T}_{\text{pre}}$ to $\mathbf{T}_{\text{post}}$. In this way, the desired cellular response directly steers the generative design of new drugs and informs the search for functionally similar molecules (Figure1).

While recent efforts have been made to address this problem (Li & Yamanishi, 2024; Kaitoh & Yamanishi, 2021; Cheng et al., 2024), three core challenges remain: (1) **Ill-posed inverse mapping.** Prior attempts try to reconstruct complete, precise molecular structures directly from macroscopic,

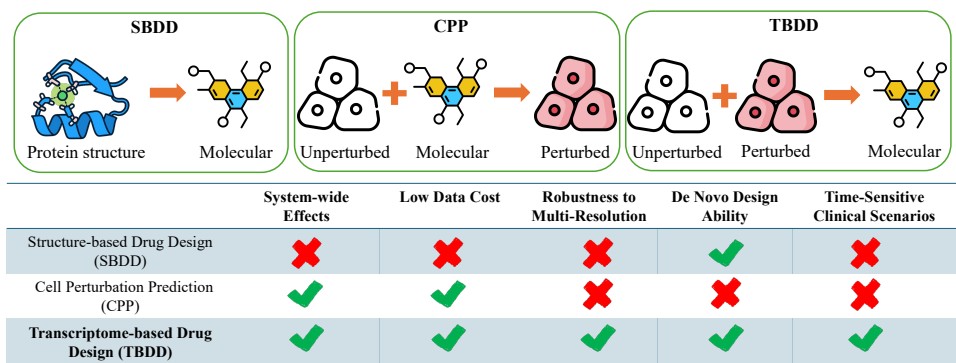

Figure 1: A comparative analysis of TBDD against existing settings.

noisy transcriptomic signals, a classic ill-posed setting (Li & Yamanishi, 2024; Kaitoh & Yamanishi, 2021; Cheng et al., 2024). Transcriptomes primarily encode *functional effects*, not full atomic blueprints. (2) **Cross-modality domain gap.** Transcriptome-expression profiles ("biological language") and molecular graphs ("chemical language") differ in information density, topology, and inductive biases; learning a direct generative map across these heterogeneous spaces is difficult and unstable (Xiao et al., 2024; Zhou et al., 2025). (3) **Multi-resolution opportunities and pitfalls.** Single-cell RNA-seq offers unprecedented resolution for drug action and cellular heterogeneity, but demands robustness to sparsity, technical noise, and batch effects while also supporting bulk data (Hafemeister & Halbritter, 2023; Van de Sande et al., 2023).

Inspired by advances in diffusion models and multimodal alignment (Rombach et al., 2022; Radford et al., 2021), which show that semantic cues can steer generation across domains, we present scTrans-Gen: a diffusion framework for transcriptome-guided *de novo* drug design using multi-resolution transcriptomic data. Instead of direct structural mapping, we introduce a function-centric condition extractor that projects cellular perturbations into a functional chemical space to guide a graph diffusion model. Additionally, we propose a Transcriptome Pseudoimage mechanism to effectively harness single-cell data by reducing noise while preserving biological heterogeneity.

We establish a rigorous evaluation suite for this new setting, covering basic coverage and diversity metrics, structural similarity measures, and estimation of perturbation effects. scTrans-Gen substantially outperforms strong baselines. We also validate practical utility via a zero-shot gene-inhibitor design scenario. Finally, recognizing that *de novo* generation may be too slow for urgent clinical needs, we introduce a generate-then-search screening pipeline built atop our learned functional representations, enabling triage for personalized, time-sensitive care. Our contributions are as follows:

- We propose and formalize, for the first time, *de novo* drug design conditioned on cellular perturbation and build an efficient generate-then-search screening framework, opening a function-oriented direction for drug discovery.
- We design scTrans-Gen, which couples function-centric conditioning with multi-domain alignment to construct a function-oriented intermediate space, effectively resolving cross-modal ill-posedness and mitigating the brittleness of direct inverse mapping.
- We demonstrate the first conditional molecular generation at fine-grained single-cell transcriptomic resolution, enabled by the Transcriptome Pseudoimage technique to combat sparsity and noise, while remaining compatible with coarse-grained bulk resolution.
- By integrating multiple comprehensive evaluation metrics and designing various real application scenarios, we demonstrate the advantages of scTrans-Gen in personalized medicine.

## 2 RELATED WORK

**Machine-Learning–Based Molecular Design.** Deep molecular design has evolved from SMILES sequence models to graph-based approaches that preserve molecular topology (Wang et al., 2025; Gómez-Bombarelli et al., 2018). Hierarchical generators such as Jin et al. (2020); You et al. (2024); Weller & Rohs (2024) efficiently construct large molecules in a coarse-to-fine manner. Yet uncon-

ditional generation is unfocused for drug-design goals. Transformer-based graph diffusion models (Liu et al., 2024; Peng et al., 2023; Hoogeboom et al., 2022; Peng et al., 2023; Schneuing et al., 2024) enable multi-conditional generation via mechanisms like AdaLN to inject external signals. *Structure-based drug design* (SBDD) remains a classical conditional paradigm that uses 3D pocket structures to guide ligand generation and optimize (Alakhdar et al., 2024; Guan et al., 2024), but its single-target perspective limits performance on multi-pathway diseases and relies on high-quality protein structures (Isert et al., 2023; Wang et al., 2018; Fahim, 2025; Ziv et al., 2025).

**Cellular-Perturbation Transcriptomics.** Transcriptomics offers a comprehensive snapshot of cellular function. Large perturbational resources, such as Subramanian et al. (2017); Gao et al. (2019); Zhang et al. (2025), provide massive gene-expression profiles under chemical or genetic perturbations. Building upon them, predictive models (Qi et al., 2024; Hetzel et al., 2022; Lotfollahi et al., 2019; Roohani et al., 2024) integrate chemistry and baseline state to forecast single-cell or bulk responses, while frameworks like Adduri et al. (2025) target heterogeneity and batch effects. Although useful for simulating responses, such models are predictive rather than generative. Emerging *transcriptome-guided generation* methods (Li & Yamanishi, 2024; Kaitoh & Yamanishi, 2021; Cheng et al., 2024) either depend on explicit statistics that risk losing information or focus on bulk data that averages heterogeneity, and they still face the ill-posedness of mapping macroscopic signals to complete structures. These issues underline the need for function-centric conditioning and architectural decomposition, which we pursue in scTrans-Gen.

## 3 PROBLEM FORMULATION

We formalize the problem in terms of three spaces. The *chemical space* ($\mathcal{M}$) comprises molecules, where each molecule $\mathbf{M} \in \mathcal{M}$ is represented as an attributed graph $G = (\mathcal{V}, \mathcal{E})$. The *transcriptome space* ($\mathcal{T}$) is defined as a $d$-dimensional vector space, with each state represented by $\mathbf{T} \in \mathbb{R}^d$. A *cellular perturbation signature* is defined as a pair $(\mathbf{T}_{\text{pre}}, \mathbf{T}_{\text{post}}) \in \mathcal{T} \times \mathcal{T}$, characterizing the transition between transcriptomic states before and after perturbation. The central objective is to learn the conditional distribution $p(\mathbf{M} \mid \mathbf{T}_{\text{pre}}, \mathbf{T}_{\text{post}})$, which quantifies the probability that molecule $\mathbf{M}$ induces the transition from $\mathbf{T}_{\text{pre}}$ to $\mathbf{T}_{\text{post}}$.

**Task 1** *(Transcriptome-based Drug Design)*: A de novo drug design model conditioned on cellular perturbation should be formalized by sampling novel molecules to satisfy a desired biological condition: $\mathbf{M}_{\text{new}} \sim p(\mathbf{M} \mid \mathbf{T}_{\text{pre}}, \mathbf{T}_{\text{post}})$.

**Task 2** *(Transcriptome-based Drug Screening)*: An efficient generate–then–search screening framework needs to be built by systematically evaluating likelihoods to rank existing drug molecules: $\mathcal{M}_k = \underset{\mathcal{M}_k \subset \mathcal{M}_l, |\mathcal{M}_k|=k}{\arg\max} \sum_{\mathbf{M} \in \mathcal{M}_k} p(\mathbf{M} \mid \mathbf{T}_{\text{pre}}, \mathbf{T}_{\text{post}})$, where $\mathcal{M}_l$ is the large-scale molecule library.

Direct end-to-end learning of $p(\mathbf{M} \mid \mathbf{T}_{\text{pre}}, \mathbf{T}_{\text{post}})$ is ill-posed. We introduce an intermediate function-oriented chemical space $\mathcal{Z}$ and factorize: $\mathbf{z} = E_\phi(\mathbf{T}_{\text{pre}}, \mathbf{T}_{\text{post}})$, $\mathbf{M}_{\text{new}} \sim p_\theta(\mathbf{M} \mid \mathbf{z})$, where $E_\phi$ is a perturbation extractor (with a Transcriptome Pseudoimage module for single-cell data), and a conditional graph diffusion generator approximates $p(\mathbf{M} \mid \mathbf{T}_{\text{pre}}, \mathbf{T}_{\text{post}})$ by $p_\theta(\mathbf{M} \mid E_\phi(\mathbf{T}_{\text{pre}}, \mathbf{T}_{\text{post}}))$, focusing learning on the linkage between biological and chemical *function*.

## 4 MULTI-RESOLUTION TRANSCRIPTOME-GUIDED DIFFUSION MODEL

### 4.1 MODEL ARCHITECTURE

Our proposed scTrans-Gen method constructs a graph diffusion model conditioned on cellular perturbation signals from gene expression profiles for controllable molecular generation. The model is primarily composed of two parts: a cellular perturbation signal feature extractor and a conditionally controlled molecular generation diffusion model. The gene perturbation information feature extractor fuses pre- and post-perturbation information and aligns it with the drug feature space. The molecular graph diffusion model controls the generation of drug molecules through conditional injection methods. The scTrans-Gen is the first to incorporate multi-resolution cellular perturbation data. Furthermore, the generated molecules can directly serve various downstream tasks, such as predicting drug mechanisms of action and high-throughput drug screening (Figure 2).

### 4.2 PERTURBATION FEATURE-GUIDED MOLECULAR GRAPH DIFFUSION MODEL

We used a conditional molecular generation diffusion model guided by the learned drug-domain perturbation representations. The core architecture is based on the Diffusion Transformer (Peebles & Xie, 2022), where the conditional features are injected to guide the denoising process.

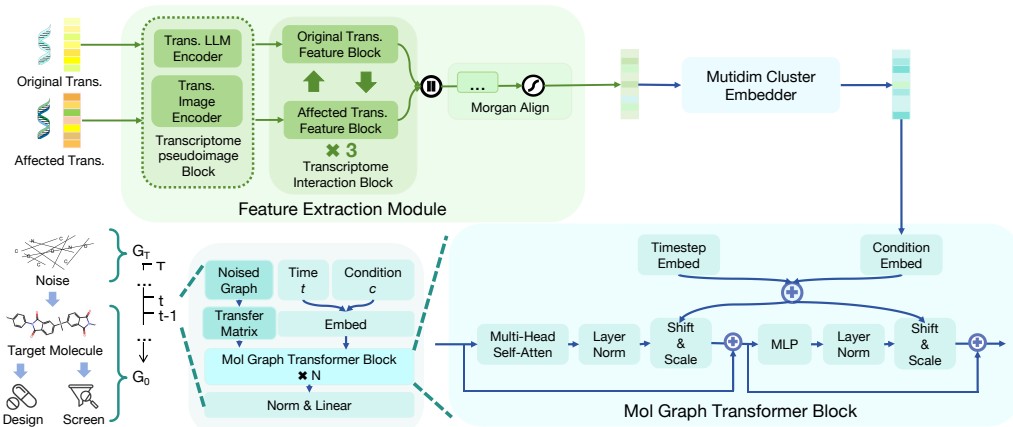

Figure 2: Overall architecture of scTrans-Gen. The model consists of a Feature Extraction Module that processes transcriptome expression data to produce a conditional embedding, and Mol Graph Transformer Blocks that use the embedding to guide the diffusion for generating a target molecule. *Note: Trans. stands for Transcriptome.*

**Molecular Graph Diffusion Model.** The model uses a Markov chain-driven forward process to progressively add noise to the molecular graph's discrete features (atom and bond types): $q\left(X_G^t \mid X_G^{t-1}\right) = \mathrm{Cat}\left(X_G^t; \tilde{p} = X_G^{t-1}\mathbf{Q}_G^t\right)$, where $X$ is the matrix representing the graph $G$ and $\mathbf{Q}$ is the graph transition matrix. A neural network-parameterized reverse process can reconstruct the graph from noise by iteratively removing it. The reverse process learns to predict the original graph $p_\theta\left(\tilde{G}^0 \mid G^t\right) = \prod_{t\in T} p_\theta\left(G^{t-1} \mid G^t\right)$. $p_\theta\left(\tilde{G}^0 \mid G^t\right)$ is combined with $q\left(G^{t-1} \mid G^t, G^0\right)$ to predict the graph reverse distribution $p_\theta\left(G^{t-1} \mid G^t\right) = q\left(G^{t-1} \mid \tilde{G}, G^t\right) p_\theta\left(\tilde{G} \mid G^t\right)$. The training objective is to minimize the negative log-likelihood:

$$\mathcal{L} = \mathbb{E}_{q(G^0)}\mathbb{E}_{q(G^t|G^0)}\left[-\mathbb{E}_{\mathbf{x}\in G^0}\log p_\theta\left(\mathbf{x} \mid G^t\right)\right]. \tag{1}$$

**Gene Perturbation Conditioned Molecular Generation.** The drug-domain structural information from the feature extractor is injected into the Mol Graph Transformer blocks of the DiT via an AdaLN-like method, guided by a multidimensional cluster embedder. We use Classifier-Free Guidance (CFG) (Ho & Salimans, 2022) to implement conditional generation:

$$\hat{p}_\theta\left(G^{t-1} \mid G^t, \mathbf{C}\right) = \log p_\theta\left(G^{t-1} \mid G^t\right) + \mathbf{s}\left(\log p_\theta\left(G^{t-1} \mid G^t, \mathbf{C}\right) - \log p_\theta\left(G^{t-1} \mid G^t\right)\right), \tag{2}$$

where $\mathbf{s}$ represents the scale of guidance and $\mathbf{C}$ represents the condition. During training, we use dynamic feature dropping and noise injection:

$$\mathbf{C} = \begin{cases} E_\theta(\mathbf{z}^t) + \boldsymbol{\epsilon} & \text{with probability } 1 - p \\ \mathbf{e}_{\mathrm{drop}} + \boldsymbol{\epsilon} & \text{with probability } p \end{cases}, \quad \boldsymbol{\epsilon} \sim \mathcal{N}(0, \mathbf{I}). \tag{3}$$

With probability $p$, a sample's embedding is replaced by a learnable dropout vector $\mathbf{e}_{drop}$; otherwise, it is processed by embedder $E_\theta$. Isotropic noise $\boldsymbol{\epsilon}$ is then added.

### 4.3 CELLULAR PERTURBATION SIGNAL FEATURE EXTRACTION

To effectively extract drug-domain control conditions from the gene transcriptome space, we designed a multi-domain alignment architecture to train the perturbation information feature extractor. This multi-domain alignment-guided architecture includes three parts: the perturbation signal feature extractor, a drug molecule graph VAE representation module, and a drug molecule fingerprint representation module. The feature extractor is the core module, aiming to extract cellular perturbation signals and map them to the drug molecular domain (Figure 3).

As illustrated in Figure 3, our cellular perturbation signal feature extraction module includes a Transcriptome Pseudoimage Block, a Transcriptome Interaction Block, and a multi-domain feature alignment module. Compared to existing methods, scTrans-Gen can perform feature fusion and extraction on both bulk and single-cell data. For single-cell transcriptome data, we specifically designed

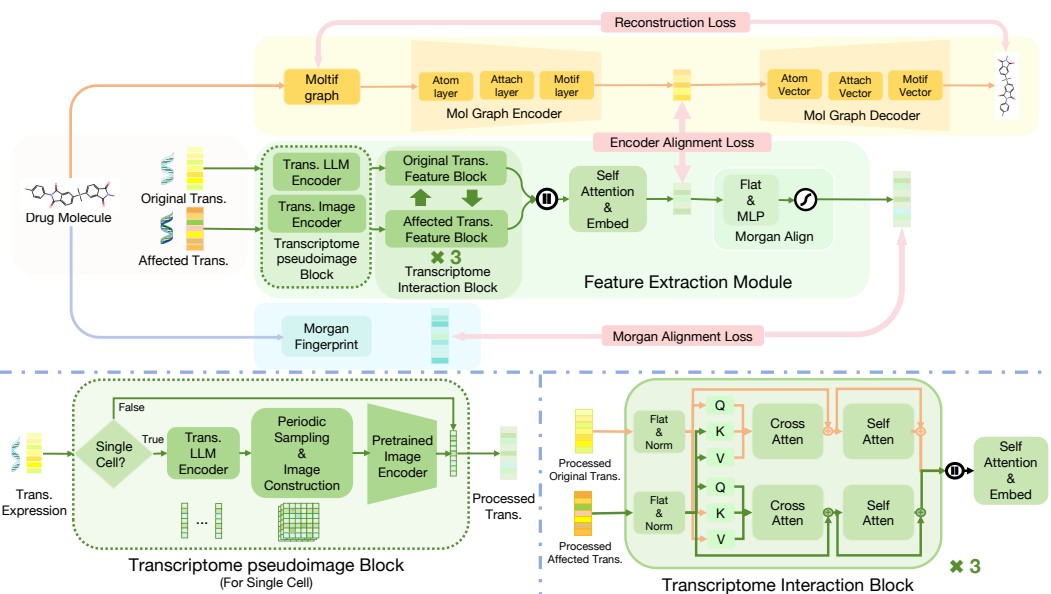

Figure 3: Detailed architecture of the Feature Extraction Module and its multi-domain alignment losses. It showcases the Transcriptome Pseudoimage Block for single-cell data and the Transcriptome Interaction Block, which feed into an alignment process with both a Mol Graph VAE and Morgan Fingerprints. *Note: Trans. stands for Transcriptome.*

the Transcriptome Pseudoimage Block, which transforms single-cell data into a pseudoimage structure for processing, enhancing the utilization of single-cell data. The paired gene data undergoes information exchange in the Transcriptome Interaction Block to extract the gene perturbation signal. The functional perturbation features are then sequentially aligned with molecular graph encoded features and molecular fingerprints to be used in the subsequent generation control task.

**Transcriptome Pseudoimage Block.** Current molecular generation tasks lack methods for extracting and integrating single-cell transcriptome data. Compared to bulk-level, single-cell data is highly sparse with significant technical noise, leading the model to capture noise rather than true biological signals. To handle the unique sparsity and noise of this data modality while preserving its high resolution, we designed the Transcriptome Pseudoimage Module. We use a pre-trained Transcriptome LLM Encoder SCimilarity (Heimberg et al., 2025) to obtain a dense embedding for each cell (from $D > 60,000$ to $d = 128$). Then, we construct a pseudoimage by sampling $N$ cells from cycle-specific clusters according to their proportions and averaging within clusters. This pseudoimage is encoded using a pretrained VAE to derive a dense feature representation. This aggregation preserves cell-type-specific signals while smoothing out noise, allowing for robust feature extraction.

**Transcriptome Interaction Module.** To facilitate interaction between the original and post-perturbation gene transcription information, we designed the Transcriptome Interaction Module. This module supports interaction for both bulk cell data and single-cell data preprocessed by the Transcriptome Pseudoimage Block. The module contains three Transcriptome Interaction blocks, each using attention mechanisms for feature interaction. Each block has separate yet interactive units for pre- and post-perturbation data. Each unit contains a cross-attention block and a self-attention block, with residual connections outside the attention modules. During inference, the paired gene transcription data are input into their corresponding units and fused in the Cross Attention module, where the data from the current unit serves as the Query, and the data from the other unit serves as the Key and Value. The subsequent self-attention module reinforces the perturbation feature information. At the module's output, a fusion unit combines the features from both units to form the functional perturbation representation.

## 4.4 MOLECULAR MULTI-DOMAIN INFORMATION ALIGNMENT ARCHITECTURE

In molecular representation, fingerprints capture structural details and enable precise control, while graph encoders represent molecules as graphs. However, molecular fingerprints are high-

dimensional and sparse, making them unsuitable for single-stage alignment. Graph encoders, on the other hand, offer limited control in diffusion-based generation tasks, hindering precise manipulation. We designed a multi-domain spatial alignment method with a two-stage training process and customized loss functions to handle the high dimensionality and sparsity of the feature spaces.

**Molecular Graph Encoder-Decoder Architecture.** We use a hierarchical graph generation architecture based on a Variational Autoencoder (VAE) to provide an alignment paradigm for conditional molecule generation. The model represents molecules through a hierarchical graph with three interrelated levels: the motif, attachment and atom layers. The hierarchical encoder proceeds in a fine-to-coarse direction for information aggregation: $\{\mathbf{h}_{\mathcal{V}}\} \xrightarrow{\mathcal{R}} \{\mathbf{h}_{\mathcal{A}_i}\} \xrightarrow{\mathcal{G}} \{\mathbf{h}_{\mathcal{S}_i}\} \xrightarrow{\mathcal{H}} \mathbf{z}_{\mathcal{G}}$, where $\mathcal{R}, \mathcal{G}, \mathcal{H}$ are non-linear transformations and $\{\mathbf{h}_v\}, \{\mathbf{h}_{\mathcal{A}_i}\}, \{\mathbf{h}_{\mathcal{S}_i}\}$ are atom, attachment, and motif layers respectively. The hierarchical decoder adopts a coarse-to-fine generation paradigm.

**Molecular Fingerprints.** As a core representation tool in chemoinformatics, molecular fingerprints map the complex topological structure and chemical information of a molecule to a fixed-dimensional numerical vector space. In this study, we chose a vectorization scheme based on Morgan fingerprints to systematically capture local structural environments.

**Stage 1: Molecular Graph Space Alignment and Reconstruction Constraint.** This stage aligns the transcriptome functional features with the latent space of a pre-trained molecular graph VAE. The overall loss function for this stage is:

$$\mathcal{L}_{\text{vae}} = \underbrace{-\mathbb{E}_{\mathbf{z}\sim\mathbf{Q}}[\log P(\mathbf{M}|\mathbf{z}_{\text{enc}})] + \lambda_{\text{KL}} D_{\text{KL}}[Q(\mathbf{z}_{\text{enc}}|\mathbf{M})||P(\mathbf{z}_{\text{enc}})]}_{\mathcal{L}_{\text{vae-ELBO}}} + \underbrace{\|E(\mathbf{z}_{\text{enc}}) - E(\mathbf{z}_f)\|^2 + \|V(\mathbf{z}_{\text{enc}}) - V(\mathbf{z}_f)\|^2}_{\mathcal{L}_{\text{vae-align}}} \quad (4)$$

where $\mathcal{L}_{\text{vae-ELBO}}$ is the standard VAE evidence lower bound loss, and $\mathcal{L}_{\text{vae-align}}$ aligns the mean and variance of the transcriptome-derived features with the VAE's latent space.

**Stage 2: Molecular Fingerprint Space Alignment.** In this stage, the aligned features are mapped to the Morgan Fingerprint space. We use a joint loss mechanism that fuses sparse-aware regression with label-guided contrastive learning:

$$\mathcal{L}_{\text{morgan}} = \underbrace{\frac{1}{N_{\text{pos}}} \sum_{(i,j)\in\mathcal{P}} \mathbf{w}_{ij}(\mathbf{A}_{ij} - \mathbf{B}_{ij})^2 + \alpha \cdot \frac{1}{N_{\text{neg}}} \sum_{(i,j)\in\mathcal{Z}} \mathbf{A}_{ij}^2}_{\mathcal{L}_{\text{reg}}} + \underbrace{\mathcal{L}_{\text{InfoNCE}} + \lambda \frac{1}{b\cdot d} \sum_{i=1}^{b} \sum_{j=1}^{d} (\mathbf{A}_{ij} \cdot \mathbb{1}(\mathbf{B}_{ij} = 0))^2}_{\mathcal{L}_{\text{contrast}}}. \quad (5)$$

$\mathcal{L}_{\text{reg}}$ is the sparse-aware regression loss, where $\mathbf{A}$ and $\mathbf{B}$ are predicted vector and target fingerprint; $\mathcal{P}$ and $\mathcal{Z}$ are sets of non-zero and zero positions in $\mathbf{B}$; $N_{\text{pos}} = |\mathcal{P}|$ and $N_{\text{neg}} = |\mathcal{Z}|$; $\mathbf{w}_{ij} = \log(1+\mathbf{B}_{ij})$ is a logarithmic weight. $\alpha = 0.4$ is used in training. $\mathcal{L}_{\text{InfoNCE}}$ is the standard contrastive loss. Another part of $\mathcal{L}_{\text{contrast}}$ is a regularization term to penalize non-zero predictions for zero-valued positions in the target fingerprint, where $\mathbb{1}(\cdot)$ denotes the indicator function, $b$ is the batch size, and $d$ is the dimension. See the appendix for more details. More pseudocode details are in the Appendix A.4.

## 4.5 DRUG SCREENER

Recognizing that the resource-intensive nature of de novo synthesis and safety testing restricts rapid clinical deployment, we propose a generate-then-screen workflow. By exploiting the efficiency of computational generation to circumvent synthesis bottlenecks, this approach identifies candidates from existing drug libraries, enabling immediate utility in urgent therapeutic settings. It uses the generated molecules as query molecules to screen large compound libraries for structurally similar analogs with established clinical data. The core of the screener is a pre-built molecular fingerprint library $\mathcal{F}$. We perform a Top-K nearest neighbor search:

$$\mathcal{F}_k = \underset{\mathcal{F}_k\subset\mathcal{F}, |\mathcal{F}_k|=k}{\arg\max} \sum_{\mathbf{f}_l\in\mathcal{F}_k} \mathbf{T}(\mathbf{f}_q, \mathbf{f}_l) = \underset{\mathcal{F}_k\subset\mathcal{F}, |\mathcal{F}_k|=k}{\arg\max} \sum_{\mathbf{f}_l\in\mathcal{F}_k} \frac{\mathbf{f}_q \cdot \mathbf{f}_l}{\|\mathbf{f}_q\|^2 + \|\mathbf{f}_l\|^2 - \mathbf{f}_q \cdot \mathbf{f}_l} \quad (6)$$

which uses the Tanimoto similarity coefficient to quantify similarity between the query fingerprint $\mathbf{f}_q$ and the fingerprint $\mathbf{f}_l$ from $\mathcal{F}$. It holds broad prospects for designing novel drugs, drug repositioning, and advancing personalized precision medicine.

## 5 EXPERIMENTS

In the experimental section, we follow the same perspective as our evaluation metrics, assessing the model's performance from three angles: macroscopic evaluation of the relationship between the

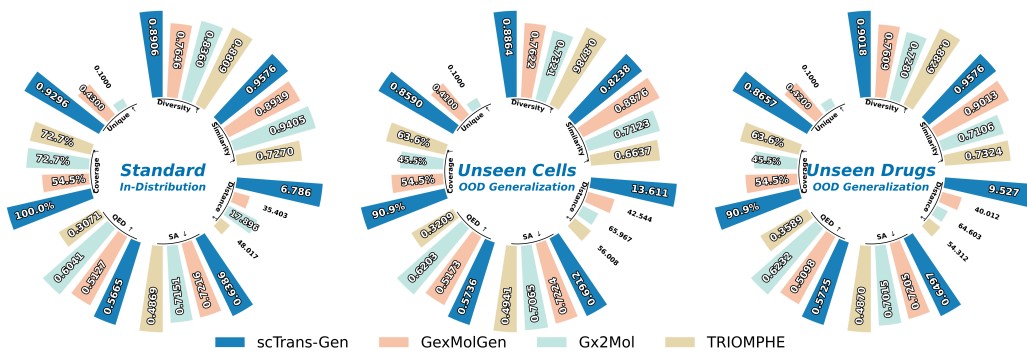

Figure 4: **Generalization performance of scTrans-Gen.** We assess the model's performance across three data splits representing different generalization challenges. **(a) In-Distribution:** performance on a random hold-out of cell-drug pairs. **(b) OOD (Unseen Cells):** generalization to held-out cell lines. **(c) OOD (Unseen Drugs):** generalization to held-out drugs. *Note: The 'Unique' metric is not applicable to the TRIOMPHE method due to its specific methodology.*

generated and target molecular sets and the chemical and medicinal properties of the generated set itself, and microscopic evaluation of the effectiveness and accuracy of scTrans-Gen conditional control generation. To demonstrate the model's generalization ability and the functional effects of the generated drugs, we designed the following three innovative evaluation experiments: zero-shot prediction of gene inhibitors, characterization of the functional effects of generated drugs, and accuracy assessment of the drug screener. To ensure reproducibility, we provide the necessary hyperparameter settings in Appendix C.1.

### 5.1 EXPERIMENTAL SETUP

**Datasets.** *Bulk Cell Data*: We used the L1000 Level 3 dataset (Subramanian et al., 2017; Gao et al., 2019), which profiles the expression of 978 landmark genes across nearly 20,000 drugs and various cell lines. For training, we split the data 85:10:5 (train:test:val) using three strategies: random, mask drug, and mask cell. *Single-Cell Data*: We also utilized the Tahoe-100M dataset (Zhang et al., 2025), the largest single-cell perturbation dataset available. It contains results from over 300 drugs applied to 50 cancer cell lines, including their untreated states. *Gene Inhibitor Dataset*: For evaluation, we built a gene inhibitor dataset from the ExCape database (Sun et al., 2017). This set contains 1,200 to 23,000 known inhibitors for each of 10 selected human genes, enabling comparison with gene knockout expression profiles. To guarantee experimental fairness, we include a detailed description of the training data in Appendix C.2.

**Evaluation Metrics.** We used three types of metrics to assess the model's generative capabilities: *Macroscopic Metrics*: To reflect the properties of the entire generated set of drug molecules: (1) Heavy Atom Type Coverage (Coverage); (2) Internal Diversity among generated samples (Diversity); (3) Fragment-based similarity to a reference set (Similarity); (4) Fréchet ChemNet Distance to a reference set (Distance); (5) Synthesizability of the target molecule (SA); (6) Uniqueness of structures in a single generated batch (Unique); (7) Quantitative Estimate of Drug-likeness (QED); (8) Validity of generated molecules (Validity). *Microscopic Metrics*: To assess the reliability of drug prediction based on gene perturbation: (1) Fraggle-based molecular scaffold similarity (Fraggle Sim.); (2) Morgan fingerprint-based atomic environment similarity (Morgan Sim.); (3) MACCS key-based binary fingerprint similarity (MACCS Sim.) (Grant & Sit, 2021; Wang et al., 2022). *Experimental Design Metrics*: Innovatively designed to reflect the functional effects of generated drugs: (1) A metric to evaluate the difference in cellular gene expression effects between the generated drug and the ground-truth drug (PRnet MSE). (2) On zero-shot data of gene inhibitor effects, a metric to evaluate the similarity between the generated molecules and known gene inhibitors (Gene Inhibitor Sim.) (Méndez-Lucio et al., 2020).

**Baselines.** For the bulk data experiments, we selected several strong and widely recognized baseline models from recent, similar tasks for comparison: GexMolGen (Cheng et al., 2024), which links gene expression differences to molecular structure design; TRIOMPHE (Kaitoh & Yamanishi, 2021), which combines target protein transcriptome perturbation data with Bayesian optimization;

Table 1: Microscopic evaluation of generation similarity. We measure structural similarity (Fraggle, Morgan, MACCS) and functional similarity (PRnet MSE). For PRnet MSE, lower is better. *Note: Exhibiting suboptimal performance on single-cell data, PRNet and similar perturbation prediction methods are not amenable to evaluation in a single-cell modality.*

| Data Type | Split | Method | Fraggle Sim. ↑ | Morgan Sim. ↑ | MACCS Sim. ↑ | PRnet MSE ↓ |
|---|---|---|---|---|---|---|
| Bulk | In-Distribution | GexMolGen | 0.3278 | 0.1098 | 0.3771 | 4.6504 |
| | | Gx2Mol | 0.3818 | 0.1556 | 0.4359 | 2.5987 |
| | | TRIOMPHE | 0.2352 | 0.0790 | 0.3301 | 7.4599 |
| | | **scTrans-Gen** | **0.8892** | **0.8228** | **0.9031** | **0.2328** |
| | Out-of-Distribution (Unseen Cells) | GexMolGen | 0.3635 | 0.1195 | 0.4033 | 4.2724 |
| | | Gx2Mol | 0.3277 | 0.1060 | 0.3814 | 3.7071 |
| | | TRIOMPHE | 0.2200 | 0.0730 | 0.2956 | 8.6310 |
| | | **scTrans-Gen** | **0.9449** | **0.9125** | **0.9411** | **0.2932** |
| | Out-of-Distribution (Unseen Drugs) | GexMolGen | 0.2921 | 0.1001 | 0.3508 | 5.0482 |
| | | Gx2Mol | 0.3738 | 0.1381 | 0.4579 | 2.7208 |
| | | TRIOMPHE | 0.2362 | 0.0802 | 0.3382 | 7.4666 |
| | | **scTrans-Gen** | **0.8592** | **0.7722** | **0.8622** | **0.4866** |
| Single-cell | In-Distribution | **scTrans-Gen** | **0.7310** | **0.6114** | **0.7590** | - |

and Gx2Mol (Li et al., 2024), which uses a VAE-LSTM fusion architecture. For single-cell data, there is currently a lack of effective molecular generation methods in the same task domain, so we focus on discussing the performance of our proposed method.

**Evaluation Setting.** To rigorously evaluate generalization capabilities beyond training data reconstruction, we established two complementary out-of-distribution (OOD) protocols: Unseen Drugs and Unseen Cell Lines. The former enforces zero molecular overlap between training and testing, compelling the model to infer chemical structures solely from functional perturbation signatures; this validates that the model learns generalized structure-function mappings, while preserving critical pharmacophores for functional equivalence. Simultaneously, the Unseen Cell Lines utilizes disjoint cell lines to challenge the model with novel transcriptomic backgrounds, demonstrating its capacity to disentangle intrinsic drug mechanisms from cellular heterogeneity and generalize pharmacological insights to previously unseen biological contexts.

## 5.2 Preliminary Evaluation of Drug Molecular Generation

This experiment compares scTrans-Gen and baseline methods from a macro perspective (Figure 4). Our method outperforms existing benchmarks across multiple key metrics. It achieves comprehensive heavy atom coverage, high diversity, strong structural similarity, and the lowest Fréchet distance, indicating superior consistency. The approach also generates significantly more unique molecules with minimal duplication. It demonstrates robust adaptability across various training strategie, random, cell-masked, and drug-masked (Yang et al., 2024). While not the highest in every SA or QED scenario, our method maintains a balanced and strong performance, avoiding the extreme limitations of baselines such as TRIOMPHE's poor drug-likeness (SA: 0.4869, QED: 0.3071) or Gx2Mol's low uniqueness (0.1000). Moreover, our model is the first successfully applied to single-cell resolution data, achieving high performance despite inherent noise and sparsity. We also extended our evaluation to include toxicity properties in Appendix B.6, confirming the model's advantageous performance in controlling toxicity, which is essential for pharmaceutical viability.

## 5.3 Evaluation of Transcriptome-guided Drug Molecular Generation

This experiment compares scTrans-Gen and baseline methods from the perspective of conditional control effectiveness. To comprehensively evaluate the effectiveness of scTrans-Gen conditional generation, we conducted quantitative experiments from two dimensions: structural accuracy and functional similarity. The experiment first quantitatively assesses the model's ability to generate the target drug structure. As shown in Table 1, our method has a significant advantage, with Fraggle, Morgan, and MACCS similarities reaching near-perfect values under all of the splits. In contrast, the baseline models performed poorly. Structural similarity does not guarantee functional equivalence. To assess this, we developed a method using PRnet to predict drug-induced expression states and measure their similarity via MSE. A lower MSE indicates that the generated drug's effect is closer to the target drug. Our method's MSE was far lower than all baselines, proving the high functional fidelity of the generated molecules. We attribute this substantial leap in performance to our method's

ability to adeptly address the challenges outlined in the Introduction, whereas competing methods largely fail to resolve the issues of *ill-posed inverse mapping* and *cross-modality domain gap*.

## 5.4 GENE INHIBITOR PREDICTION

To assess scTrans-Gen's utility in drug development and its capacity to capture functional biological mechanisms, we established a rigorous zero-shot benchmark targeting 10 canonical genes (e.g., AKT1, EGFR, TP53) backed by extensive inhibitor libraries to ensure statistical stability. to ensure a fair and unbiased comparison, we enforced a strict zero-shot protocol: all models were trained exclusively on standard drug-perturbation transcriptomes and were never exposed to gene knockout (KO) data. In this protocol, models trained exclusively on drug-perturbation data were tasked with generating molecules conditioned on unseen gene knockout (KO) transcriptomic signatures, positing that a functionally aware model should generate structures similar to known inhibitors that mimic these phenotypic effects. Performance was quantified by the arithmetic mean of maximum structural similarity scores (using Fraggle, Morgan, and MACCS fingerprints) between generated candidates and ground-truth inhibitor sets.

Table 2: Zero-shot Gene inhibitor similarity.

| Target Gene | Gex-MolGen | TRIOM-PHE | Gx2Mol | scTrans-Gen |
|---|---|---|---|---|
| AKT1 | 0.7284 | 0.5401 | 0.7431 | **0.8037** |
| AKT2 | 0.7119 | 0.5151 | 0.7063 | **0.7545** |
| AURKB | 0.7440 | 0.5529 | 0.7194 | **0.7604** |
| CTSK | 0.7487 | 0.5352 | 0.6986 | **0.7512** |
| EGFR | 0.7467 | 0.5405 | 0.7378 | **0.7822** |
| HDAC1 | 0.7196 | 0.5188 | 0.6971 | **0.7717** |
| MTOR | 0.7940 | 0.5274 | 0.7448 | **0.8076** |
| PIK3CA | 0.7638 | 0.5243 | 0.7257 | **0.8088** |
| SMAD3 | 0.8448 | 0.5902 | 0.8428 | **0.8811** |
| TP53 | 0.8093 | 0.5877 | 0.7932 | **0.8160** |

As detailed in Table 2, scTrans-Gen significantly outperformed all baseline methods, consistently achieving the highest similarity scores across all 10 targets. This superior zero-shot performance validates that the model effectively bridges the modality gap, organizing transcriptomic perturbations according to underlying Mechanisms of Action (MoA). By successfully translating gene function signals into specific inhibitor structures, scTrans-Gen demonstrates a robust ability to extract and transfer biologically meaningful functional information for de novo drug design.

## 5.5 DRUG SCREENER

To assess translational utility, we employed a generate-then-search workflow, utilizing a generated molecule as structural probes to query large-scale drug databases. Table 3 details screening performance across retrieval thresholds $k$ based on three metrics: average Structural Similarity between the generated molecules and the top-$k$ retrieved candidates (MACCS), the probability of the ground-truth drug appearing within the retrieved candidates (Hit Rate), and the functional divergence in predicted gene expression effects

Table 3: Performance of the drug screener.

| Top-K | MACCS Sim.↑ | Hit Rate↑ | PRnet MSE↓ |
|---|---|---|---|
| 5 | 0.9714 | 0.6467 | 0.1668 |
| 10 | 0.9554 | 0.8563 | 0.1353 |
| 15 | 0.9421 | 0.8922 | 0.1228 |
| 20 | 0.9269 | 0.9116 | 0.1093 |

between the retrieved candidates and the ground truth (PRnet MSE). Expanding $k$ reveals a characteristic trade-off: while structural similarity naturally attenuates due to the inclusion of distant neighbors, retrieval efficacy improves significantly, evidenced by higher Hit Rates and enhanced functional alignment (lower PRnet MSE). Notably, at a threshold of $k = 15$, the model achieves a Hit Rate approaching 90%, demonstrating a robust capability to identify target drugs based on functional transcriptomic inputs. This demonstrates scTrans-Gen's capacity to distill vast chemical libraries into a clinically manageable panel (e.g., 10–20 compounds) with high structural and functional fidelity, offering a pragmatic solution for time-critical therapeutic applications.

## 5.6 BIOCHEMICAL INTERPRETABILITY ANALYSIS

To systematically evaluate the interpretability of scTrans-Gen, we analyzed the model's learned representations from two complementary dimensions: the biological relevance of the functional latent space and the chemical structural fidelity of the generated molecules.

**Biological Interpretability Analysis.** Since scTrans-Gen relies on phenotypic changes (TBDD) without explicit affinity metrics, we employed stratified UMAP to verify mechanistic principles. First, projecting distinct inhibitors within fixed cellular backgrounds (Figure 7, top) revealed discrete clustering by inhibitor type. This topological separation implies the model encodes mechanism-

Table 4: Ablation study of the feature extractor. *Note: w/o Extractor is trained with L1000 level 5.*

| Dataset | Method | Validity↑ | Coverage↑ | Diversity↑ | Distance↓ | SA↓ | QED↑ | Morgan_Sim↑ |
|---|---|---|---|---|---|---|---|---|
| L1000 | w/o Extractor | 0.8775 | 90.91% | 0.7504 | 8.4982 | 0.8355 | 0.5426 | 0.1824 |
| | w/o Alignment | 0.3000 | 63.64% | 0.7662 | 82.7183 | 0.7651 | 0.4556 | 0.0886 |
| | w/o Interaction | 0.2400 | 36.36% | 0.6982 | 51.0830 | 0.6933 | 0.4400 | 0.2527 |
| | **scTrans-Gen** | **0.9350** | **100.00%** | **0.8906** | **6.7856** | **0.6386** | **0.5665** | **0.8228** |
| Tahoe | w/o VAE | 0.9650 | 81.82% | 0.8693 | 43.0227 | 0.6043 | 0.4588 | 0.2674 |
| | w/o Fingerprint | 0.9400 | 81.82% | 0.8600 | 29.6223 | 0.6357 | 0.3048 | 0.3219 |
| | **scTrans-Gen** | **0.9800** | **90.91%** | **0.8771** | **27.6223** | **0.5994** | **0.4946** | **0.6114** |

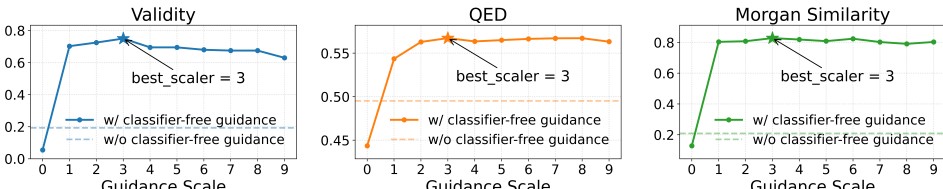

Figure 5: Effect of classifier-free guidance (CFG) scaler on three metrics.

specific signatures, mapping perturbations to Mechanisms of Action (MoA) rather than fitting noise. Second, visualizing identical inhibitors across diverse cell lines (Figure 7, bottom) exhibited stratification by cellular identity, confirming the model dynamically adapts functional representations to biological contexts rather than overfitting. Collectively, these results demonstrate that the latent space effectively disentangles functional drug impacts from cellular backgrounds, supporting function-oriented drug discovery.

**Chemical Structural Interpretability Analysis.** For the chemical structural analysis, we examined the generative diversity and structural logic of the output molecules through stochastic multi-sampling. We visualized multiple molecules generated from the same transcriptomic condition (as shown in Figure 6 and Table 9). The results indicate that while the generated molecules maintain high similarity scores (Fraggle/Morgan/MACCS) to the reference drugs, they exhibit significant diversity in their SMILES representations. Generated molecules are not identical to training targets but share critical functional groups (pharmacophores) and local chemical environments. This confirms the model has learned the underlying mechanism of how specific chemical substructures drive transcriptomic changes.

## 5.7 ABLATION STUDIES

**Feature Extractor**: We conducted ablation studies to validate the contribution of key modules in our feature extractor on both bulk and single-cell data (Table 7). The results confirm that each component is crucial. On bulk data, removing the domain alignment module led to a catastrophic performance drop. On single-cell data, the dual-domain alignment mechanism proved essential for balancing structural similarity and drug-likeness.

**Impact of CFG Guidance Strength**: We explored the effect of different classifier-free guidance strengths, revealing a trade-off between molecule quality and conditional adherence. As shown in Figure 5, performance peaks around a guidance strength of 3, establishing it as the optimal point (Karras et al., 2024).

**Transcriptome Pseudoimage Block:** An ablation experiment for Pseudoimage Block is shown in Appendix A.1 (Table 5), demonstrating the effectiveness of this module.

## 6 CONCLUSION

We introduce a function-driven strategy for textitde novo drug design using cellular perturbation responses. Our model, scTrans-Gen, employs function-centric conditioning and graph diffusion to resolve cross-modal ambiguity. It introduces a "pseudoimage" representation for conditional molecular generation at single-cell resolution, capturing cellular heterogeneity. Evaluations and a screening workflow confirm its strong performance and practical value. This approach provides a general, function-aware foundation for targeted drug discovery and personalized medicine.

## 7 ETHICS STATEMENT

This research was conducted in accordance with all relevant ethical guidelines and regulations. We are committed to responsible research practices and affirm that our work complies with the ethical standards.

## 8 REPRODUCIBILITY STATEMENT

To ensure reproducibility of our work, we have made our anonymously source code available at `https://anonymous.4open.science/r/scTrans-Gen`. Our experiments utilized exclusively open-access data, including the L1000 dataset (bulk RNA-seq) (Subramanian et al., 2017; Gao et al., 2019), Tahoe-100M (single-cell data) (Zhang et al., 2025), and ExCape (gene inhibitor information) (Sun et al., 2017). All hyperparameters used for training are explicitly documented in the configuration files within the code repository. For detailed implementation and reproduction steps, please refer to the provided code and README documentation.

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

## A METHODOLOGY DETAILS

### A.1 TRANSCRIPTOME PSEUDOIMAGE BLOCK

Current molecular generation tasks lack methods for extracting and integrating single-cell transcriptome data. Compared to bulk cell data, single-cell data is highly sparse with significant technical noise, which can lead the model to capture noise rather than true biological signals during learning. To handle the unique sparsity and noise of this data modality while preserving its high resolution, we designed the Transcriptome Pseudoimage Module. We use a pre-trained Transcriptome LLM Encoder to obtain a dense embedding for each cell. Then, for a given perturbation, we group cell embeddings according to cell cycle proportions and randomly sample data to form a pseudoimage. This aggregation preserves cell-type-specific signals while smoothing out noise, allowing for robust feature extraction. Specifically:

1. Transcriptome LLM Encoder: We use a Transcriptome LLM Encoder to encode individual cell transcriptome data, transforming each high-dimensional, sparse single-cell expression profile ($D > 60,000$) into a low-dimensional, dense cell embedding vector ($d = 128$). We use the pre-trained representation model SCimilarity, a deep metric-learning framework designed to learn a unified and interpretable representation for scRNA-seq data, enabling efficient searching for transcriptionally similar cells in large-scale cell atlases. Specifically, the model was trained on a large-scale human scRNA-seq/snRNA-seq dataset spanning multiple tissues and diseases, containing approximately 7.9 million single-cell profiles from 56 studies, and was used to build a searchable reference atlas of 23.4 million cells from 412 studies. Its core objective is to create a foundational model of cell states, generating an effective single-cell representation that can be used across applications without retraining. By learning a low-dimensional embedding space ($d = 128$), it places transcriptionally similar cell profiles close to each other while keeping dissimilar ones apart. This model helps to centralize and extract drug functional information from single-cell perturbation data.

2. Transcriptome Pseudoimage Encoder: After obtaining the $d$-dimensional embedding vector for each cell, we randomly sample $k$ cells ($k = 15$) from the feature vector subsets of different cell cycles (G1, S, G2/M) within the same cell line. We compute the mean to obtain more stable cell cycle cluster subsets. Then, according to the proportions of different cycles in the cell line, we sample a fixed total of $N$ elements from these cluster subsets to form an $N \times d$ cell line representation, which we term a pseudoimage. We use a pre-trained encoder to extract features from this pseudoimage, reducing the large, sparse single-cell data into a more information-dense feature representation.

In summary, the Transcriptome Pseudoimage Module constructs an image-level representation for each condition (i.e., a specific cell line + a specific drug perturbation) that represents the state of the cell population under that condition. This approach ensures the model can handle the inherent sparsity and noise of single-cell data while effectively leveraging the information provided by its high resolution.

**Transcriptome Pseudoimage Block Ablation Study.** To validate the necessity of our architectural design, we conducted comprehensive ablation studies on the Transcriptome Pseudoimage mechanism and encoder strategy. Results demonstrate that constructing 2D pseudoimages via cell-cycle–stratified sampling is crucial for preserving local biological heterogeneity while mitigating single-cell sparsity, yielding significantly higher structural fidelity than random sampling. Furthermore, comparative analysis against MLP and scratch-trained CNN baselines (Table 5) confirms that employing a pretrained and fine-tuned vision encoder provides the essential inductive bias to extract information-dense features from these structured inputs; this strategy not only secures superior metric performance but also ensures training stability and rapid convergence, effectively bridging the modality gap between transcriptomic profiles and molecular structures.

### A.2 MOLECULAR GRAPH ENCODER-DECODER ARCHITECTURE

We use a hierarchical graph generation architecture based on a Variational Autoencoder (VAE) to provide an alignment paradigm for conditional molecule generation. The model utilizes structural motifs as basic building blocks and represents molecules through a hierarchical graph with three interrelated levels: the motif layer, the attachment layer, and the atom layer. This design allows

Table 5: Ablation study of Transcriptome Pseudoimage Block. We compare different encoder architectures and pretraining strategies.

| Method | Coverage ↑ | Diversity ↑ | Similarity ↑ | Distance ↓ | SA ↓ | Unique ↑ | QED ↑ | Fraggle Sim ↑ | Morgan Sim ↑ | MACCS Sim ↑ |
|---|---|---|---|---|---|---|---|---|---|---|
| w/o pseudoimage | 7 (63.64%) | 0.75 | 0.74 | 26.89 | 0.61 | 0.34 | 0.59 | 0.23 | 0.07 | 0.23 |
| MLP | 7 (63.64%) | 0.79 | 0.78 | 13.26 | 0.68 | 0.45 | 0.55 | 0.45 | 0.29 | 0.46 |
| Conv | 9 (81.82%) | 0.84 | 0.90 | 8.15 | 0.64 | 0.88 | 0.57 | 0.74 | 0.69 | 0.75 |
| w/o pretrain | 10 (90.91%) | 0.87 | 0.94 | 6.89 | 0.64 | 0.90 | 0.56 | 0.86 | 0.80 | 0.89 |
| **Ours (Pretrained + Finetuned)** | **11 (100.00%)** | **0.89** | **0.96** | **6.79** | **0.64** | **0.93** | **0.57** | **0.89** | **0.82** | **0.90** |

the model to integrate information at multiple resolutions, enabling efficient processing of large molecules.

The hierarchical encoder uses a deep architecture based on multi-resolution graph representations, designed to capture the hierarchical structure of molecular graphs. It constructs a three-level topological representation: the atom layer stores the atom and bond information of the original molecular graph; the attachment layer abstracts the connection points between motifs, with each node representing a set of junction atoms between a motif and its neighbors; the motif layer describes the higher-order topological connections of structural motifs, forming a tree-like coarse-grained structure. The encoding process proceeds in a fine-to-coarse direction for information aggregation. Let $\mathcal{F}, \mathcal{G}, \mathcal{H}$ be the non-linear transformation functions for the atom, attachment, and motif layers, respectively. Let $\{\mathbf{h}_{\mathcal{V}}\}, \{\mathbf{h}_{\mathcal{A}_i}\}, \{\mathbf{h}_{\mathcal{S}_i}\}$ represent the corresponding layer features, and $z_{\mathcal{G}}$ be the final graph representation: $\{\mathbf{h}_{\mathcal{V}}\} \xrightarrow{\mathcal{F}} \{\mathbf{h}_{\mathcal{A}_i}\} \xrightarrow{\mathcal{G}} \{\mathbf{h}_{\mathcal{S}_i}\} \xrightarrow{\mathcal{H}} \mathbf{z}_{\mathcal{G}}$.

The hierarchical decoder adopts an autoregressive coarse-to-fine generation paradigm, efficiently synthesizing molecular graphs through the stepwise assembly of structural motifs. Its core lies in a three-level coupled decision mechanism: First, based on the current molecular state and the latent vector $\mathbf{z}_{\mathcal{G}}$, the decoder samples a new motif $\mathcal{S}_t$ from a predefined vocabulary $V_{\mathcal{S}}$ via a motif selection module. This process uses an attention mechanism to align with the semantic information of the encoder's motif layer. Second, it performs attachment configuration prediction for the selected motif, determining the set of junction atoms $\mathcal{A}_t$ from a motif-specific vocabulary $V_{\mathcal{A}}(\mathcal{S}_t)$, significantly compressing the combinatorial search space. Finally, it connects the new motif to the current molecular graph through atom-level connection resolution, predicting bond types (single, double, or triple) to complete the local topological expansion.

## A.3 Drug Screener

De novo drug design provides a powerful generative paradigm for discovering structurally novel therapeutic molecules. However, translating abstract, computer-generated molecular structures into clinically viable drug candidates is a major challenge in drug discovery. Given that New Chemical Entities (NCEs) often require over a decade of development, with substantial financial investment and multiple stages of attrition risk from laboratory synthesis to market approval, we propose an innovative computational framework to bridge generative AI with the clinical translation pathway. The core objective of this framework is to establish a functional link between *de novo* generated molecules and reference compounds with established clinical data, thereby providing critical decision support for lead compound optimization.

The framework's workflow begins with transcriptomic data from pre- and post-disease states, which can be at the bulk or single-cell level. Pre-disease healthy state data can be sourced from a patient's own healthy tissue or from a standardized control group. Our conditional generative model learns and encodes the gene expression changes caused by the disease, subsequently generating a set of de novo molecular structures designed to reverse this cellular perturbation state.

These generated molecules, carrying specific therapeutic knowledge, are then used as query molecules. We designed a cascaded filter for rapidly identifying structurally similar analogs of these query molecules in a large compound library. The core of this filter is a pre-built molecular fingerprint database, where matches are found by performing a Top-K nearest neighbor search. The computational engine relies on the Tanimoto similarity coefficient to quantify the similarity between the query molecule's fingerprint vector ($\mathbf{f}_l$) and a database molecule's fingerprint vector ($\mathbf{f}_d$). Through this process, we can efficiently screen for a set of known compounds that are most structurally similar to the generated molecules, which are then considered potential drug candidates for the specific patient or disease state.

A.4 PSEUDOCODE FOR TRAINING LOSS

Training pseudocode of stage 1 and 2 are as following.

---

**Algorithm 1** Pseudocode for Stage-1 Loss $\mathcal{L}_{\text{vae}}$

---

1: **Input**: Graph matrix $\mathbf{X_G}$, feature extractor $F$, VAE encoder $Q(z|\mathbf{X_G})$, VAE decoder $P(\mathbf{X_G}|z)$, KL weight $\lambda_{\text{KL}}$
2: **Output**: $\mathcal{L}_{\text{vae}}$
3: $(\mu_{\text{enc}}, \sigma_{\text{enc}}) \leftarrow z_{\text{enc}} \leftarrow Q(\mathbf{X_G})$
4: $(\mu_f, \sigma_f) \leftarrow z_f \leftarrow F(\mathbf{X_G})$
5: $\mathcal{L}_{\text{vae-ELBO}} \leftarrow -\mathbb{E}_{z_{\text{enc}} \sim Q}[\log P(\mathbf{X_G}|z_{\text{enc}})] + \lambda_{KL} D_{KL}[Q(z_{\text{enc}}|\mathbf{X_G})||P(z_{\text{enc}})]$ ▷ Standard ELBO
6: $\mathcal{L}_{\text{vae-align}} \leftarrow ||\mu_{\text{enc}} - \mu_f||^2 + ||\sigma_{\text{enc}}^2 - \sigma_f^2||^2$
7: $\mathcal{L}_{\text{vae}} \leftarrow \mathcal{L}_{\text{vae-ELBO}} + \mathcal{L}_{\text{vae-align}}$
8: **return** $\mathcal{L}_{\text{vae}}$

---

**Algorithm 2** Pseudocode for Stage-2 Loss $\mathcal{L}_{\text{morgen}}$

---

1: **Input**: Predict vector $\mathbf{A}$, target fingerprint $\mathbf{B}$, SMILES label list $\mathbf{S}$, temperature $\tau = 0.1$, sparse weight $\lambda = 0.15$, $\alpha = 0.4$
2: **Output**: $\mathcal{L}_{\text{morgan}}$
3: $\mathcal{L}_{\text{reg}} \leftarrow \text{RegressionLoss}(\mathbf{A}, \mathbf{B}, \alpha)$
4: $\mathcal{L}_{\text{contrast}} \leftarrow \text{ContrastLoss}(\mathbf{A}, \mathbf{B}, \mathbf{S}, \tau, \lambda)$
5: $\mathcal{L}_{\text{morgan}} \leftarrow \mathcal{L}_{\text{reg}} + \mathcal{L}_{\text{contrast}}$
6: **return** $\mathcal{L}_{\text{morgan}}$
7: **function** CONTRASTLOSS($\mathbf{A}$, $\mathbf{B}$, $\mathbf{S}$, $\tau$, $\lambda$)
8:     **Input**: $\mathbf{A} \in \mathbb{R}^{b \times 2048}$, $\mathbf{B} \in \mathbb{R}^{b \times 2048}$ (non-negative integers), $\mathbf{S}$ (length $b$), $\tau$, $\lambda$
9:     **Output**: $\mathcal{L}_{\text{contrast}}$
10:     $\mathbf{A} \leftarrow \text{Normalize}(\mathbf{A})$
11:     $\mathbf{B} \leftarrow \text{Normalize}(\mathbf{B})$
12:     $\mathbf{Matrix} \leftarrow \mathbf{A} \cdot \mathbf{B}^\top / \tau$     ▷ Similarity matrix $\in \mathbb{R}^{b \times b}$
13:     $\mathbf{L} \leftarrow [0, 1, \ldots, b-1]$     ▷ Labels for diagonal elements
14:     $\mathbf{Mask}_{\text{same}} \leftarrow \text{Boolean}(S_i = S_j \text{ for all } i, j)$     ▷ Mask for same string labels
15:     $\mathbf{Mask}_{\text{same}}[\text{diagonal}] \leftarrow \text{False}$     ▷ Exclude diagonal
16:     $\mathbf{Matrix}[\mathbf{Mask}_{\text{same}}] \leftarrow -\infty$     ▷ Set non-diagonal same-label entries to large negative
17:     $\mathcal{L}_{\text{InfoNCE}} \leftarrow \text{CrossEntropy}(\mathbf{Matrix}, \mathbf{L})$
18:     **if** $\lambda > 0$ **then**
19:         $\mathbf{Mask}_{\text{zero}} \leftarrow (\mathbf{B} = 0)$     ▷ Mask for zero positions in $\mathbf{B}$
20:         $\mathcal{L}_{\text{sparse}} \leftarrow \text{Mean}((\mathbf{A} \cdot \mathbf{Mask}_{\text{zero}})^2)$
21:         $\mathcal{L}_{\text{contrast}} \leftarrow \mathcal{L}_{\text{InfoNCE}} + \lambda \cdot \mathcal{L}_{\text{sparse}}$
22:     **end if**
23:     **return** $\mathcal{L}_{\text{contrast}}$
24: **end function**
25: **function** REGRESSIONLOSS($\mathbf{A}$, $\mathbf{B}$, $\alpha$)
26:     **Input**: $\mathbf{A} \in \mathbb{R}^{b \times 2048}$, $\mathbf{B} \in \mathbb{R}^{b \times 2048}$ (non-negative integers), $\alpha$
27:     **Output**: $\mathcal{L}_{\text{reg}}$
28:     $\mathbf{W} \leftarrow \text{ZerosLike}(\mathbf{B})$     ▷ Initialize weight matrix
29:     $\mathbf{Mask}_{\text{pos}} \leftarrow (\mathbf{B} > 0)$     ▷ Non-zero position mask
30:     $\mathbf{Mask}_{\text{neg}} \leftarrow (\mathbf{B} = 0)$     ▷ Zero position mask
31:     $\mathbf{W}[\mathbf{Mask}_{\text{pos}}] \leftarrow \log(1 + \mathbf{B}[\mathbf{Mask}_{\text{pos}}])$     ▷ Weights for non-zero positions
32:     $\mathcal{L}_{\text{pos}} \leftarrow \text{Sum}(\mathbf{W} \cdot (\mathbf{A} - \mathbf{B})^2 \cdot \mathbf{Mask}_{\text{pos}}) / (\text{Sum}(\mathbf{Mask}_{\text{pos}}) + \epsilon)$
33:     $\mathcal{L}_{\text{neg}} \leftarrow \text{Sum}(\mathbf{A}^2 \cdot \mathbf{Mask}_{\text{neg}}) / (\text{Sum}(\mathbf{Mask}_{\text{neg}} + \epsilon))$
34:     $\mathcal{L}_{\text{reg}} \leftarrow \mathcal{L}_{\text{pos}} + \alpha \cdot \mathcal{L}_{\text{neg}}$
35:     **return** $\mathcal{L}_{\text{reg}}$
36: **end function**

---

Table 6: Generalization performance of scTrans-Gen. We assess the model's performance across three data splits representing different generalization challenges (all).

| Data Type | Split | Method | Coverage ↑ | Diversity ↑ | Similarity ↑ | Distance ↓ | SA ↓ | Unique ↑ | QED ↑ |
|---|---|---|---|---|---|---|---|---|---|
| Bulk | In-Distribution | GexMolGen | 6 (54.55%) | 0.7646 | 0.8919 | 35.4027 | 0.7216 | 0.4300 | 0.5127 |
| | | Gx2Mol | 8 (72.73%) | 0.8360 | 0.9405 | 17.8963 | 0.7151 | 0.1000 | 0.6041 |
| | | TRIOMPHE | 8 (72.73%) | 0.8809 | 0.7270 | 48.0169 | 0.4869 | - | 0.3071 |
| | | **scTrans-Gen** | **11 (100.00%)** | **0.8906** | **0.9576** | **6.7856** | **0.6386** | **0.9296** | **0.5665** |
| | Out-of-Distribution (Unseen Cells) | GexMolGen | 6 (54.55%) | 0.7622 | 0.8876 | 42.5445 | 0.7224 | 0.4100 | 0.5173 |
| | | Gx2Mol | 5 (45.45%) | 0.7321 | 0.7123 | 65.9671 | 0.7065 | 0.1000 | 0.6203 |
| | | TRIOMPHE | 7 (63.64%) | 0.8786 | 0.6637 | 56.0078 | 0.4941 | - | 0.3209 |
| | | **scTrans-Gen** | **10 (90.90%)** | **0.8864** | **0.8238** | **13.6113** | **0.6912** | **0.8590** | **0.5736** |
| | Out-of-Distribution (Unseen Drugs) | GexMolGen | 6 (54.55%) | 0.7609 | 0.9013 | 40.0122 | 0.7205 | 0.4200 | 0.5098 |
| | | Gx2Mol | 5 (45.45%) | 0.7280 | 0.7106 | 64.6032 | 0.7015 | 0.1000 | 0.6232 |
| | | TRIOMPHE | 7 (63.64%) | 0.8829 | 0.7324 | 54.3125 | 0.4870 | - | 0.3589 |
| | | **scTrans-Gen** | **10 (90.90%)** | **0.9018** | **0.9576** | **9.5265** | **0.6497** | **0.8657** | **0.5725** |
| Single-cell | In-Distribution | **scTrans-Gen** | **10 (90.91%)** | **0.8771** | **0.8137** | **27.6223** | **0.6984** | **0.8547** | **0.4946** |

# B  MORE EXPERIMENTAL RESULTS AND DISCUSSIONS

## B.1  QUALITY OF GENERATED MOLECULAR SETS (BASIC EVALUATION)

We used random conditions to guide molecule generation to create a set of molecules and discussed the generative capabilities of our model versus the baselines. The main focus is on the relationship between the generated and target molecular sets and the chemical and medicinal properties of the generated set itself, without delving into the accuracy of conditional control. On bulk data, since the L1000 dataset is constructed from the cross-interaction of n cell lines and m drugs, we compared the model's performance against baselines under different training splits: random, cell-masked, and drug-masked. On the single-cell dataset, due to the lack of comparable methods, we present the performance of our model on Tahoe-100M.

**Highlight Metrics.** We evaluated the performance of the generated molecules on several key metrics, comparing our new method (scTrans-Gen) with existing benchmarks (GexMolGen, Gx2Mol, and TRIOMPHE). Under three training strategies (random, cell, and drug), our method demonstrated significant advantages in overall molecular generation capabilities. Specifically, in terms of Coverage, our method achieved 100% coverage (11 heavy atom types) under random training, far surpassing other methods (54.55%–72.73%), reflecting its excellent ability to generate structurally diverse molecules. Similarly, on Diversity and Similarity metrics, our method showed high performance (Diversity up to 0.9018, Similarity up to 0.9576), while having the lowest Fréchet distance (only 6.7856 in random training), indicating that the generated molecules are structurally closer to the reference dataset. Furthermore, our method's Uniqueness score was consistently leading (e.g., 0.9296 in random training), significantly better than the comparison methods, highlighting its robustness and innovation in avoiding the generation of duplicate molecules.

**Detailed Analysis.** Although our method did not achieve the highest values in all scenarios for Synthesizability (SA) and Quantitative Estimate of Drug-likeness (QED), this is mainly due to the inherent flaws of the baseline methods. For instance, TRIOMPHE has a low SA score (0.4869 in random training), but its QED score is extremely low (0.3071), indicating that its generated molecules often lack drug-likeness and cannot form reasonable drug configurations. Similarly, Gx2Mol excels in QED (e.g., 0.6041), but its Unique score is extremely low (0.1000), suggesting a high degree of repetition in its generated molecules, making it unreliable for producing a diverse set of candidates. In contrast, our method maintains a balanced and excellent performance on both SA (lowest at 0.6386) and QED (highest at 0.5736), avoiding these pitfalls and ensuring a comprehensive balance between utility and diversity in the generated results.

**Training Methods.** Comparing the impact of different training methods, our model maintained a high level of performance across all settings. In random training, our model achieved 100% coverage and high diversity (0.8906), showing the best overall performance. In cell-masked training, coverage was 90.90% and diversity was 0.8864, while similarity (0.8238) was slightly lower than the peak values of other methods, which can be attributed to the instability of baselines like Gx2Mol (which has overly high similarity at 0.9405 but a very low Uniqueness score). In drug-masked training, our model's diversity further improved to 0.9018 and similarity reached 0.9576, while maintaining

a high Uniqueness of 0.8657. Overall, the stability and versatility of our method across random, cell-oriented, and drug-oriented training significantly exceed those of the baseline methods.

**Single-Cell Data.** This study is the first to extend a molecular generation model to a single-cell resolution dataset, Tahoe-100M, validating our method's applicability in complex biological environments and breaking the traditional model's dependency on bulk cell data. As shown in the table, our method maintains a significant advantage in the single-cell scenario: its Coverage (90.91%) remains leading, approaching full coverage of heavy atom types. Its Diversity (0.8771) and Uniqueness (0.8547) remain highly competitive, confirming the model's ability to stably generate non-redundant molecules from highly heterogeneous cellular data. Although the Distance (27.6223) to the reference set is larger compared to the bulk dataset results, this is due to the inherent technical limitations of single-cell data Tahoe-100M, as the first single-cell atlas used to validate molecular generation, has inherent gene expression sparsity and technical noise that significantly increase modeling complexity.

## B.2 ABLATION STUDIES FOR FEATURE EXTRACTOR

Table 7: Performance comparison on bulk (L1000) and single-cell (Tahoe-100M) datasets. ↑ indicates the higher the better, and ↓ indicates the lower the better (all).

| Data Type | Method | Validity↑ | Coverage↑ | Diversity↑ | Similarity↑ | Distance↓ | SA↓ | Unique↑ | QED↑ | Fraggle_Sim↑ | Morgan_Sim↑ | MACCS_Sim↑ |
|---|---|---|---|---|---|---|---|---|---|---|---|---|
| Bulk | w/o Extractor | 0.8775 | Cover 10 (90.91%) | 0.7504 | 0.9539 | 8.4982 | 0.8355 | 0.9153 | 0.5426 | 0.3942 | 0.1824 | 0.4640 |
| | w/o Alignment | 0.3000 | Cover 7 (63.64%) | 0.7662 | 0.6769 | 82.7183 | 0.7651 | 0.2857 | 0.4556 | 0.2991 | 0.0886 | 0.3826 |
| | w/o Interaction | 0.2400 | Cover 4 (36.36%) | 0.6982 | 0.7533 | 51.0830 | 0.6933 | 0.6452 | 0.4400 | 0.4327 | 0.2527 | 0.5464 |
| | scTrans-Gen | **0.9350** | **Cover 11 (100%)** | **0.8906** | **0.9576** | **6.7856** | **0.6386** | **0.9296** | **0.5665** | **0.8892** | **0.8228** | **0.9031** |
| Single-cell | w/o VAE | 0.9650 | Cover 9 (81.82%) | 0.8693 | 0.7331 | 43.0227 | 0.6043 | 0.8945 | 0.4588 | 0.4604 | 0.2674 | 0.5040 |
| | w/o Fingerprint | 0.9400 | Cover 9 (81.82%) | 0.8600 | 0.8013 | 29.6223 | 0.6357 | 0.8704 | 0.3048 | 0.5819 | 0.3219 | 0.6693 |
| | scTrans-Gen | **0.9800** | **Cover 10 (90.91%)** | **0.8771** | **0.8137** | **27.6223** | **0.5994** | **0.9082** | **0.4946** | **0.7310** | **0.6114** | **0.7590** |

The experimental results show that the full model (scTrans-Gen) significantly outperforms the control groups on key metrics. Compared to using raw L1000 Level 5 data directly(w/o Extractor), the microscopic structural similarity metrics improved markedly. The domain alignment module plays a decisive role in cross-domain feature mapping; its absence leads to a catastrophic drop in performance, both in overall molecular set metrics and similarity metrics. The transcriptome interaction module greatly improves the completeness of difference feature extraction through the interaction of transcriptome pairs.

For the complex characteristics of single-cell data, the dual-domain alignment mechanism fusing molecular fingerprints and VAE graph space is core to ensuring the model's alignment capability. Relying solely on molecular fingerprints led to a sharp decline in macroscopic similarity (0.8137 - 0.7331), while using only VAE alignment caused a catastrophic drop in drug-likeness (QED: 0.4946 - 0.3048, a 38.4% decrease), confirming the complementarity of chemical and topological representations. In the single-cell scenario, the absence of molecular fingerprints severely weakened structural similarity (MACCS_Similarity plummeted by 44.3%), while the absence of VAE significantly harmed biological plausibility (the QED value revealed a deterioration in the drug-likeness of generated molecules). The dual-domain alignment mechanism, by strengthening chemical structure features and molecular topology constraints, enables the model to maintain high structural fidelity even with the interference of transcriptome noise.

## B.3 IMPACT OF CFG GUIDANCE STRENGTH

We explored the effect of different CFG guidance strengths on the generation results. We found that as the guidance strength increases, the performance metrics for molecular generation first rise and then slightly decline. This reveals a trade-off between potency and drug-likeness, providing a basis for selecting the optimal hyperparameter in practical applications. The experiment establishes a strength of 3 as the global optimum. The CFG strength acts as a lever to control potency and drug-likeness, low strength leads to insufficient potency, while high strength causes structural distortion. This finding provides a general theoretical framework for hyperparameter optimization in drug generation tasks. Unlike the previous experiments, which were tested on the entire dataset, this data was tested on a single batch (batchsize=200) to show the trend of the metrics.

Table 8: Performance of training w/o cfg under different guidance scale. ↑ means higher is better, ↓ means lower is better.

| Metric | train w/o cfg | 0 | 1 | 2 | 3 | 4 | 5 | 6 | 7 | 8 | 9 |
|---|---|---|---|---|---|---|---|---|---|---|---|
| Validity↑ | 0.1934 | 0.0550 | 0.7025 | 0.7250 | **0.7500** | 0.6950 | 0.6950 | 0.6800 | 0.6750 | 0.6750 | 0.6300 |
| Coverage↑ | Cover 7 | Cover 6 | Cover 8 | Cover 8 | **Cover 10** | Cover 9 | Cover 9 | Cover 9 | Cover 8 | Cover 8 | Cover 7 |
| Diversity↑ | 0.7942 | 0.7222 | 0.7946 | 0.8007 | 0.8013 | 0.7984 | **0.8027** | 0.7998 | 0.8009 | 0.8011 | 0.8003 |
| Similarity↑ | 0.8691 | 0.7664 | **0.9598** | 0.9589 | 0.9581 | 0.9571 | 0.9573 | 0.9564 | 0.9573 | 0.9564 | 0.9574 |
| Distance↓ | 28.9203 | 45.4423 | 10.5358 | **9.3103** | 9.2467 | 10.4150 | 9.8974 | 10.2288 | 9.9570 | 10.0778 | 10.7401 |
| Fraggle_Sim↑ | 0.3441 | 0.3289 | 0.8734 | 0.8708 | **0.8896** | 0.8785 | 0.8705 | 0.8843 | 0.8800 | 0.8654 | 0.8600 |
| Morgan_Sim↑ | 0.2090 | 0.1281 | 0.8036 | 0.8076 | **0.8279** | 0.8187 | 0.8083 | 0.8239 | 0.8024 | 0.7906 | 0.8029 |
| MACCS_Sim↑ | 0.3799 | 0.3866 | 0.8693 | 0.8661 | **0.8866** | 0.8787 | 0.8720 | 0.8860 | 0.8782 | 0.8674 | 0.8610 |
| QED↑ | 0.4952 | 0.4435 | 0.5435 | 0.5629 | **0.5673** | 0.5635 | 0.5650 | 0.5664 | 0.5671 | 0.5672 | 0.5632 |
| SA↓ | 0.6148 | **0.5468** | 0.6684 | 0.6708 | 0.6712 | 0.6670 | 0.6655 | 0.6626 | 0.6554 | 0.6542 | 0.6561 |
| Unique↑ | 0.9275 | **1.0000** | 0.9395 | 0.9300 | 0.9276 | 0.9281 | 0.9317 | 0.9338 | 0.9370 | 0.9370 | 0.9325 |

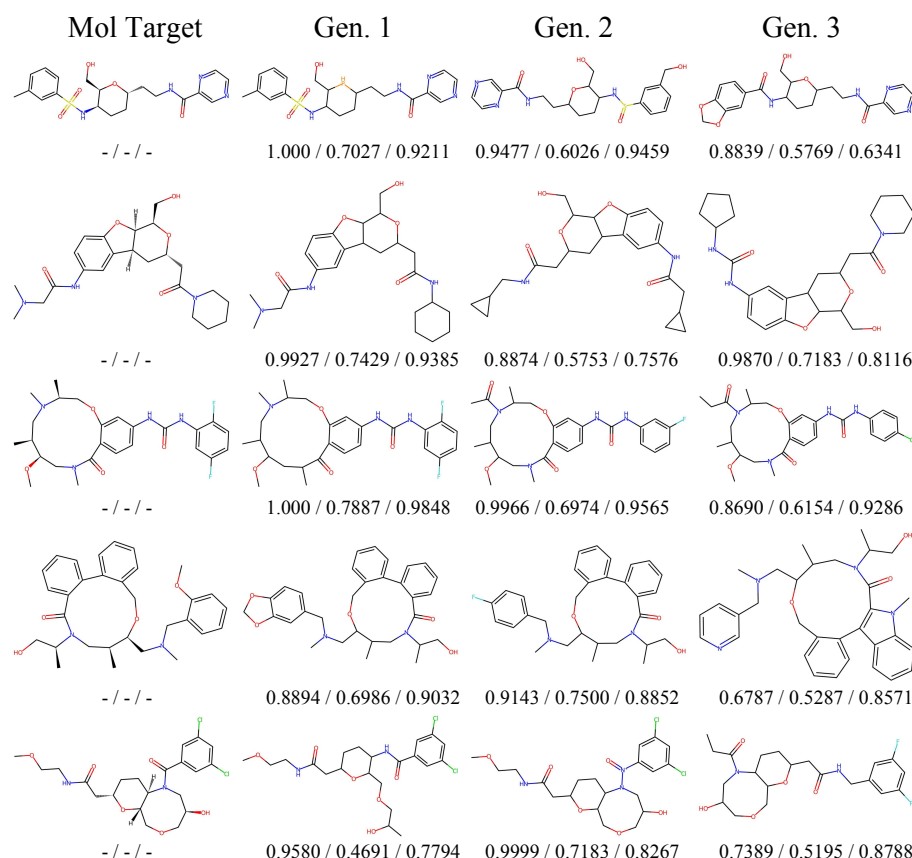

Figure 6: Molecular structure diagrams generated through multiple sampling. *Note: The indicators in the chart represent, from left to right: Fraggle/Morgan/MACCS scores.*

### B.4 MOLECULAR STRUCTURE VISUALIZATION

To further confirm that the high similarity is not a simple copy, we randomly sampled several real drugs from the test set and compared them with multiple molecules generated by the model through multiple samplings (Figure 6, Table 9). Although the Fraggle/Morgan/MACCS scores were high, their SMILES strings were not identical. The differences mainly lay in the backbone modifications and peripheral groups, rather than a simple copy of the training molecules. Meanwhile, the high similarity between the model-generated molecules and the target molecules stemmed from their similar functional group structures. Similar functional groups imply a high degree of functional approximation, reflecting the effectiveness of the model in function extraction.

Table 9: Molecular SMILE expressions and similarities generated from multiple sampling

| Target SMILES | Generated SMILES | Fraggle ↑ | Morgan ↑ | MACCS ↑ |
|---|---|---|---|---|
| Cc1cccc(c1)S(=O)(=O)N[C@@H]1CC[C@@H](CCNC(=O)c2cnccn2)O[C@@H]1CO | Cc1cccc(S(=O)(=O)NC2CCC(CCNC(=O)c3cnccn3)PC2CO)c1 | 1.000 | 0.7027 | 0.9211 |
| | O=C(NCCC1CCC(NS(=O)c2cccc(CO)c2)C(CO)O1)c1cnccn1 | 0.9477 | 0.6026 | 0.9459 |
| | O=C(NC1CCC(CCNC(=O)c2cnccn2)OC1CO)c1ccc2c(c1)OCO2 | 0.8839 | 0.5769 | 0.6341 |
| CN(C)CC(=O)Nc1ccc2O[C@@H]3[C@@H](C[C@@H](CC(=O)N4CCCCC4)O[C@@H]3CO)c2c1 | CN(C)CC(=O)Nc1ccc2c(c1)C1CC(CC(=O)NC3CCCCC3)OC(CO)C1O2 | 0.9927 | 0.7429 | 0.9385 |
| | O=C(CC1CC2c3cc(NC(=O)CC4CC4)ccc3OC2C(CO)O1)NCC1CC1 | 0.8874 | 0.5753 | 0.7576 |
| | O=C(Nc1ccc2c(c1)C1CC(CC(=O)N3CCCCC3)OC(CO)C1O2)NC1CCCC1 | 0.9870 | 0.7183 | 0.8116 |
| CO[C@@H]1CN(C)C(=O)c2ccc(NC(=O)Nc3cc(F)ccc3F)cc2OC[C@H](C)N(C)C[C@@H]1C | COC1CC(C)C(=O)c2ccc(NC(=O)Nc3cc(F)ccc3F)cc2OCC(C)N(C)CC1C | 1.000 | 0.7887 | 0.9848 |
| | COC1CN(C)C(=O)c2ccc(NC(=O)Nc3cccc(F)c3)cc2OCC(C)N(C(C)=O)CC1C | 0.9966 | 0.6974 | 0.9565 |
| | CCC(=O)N1CC(C)C(OC)CN(C)C(=O)c2ccc(NC(=O)Nc3ccc(Cl)cc3)cc2OCC1C | 0.8690 | 0.6154 | 0.9286 |
| COc1ccccc1CN(C)C[C@@H]2OCc3ccccc3-c4ccccc4C(=O)N(C[C@@H]2C)[C@@H](C)CO | CC1CN(C(C)CO)C(=O)c2ccccc2-c2ccccc2COC1CN(C)Cc1ccc2c(c1)OCO2 | 0.8894 | 0.6986 | 0.9032 |
| | CC1CN(C(C)CO)C(=O)c2ccccc2-c2ccccc2COC1CN(C)Cc1ccc(F)cc1 | 0.9143 | 0.7500 | 0.8852 |
| | CC1CN(C(C)CO)C(=O)c2c(c3ccccc3n2C)-c2ccccc2COC1CN(C)Cc1cccnc1 | 0.6787 | 0.5287 | 0.8571 |
| COCCNC(=O)C[C@@H]1CC[C@@H]2[C@H](COC[C@H](O)CN2C(=O)c2cc(Cl)cc(Cl)c2)O1 | COCCNC(=O)CC1CCC(NC(=O)c2cc(Cl)cc(Cl)c2)C(COCC(C)O)O1 | 0.9580 | 0.4691 | 0.7794 |
| | COCCNC(=O)CC1CCC2C(COC(O)CN2[N+](=O)c2cc(Cl)cc(Cl)c2)O1 | 0.9999 | 0.7183 | 0.8267 |
| | CCC(=O)N1CC(O)COCC2OC(CC(=O)NCc3cc(F)cc(F)c3)CCC21 | 0.7389 | 0.5195 | 0.8788 |

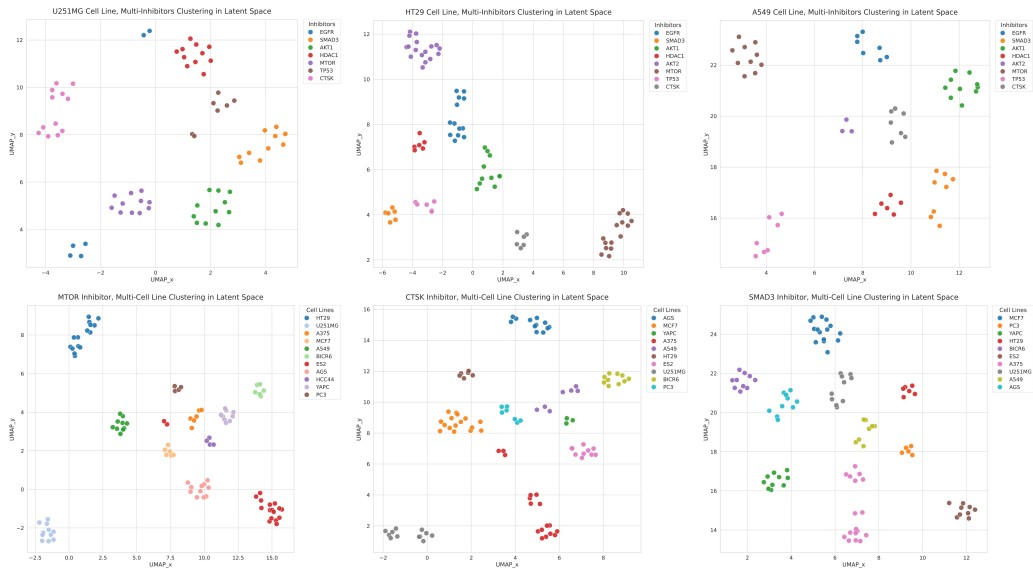

Figure 7: Latent space visualization using the human gene inhibitor dataset. UMAP projections reveal the model's ability to disentangle biological mechanisms from cellular contexts. (Top) Distinct clustering of different inhibitors within the same cell line demonstrates the encoding of mechanism-specific functional signatures. (Bottom) Stratification of identical inhibitors across diverse cell lines confirms the model's sensitivity to cellular heterogeneity.

## B.5    LATENT SPACE ANALYSIS AND VISUALIZATION OF BIOLOGICAL INTERPRETABILITY

Since scTrans-Gen operates within the framework of Transcriptome-based Drug Design (TBDD), it lacks explicit indicators for target binding affinity. To rigorously investigate the biological interpretability of the model and verify whether the learned latent representations capture authentic mechanistic principles rather than mere statistical artifacts, we conducted a stratified visualization analysis using Uniform Manifold Approximation and Projection (UMAP) on the human gene inhibitor dataset. Our visualization strategy was designed to probe the latent space from two complementary perspectives: functional specificity and biological context sensitivity.

First, to validate the model's capability to encode mechanism-specific functional signatures, we isolated the cellular background by projecting latent embeddings of distinct gene inhibitors within a single cell line (U251MG, HT29, A549). As illustrated in the top row of Figure 7, the resulting manifold reveals a striking structural organization where samples form discrete, tight clusters according to the inhibitor type. This distinct separation implies that the model effectively extracts and encodes the unique transcriptomic perturbations associated with specific therapeutic targets, effectively mapping phenotypic changes to their underlying Mechanisms of Action (MoA).

Second, to demonstrate that the model maintains sensitivity to cellular heterogeneity and is not overfitting to a generic drug signature, we visualized the embeddings of identical inhibitors (MTOR, CTSK, SMAD3) across diverse cell lines. The bottom row of Figure 7 exhibits clear stratification driven by cellular identity, confirming that the model dynamically adapts its functional representations based on the biological context.

Collectively, these visualization results provide strong evidence for the model's validity. The ability to simultaneously achieve high intra-class compactness for inhibitors (demonstrating mechanistic understanding) and inter-class separability for cell lines (demonstrating context awareness) proves that the intermediate latent space operates as a biologically meaningful manifold. This indicates that our framework successfully disentangles the specific functional impact of drug perturbations from complex cellular background effects, establishing a robust foundation for function-oriented drug discovery.

### B.6 EVALUATION OF TOXICITY PROPERTIES

To further assess the pharmacological viability and safety profile of the generated compounds, we extended our evaluation to include toxicity-related properties using the ADMETlab predictor (Fu et al., 2024). We compared molecules generated by scTrans-Gen against those from baseline methods (GexMolGen, Gx2Mol, TRIOMPHE) as well as the ground-truth drugs from the L1000 test set. The evaluation covers a broad spectrum of toxicity risks, including mutagenicity (Ames), cardiotoxicity (hERG), and organ-specific toxicities. The results, summarized in Table 10, demonstrate that scTrans-Gen achieves a competitive safety profile. Although our model was not explicitly optimized for these specific toxicity during training, the generated molecules exhibit toxicity scores that are consistently within a reasonable range, often matching or outperforming both the baseline methods and the ground-truth reference drugs (e.g., in Eye Irritation and Rat Oral Acute Toxicity).

Table 10: Comparison of predicted toxicity properties across generative models and the ground truth (L1000). Arrows indicate whether lower ($\downarrow$) or higher ($\uparrow$) scores are desirable.

| Metric | GexMolGen | Gx2Mol | TRIOMPHE | scTrans-Gen | L1000 (GT) |
|---|---|---|---|---|---|
| Ames Mutagenicity $\downarrow$ | 0.5003 | 0.4632 | 0.5965 | 0.4850 | 0.5527 |
| hERG Blockers ($10\mu$M) $\downarrow$ | 0.4361 | 0.4236 | 0.4003 | **0.3983** | 0.2973 |
| Hematotoxicity $\downarrow$ | 0.4435 | **0.3436** | 0.3993 | 0.3590 | 0.5149 |
| Respiratory Toxicity $\downarrow$ | 0.4755 | 0.5332 | 0.8141 | **0.4684** | 0.4715 |
| Carcinogenicity $\downarrow$ | 0.4523 | **0.4511** | 0.5439 | 0.4714 | 0.5345 |
| DILI (Liver Injury) $\downarrow$ | 0.6968 | 0.6923 | 0.6623 | **0.6506** | 0.6733 |
| ROA (Rat Oral Acute Tox.) $\downarrow$ | 0.3475 | 0.3331 | 0.6658 | **0.3214** | 0.3414 |
| FDAMDD (Max Daily Dose) $\uparrow$ | 0.4875 | 0.5498 | **0.6136** | 0.5918 | 0.5272 |
| Eye Irritation $\downarrow$ | 0.1983 | 0.2082 | 0.2682 | **0.0942** | 0.2238 |
| Eye Corrosion $\downarrow$ | 0.0311 | 0.0264 | 0.1485 | **0.0141** | 0.0124 |

## C MORE EXPERIMENTAL DETAILS

### C.1 MODEL TRAINING SETUP

We provide a comprehensive description of the model architecture complexity and the specific hyperparameter settings used during the training phases(Table 11, Table 12). For full reproducibility, we refer readers to the specific configuration files available in our source code repository.

Table 11: Summary of Model Parameters.

| Component | Description | Parameters |
|---|---|---|
| Diffusion Model | Graph Diffusion Transformer | $\sim 501.0$ M |
| Feature Extractor | Feature extraction and alignment modules | $\sim 7.8$ M |
| Graph VAE Encoder | Encodes molecular graphs | $\sim 2.4$ M |
| Graph VAE Decoder | Reconstructs molecular graphs | $\sim 2.9$ M |

Table 12: Hyperparameter Settings for Training Phases.

| Hyperparameter | Alignment Phase | Diffusion Phase |
|---|---|---|
| Hardware | NVIDIA A100 (40GB) | NVIDIA A100 (40GB) |
| Total Training Time | $\sim 15$ GPU hours | $\sim 48$ GPU hours |
| Training Steps | 30k | 40k |
| Batch Size | 64 | 400 |
| Learning Rate | $1 \times 10^{-4}$ | $2 \times 10^{-4}$ |
| Optimizer | Adam | Adam |
| Diffusion Steps ($T$) | – | 500 |

## C.2 DATA INTEGRITY AND PREVENTION OF LEAKAGE.

To ensure the validity of our evaluation and the generalization capability of the model, we strictly enforced data isolation protocols across all learnable modules. The Molecular Graph VAE and the multi-domain feature extractor were trained exclusively on the designated training splits of the TBDD dataset, with no exposure to molecules or transcriptomes from the validation or test sets. Regarding the use of SCimilarity, it serves solely as a generic, frozen dimensionality-reduction tool. It was pre-trained on a broad human cell atlas for general cell-state embedding and was not fine-tuned on our L1000, Tahoe-100M, or ExCAPE datasets. Thus, it contains no task-specific supervision regarding drug-perturbation mappings. Similarly, the Morgan fingerprint alignment relies on deterministic RDKit computations without learning. These rigorous measures ensure that the model's performance stems from learning authentic structure-function mappings rather than data leakage or memorization.

