# OpenReview forum: "From Genomic Whispers to Therapeutics: Multi-Resolution Transcriptome-Guided Diffusion Models for Drug Design and Screening"
_ICLR.cc/2026/Conference — ICLR 2026 Conference Withdrawn Submission_

### Official Review · Reviewer_ZVqr · 2025-10-25

**Soundness:** 2
**Presentation:** 2
**Contribution:** 3
**Rating:** 6
**Confidence:** 3

**Summary:**

The paper introduces scTrans gen, a diffusion model for drug design conditioned on transcriptomic data rather than protein structures. Given pre- and post-perturbation transcriptomes, the model generates molecules that induce that cellular response.
The model uses a "Transcriptome Pseudoimage" module to handle sparse single-cell data (converts gene expression to images processed by a vision encoder), a transcriptome interaction module to extract perturbation features, and a graph diffusion generator with multi-domain alignment to molecular fingerprints and graph VAE representations via two-stage training.
They evaluate on L1000 (bulk, ~20k drugs) and Tahoe-100M (single-cell, 300+ drugs). Results show large improvements over baselines on structural similarity (Fraggle Sim: 0.89 vs 0.33, Morgan Sim: 0.82 vs 0.11). They also demonstrate zero-shot gene inhibitor design and a screening workflow. The paper claims this avoids the "ill-posed" inverse mapping problem by learning a "function-oriented chemical space" as an intermediate representation.

**Strengths:**

* Substantial improvements over baselines (e.g., Fraggle Sim: 0.89 vs 0.33, Morgan Sim: 0.82 vs 0.11 for GexMolGen). Evaluation spans macro (validity/QED/diversity/FCD) and micro (Fraggle/Morgan/MACCS) plus a screening workflow.
* The model is complex, but there's good ablation studies

**Weaknesses:**

* Presentation issues:
	* Figure 4 is hard to read. Why is this not just another table?
	* Table 1, the red diff numbers are distracting and add no information
	* Table 4 is unmentioned in the text. Instead the text points to Table 6 for ablation?
* Their model is very complex. Multi stage training, complex loss functions, multiple pretrained parts. This limits reproducibility, and scalability, and makes modifications harder. I know this is not actionable feedback, but I still wonder if a simpler architecture using the same insights wouldn't have worked similiarly well (I've seen the ablations).
* I don't think that forcing the generation through a feature extraction bottleneck makes the problem any less ill-posed. It may make the model easier to evaluate, and maybe improves scores but it doesn't fix the fundamental issue?
* I would've liked to see an ablation of the Image Encoder. This component seem ad-hoc to me. Why is a vision encoder appropriate for gene expression patterns?

**Questions:**

* "de novo generation may be too slow for urgent clinical needs". How long does the de novo generation take? Even if it's in the few-minutes range, I find it hard to imagine a clinical scenario where that could possibly be the bottleneck?
* You align to a Mol Graph VAE and Morgan fingerprints (Fig. 3). Were those components pretrained on any molecules that appear in our eval splits? Is the extractor pretrained on a larger corpus or only on the train split here?
* TRIOMPHE has very bad Morgan Sim. Was this number taken from the prior paper or did the authors reproduce it themselves?
* What was SCimilarity trained on? Is there leakage with the test set?

---

> ### Author Response · Authors · 2025-11-19
>
> We sincerely thank the reviewer for the constructive and encouraging feedback, especially for recognizing the strong empirical performance and ablation studies of **scTrans-Gen**. Below, we address each concern in detail.
>
> ---
>
> ### Weakness 1 — Presentation Issues (Figure 4, Table 1, Table 4)
>
> **Figure 4: readability and table alternative**
>
> We agree that the current version of Figure 4 is dense and difficult to read. In the revision, we will:
>
> * Enlarge the fonts and labels and simplify the visual design (fewer visual elements, clearer legends).
> * Provide an equivalent numeric table in the appendix so that readers can easily look up exact values.
> * If space permits, we will replace the current figure in the main text with a simplified bar-plot version and keep the full detailed table in the appendix.
>
> **Table 1: red “diff” numbers**
>
> The red “diff” values in Table 1 were originally intended to help readers quickly see the improvements over baselines. We agree that the red color is visually distracting. In the revised version, we will:
>
> * Remove the red coloring and either show the differences in standard black font or move them to the appendix.
> * Summarize the key improvements in the main text with short sentences instead of visually heavy in-table annotations.
>
> **Table 4: missing reference in the text**
>
> You are correct that there is an inconsistency: the ablation discussion currently points to Table 6 while the main feature-extractor ablation results are in Table 4. This is an editing oversight. In the revision, we will:
>
> * Explicitly refer to **Table 4** in the corresponding ablation paragraph.
> * Remove redundant references to Table 6 or merge redundant content so that each experiment is consistently mapped to a single main table.
>
> ---
>
> ### Weakness 2 — Model Complexity, Reproducibility, and Scalability
>
> We appreciate the reviewer’s concerns about model complexity.
>
> **Core vs. auxiliary components**
>
> In the current text we did not sufficiently emphasize which components are essential and which are supporting. To clarify:
>
> * The **core generative capacity** comes from the **graph diffusion model** (a DiT-based graph diffusion backbone).
> * The other modules — **Transcriptome Pseudoimage**, **Transcriptome Interaction**, and the **multi-domain alignment heads** — are **conditioning and alignment modules**, whose role is to encode perturbation signals and bridge the transcriptome and molecular spaces for the TBDD task.
> * The loss functions, while numerous in notation, are all organized around this single goal: aligning functional transcriptome features to a chemical representation that can be used for conditional graph diffusion.
>
> We will revise Section 4 to explicitly present scTrans-Gen as a “graph diffusion backbone + modular conditioning stack”, which should make the conceptual structure clearer.
>
> **Reproducibility and modularity**
>
> To address reproducibility and scalability:
>
> * We have already released anonymized code, including full training scripts, configuration files, and hyperparameters. We also plan to release trained weights after the review process, to further lower the barrier for reproduction and adaptation.
> * All “complex” components (VAE, graph encoder, alignment heads) are implemented in a **modular** way; they can be replaced by simpler alternatives (e.g., shallower encoders or simplified alignment layers) with minimal code changes.
> * Our ablations show that each component provides measurable gains. At the same time, when computational resources are limited, the model can be run with a reduced configuration (e.g., fewer alignment stages) while still accomplishing the core TBDD task.
>
> We will highlight this modular design and provide concrete configuration examples in the appendix.

---

> ### Author Response · Authors · 2025-11-19
>
> ### Weakness 3 — Ill-posedness and the “Function-Oriented Bottleneck”
>
> We appreciate the reviewer’s deep question regarding the ill-posed nature of mapping transcriptomes to molecular structures.
>
> We fully agree that **the underlying inverse problem is ill-posed**: macroscopic, noisy transcriptomic responses cannot uniquely determine a precise molecular structure. Our work does **not** claim to eliminate this ill-posedness. Instead, we aim to **reformulate and regularize** the problem via a function-oriented intermediate space:$z = E_\phi(T_{\text{pre}}, T_{\text{post}}), \quad M \sim p_\theta(M \mid z).$
>
> Rather than learning a direct mapping “transcriptome → unique structure”, we:
>
> * Map $(T_{\text{pre}}, T_{\text{post}})$ to a **functional embedding** ($z$) that captures the perturbation signature in a chemical-friendly latent space.
> * Let the diffusion model **sample** multiple candidate structures from $p_\theta(M \mid z)$, naturally supporting **one-to-many mappings**: one functional pattern to many structurally diverse but functionally similar molecules.
>
> Thus, our claim is not that we “fix” ill-posedness, but that we **avoid the most brittle form of the inverse problem** (direct end-to-end reconstruction) and instead “re-express” it in a way that better supports generalization and functional equivalence (as evidenced by OOD splits and gene inhibitor experiments).
>
> We will revise the paper to:
>
> * Replace wording such as “avoid the ill-posed inverse mapping problem” with more careful phrasing like “mitigate the brittleness of direct inverse mapping by introducing a function-oriented intermediate space”.

---

> ### Author Response · Authors · 2025-11-19
>
> ### Weakness 4 — Image Encoder / Pseudoimage Design
>
> We appreciate the reviewer’s skepticism about the Transcriptome Pseudoimage mechanism and the use of a vision-style encoder.
>
> #### Motivation: between pseudo-bulk and single-cell resolution
>
> Single-cell data suffer from:
>
> * Extremely high dimensionality (>60k genes) and >90% sparsity;
> * Strong technical noise and batch effects.
>
> Standard **pseudo-bulk** approaches reduce noise by aggregation but collapse cell-to-cell heterogeneity[1]. Our **Transcriptome Pseudoimage** is designed as a compromise:
>
> 1. Use **SCimilarity** to map each cell from a sparse 60k+ dimensional count vector to a 128-d dense embedding.
> 2. Group cells into biologically meaningful clusters (by **cell-cycle phase**) and aggregate within each group to form “mini pseudo-cells”.
> 3. Arrange these mini pseudo-cells into a 2D pseudoimage to expose local spatial structure corresponding to heterogeneity across cell-cycle stages and cell states.
>
> This retains some single-cell heterogeneity while still denoising through local averaging.
>
> #### Why a vision encoder?
>
> We do **not** treat gene expression as natural images. Instead, we exploit the fact that a large, pretrained encoder is very effective at extracting **low-dimensional, information-dense features from 2D arrays**[2], especially when finetuned:
>
> * We finetune the pretrained encoder on our pseudoimage task so it specializes to transcriptome-derived pseudoimages rather than natural images.
> * Empirically, this combination yields more stable training and better performance than training from scratch or using simpler encoders.
>
> We have added an ablation over encoders on the single-cell (Tahoe) setting. A condensed version is below (detailed version in the appendix):
>
> | Method                            | Coverage ↑       | Diversity ↑ | Similarity ↑ | Distance ↓ | SA ↓ | Unique ↑ | QED ↑ | Fraggle Sim ↑ | Morgan Sim ↑ | MACCS Sim ↑ |
> | --------------------------------- | ---------------- | ----------- | ------------ | ---------- | ---- | -------- | ----- | ------------- | ------------ | ----------- |
> | w/o pseudoimage                   | 7 (63.64%)       | 0.75        | 0.74         | 26.89      | 0.61 | 0.34     | 0.59  | 0.23          | 0.07         | 0.23        |
> | MLP                               | 7 (63.64%)       | 0.79        | 0.78         | 13.26      | 0.68 | 0.45     | 0.55  | 0.45          | 0.29         | 0.46        |
> | Conv (from scratch)               | 9 (81.82%)       | 0.84        | 0.90         | 8.15       | 0.64 | 0.88     | 0.57  | 0.74          | 0.69         | 0.75        |
> | w/o pretrain (same arch)          | 10 (90.91%)      | 0.87        | 0.94         | 6.89       | 0.64 | 0.90     | 0.56  | 0.86          | 0.80         | 0.89        |
> | **Ours (pretrained + finetuned)** | **11 (100.00%)** | **0.89**    | **0.96**     | **6.79**   | 0.64 | **0.93** | 0.57  | **0.89**      | **0.82**     | **0.90**    |
>
> Two important observations:
>
> * The **pretrained+finetuned encoder** consistently outperforms from-scratch CNN/MLP alternatives in both **macro** (Coverage, Unique) and **micro** (Fraggle, Morgan, MACCS) metrics.
> * Training a comparable encoder **from scratch** converges much more slowly and is significantly less stable, especially on the large and noisy Tahoe-100M dataset.
>
> We will add this table to the appendix and clarify in the main text that the choice of a pretrained encoder was driven by **empirical performance, stability, and convergence speed**, not by ad-hoc intuition alone.
>
> #### Cell-cycle–stratified sampling vs. random sampling
>
> We also investigated the effect of **cell-cycle–stratified sampling** when constructing pseudoimages[3]. In the paper we use **cell-cycle–stratified sampling** as our default strategy, and we will explicitly state this in the revision.
>
> We compared random sampling and cell-cycle–stratified sampling (both with the same total number of cells per pseudoimage). Random sampling yields lower structural similarity (e.g., Morgan ≈ 0.50), while cell-cycle–stratified sampling reaches Morgan ≈ 0.61. This supports the intuition that preserving known biological structure (cell-cycle phases) makes the encoder’s job easier and improves downstream generation.

---

> ### Author Response · Authors · 2025-11-19
>
> ### Question 1 — “de novo generation may be too slow for urgent clinical needs”
>
> We appreciate this opportunity to clarify. Our intention was not to suggest that **computation** is the bottleneck in urgent clinical scenarios:
>
> * Given a condition $(T_{\text{pre}}, T_{\text{post}})$, generating a sufficient number of candidate molecules on a modern GPU is **fast** (seconds to a few minutes).
> * The **true latency** in clinical translation comes from **downstream wet-lab steps**: route design, compound synthesis, in vitro and in vivo testing, and safety evaluation, which can take months or years.
>
> The motivation for the **generate–then–search** workflow is therefore:
>
> * When de novo synthesis and validation cannot meet an urgent time window, use the generated molecules as **functional queries** to search large existing drug libraries.
> * This allows clinicians or researchers to rapidly identify **available or quickly accessible** compounds that are functionally close to the desired perturbation.
>
> We will revise the introduction to explicitly state that the bottleneck is in laboratory and regulatory timelines, not in model inference time.
>
> ---
>
> ### Question 2 — Pretraining of Mol Graph VAE, Morgan Alignment, and Extractor
>
> Thank you for raising the concern about potential data leakage.
>
> * The **molecular graph VAE** is trained **only on the training-set molecules** from our TBDD setting; it is not pretrained on any molecules that appear exclusively in the validation or test splits.
> * The **Morgan fingerprint alignment** uses fingerprints computed with RDKit, which do not involve learning and therefore do not introduce leakage.
> * The **feature extractor** (including the multi-domain alignment) is trained only on the training split and does not see validation/test pairs during training.
>
> We do **not** use any external labeled datasets containing our evaluation molecules or transcriptomes to pretrain these components. In the revision, we will add a “Data Usage & Pretraining” subsection explicitly listing, for each module, which data it sees and at what stage, to make the absence of leakage fully transparent.
>
> ---
>
> ### Question 3 — TRIOMPHE Morgan Similarity Evaluation
>
> All baseline results, including TRIOMPHE’s Morgan similarity, are **our own reproductions** under the TBDD setting, not copied from prior papers.
>
> * We use the **official implementations, weights, and recommended hyperparameters** provided by the authors where available.
> * Our task, data splits, and metric suite differ from the original TRIOMPHE setting (e.g., L1000/Tahoe perturbation-based conditional generation, larger diversity and function-oriented metrics), so we cannot directly compare to the numbers reported in their paper.
>
> As discussed in our response to other reviewers, many baseline methods generate **highly repetitive structures** for different transcriptomic conditions in our large-scale setting. This leads to:
>
> * Significantly reduced **Unique** scores,
> * And indirectly affects structural and functional metrics, including Morgan similarity.
>
> To increase transparency, we will:
>
> * Describe our training and evaluation protocol for baselines,
> * And explain how our setting differs from their original evaluation.

---

> ### Author Response · Authors · 2025-11-19
>
> ### Question 4 — SCimilarity Training Data and Leakage
>
> Finally, we address the question about SCimilarity.
>
> * **SCimilarity** is a large-scale foundation model for scRNA-seq/snRNA-seq embeddings, trained on millions of cells from numerous studies and tissue types, primarily to enable **search over similar cells** in a broad human cell atlas[4].
> * In our work, SCimilarity is used **only as a frozen encoder** to map raw single-cell expression profiles to dense 128-d embeddings. We **do not finetune it on our TBDD data**, and it is not trained on L1000, Tahoe-100M, or ExCAPE perturbation datasets.
>
> To avoid leakage in our evaluation:
>
> * For the **Unseen Cell Lines** and gene-inhibitor experiments, we choose cell lines and contexts that are not present in the SCimilarity training data (to the best of our knowledge based on the published description).
> * Even if there is some overlap at the level of broad tissue types, SCimilarity provides only a **generic cell-state embedding**; all **drug- and perturbation-specific structure–function mappings** are learned exclusively by our feature extractor and diffusion model on the TBDD training data.
>
> Thus, SCimilarity serves as a **generic dimensionality-reduction tool** for single-cell profiles, not as a source of task-specific supervision, and does not leak information about our test perturbations or generated molecules.
>
> ---
>
> We again thank Reviewer ZVqr for the thoughtful comments and for acknowledging the strength of our empirical results and ablations. In the revised version, we will:
>
> * Clarify the conceptual structure of the model and the role of the function-oriented space,
> * Improve figure/table presentation and cross-referencing,
> * Make data usage and pretraining choices fully transparent,
> * And include the added encoder/pseudoimage ablations and sampling analyses discussed above.
>
> We believe these revisions will significantly improve the clarity, transparency, and perceived robustness of scTrans-Gen.
>
> ---
>
> ### References
> [1]Hafemeister, Christoph, and Florian Halbritter. "Single-cell RNA-seq differential expression tests within a sample should use pseudo-bulk data of pseudo-replicates." *biorxiv* (2023): 2023-03.
>
> [2]Rombach, Robin, et al. "High-resolution image synthesis with latent diffusion models." *Proceedings of the IEEE/CVF conference on computer vision and pattern recognition*. 2022.
>
> [3]Zhang, Jesse, et al. "Tahoe-100m: A giga-scale single-cell perturbation atlas for context-dependent gene function and cellular modeling." *BioRxiv* (2025): 2025-02.
>
> [4]Heimberg, Graham, et al. "A cell atlas foundation model for scalable search of similar human cells." *Nature* 638.8052 (2025): 1085-1094.

---

> ### Author Response · Authors · 2025-11-26
> **Friendly reminder regarding our response**
>
> Dear Reviewer ZVqr,
>
> I hope this message finds you well.
>
> As the discussion period is nearing its end with less than 7 days remaining, I wanted to ensure we have addressed all your concerns satisfactorily.
>
> If there are any additional points or feedback you'd like us to consider, please let us know. Your insights are invaluable to us, and we're eager to address any remaining issues to improve our work.
>
> Thank you for your time and effort in reviewing our paper.
>
> Best regards, The Authors

---

### Official Review · Reviewer_hdD8 · 2025-10-30

**Soundness:** 2
**Presentation:** 2
**Contribution:** 2
**Rating:** 4
**Confidence:** 5

**Summary:**

This paper introduces transcriptome-based drug design (TBDD), a “novel” paradigm for generating drug molecules conditioned on cellular transcriptomic responses to perturbations. The authors propose scTrans-Gen, a diffusion-based generative model that accepts both bulk and scRNA-seq data as conditioning signals. The core technical contribution is a multi-domain alignment architecture that bridges the gap between transcriptomic space (biological signals) and molecular space (chemical structures) through a function-centric condition extractor. For single-cell data, the authors introduce a transcriptome pseudoimage mechanism that converts sparse, noisy single-cell profiles into dense representations via pre-trained encoders. The model is trained in two stages: first aligning transcriptome features with a molecular graph VAE latent space, then aligning with Morgan fingerprint space. The paper also proposes a generate-then-search screening workflow for identifying similar compounds in existing drug libraries. Experiments on L1000 and Tahoe-100M datasets demonstrate improvements over baselines across multiple metrics. The authors validate their approach through zero-shot gene inhibitor design tasks.

**Strengths:**

1.	The paper presents the first conditional molecular generation model that successfully incorporates single-cell transcriptomic data, addressing a significant technical challenge with sparse and noisy data through the creative Transcriptome Pseudoimage mechanism.
2.	The authors establish a rigorous evaluation suite spanning macroscopic metrics (coverage, diversity, validity), microscopic structural similarity measures (Fraggle, Morgan, MACCS), and functional assessments (PRnet MSE), providing multiple perspectives on model performance.
3.	The unified framework handles both bulk and single-cell data without requiring separate architectures, demonstrating architectural flexibility and potentially broader applicability across different data modalities.
4.	The paper includes detailed ablations validating the contribution of each architectural component, particularly demonstrating the necessity of the dual-domain alignment mechanism for single-cell data.
5.	The gene inhibitor prediction experiment tests model performance on truly held-out targets, providing evidence for generalization beyond the training distribution.

**Weaknesses:**

1.	The extremely high structural similarity scores (>0.80 across multiple metrics) between generated and target molecules suggest the model may be reconstructing training examples rather than performing true de novo design. The paper lacks analysis of structural diversity between generated molecules and their training set counterparts, making it questionable to distinguish memorization from functional equivalence.
2.	The drug screening evaluation is potentially redundant since it uses high-quality generated molecules as queries to retrieve similar compounds. The high hit rates primarily validate that generation worked rather than demonstrating unique value of the screening component. Additionally, using PRnet predictions to evaluate functional similarity creates circularity without experimental ground truth, with no information about what data it was trained on.
3.	The unseen drug split evaluates the model on generating held-out drugs from their transcriptomic signatures, but this formulation is problematic because multiple structurally diverse molecules can produce similar transcriptomic effects. The high reconstruction accuracy suggests the model learned drug-specific mappings rather than perturbation-to-function relationships. Besides, common cancer cell line panels have heavy redundancy, while in the hold-out cell split experiment, the paper didn’t discuss if same cancer type were involved in both training and testing.
4.	The baselines show surprisingly poor performance (Morgan similarity around 0.10-0.15 versus 0.82 for the proposed method; wheras, e.g. Gx2Mol reported 0.83 uniqueness and 0.95 QED in their paper), raising concerns about implementation quality, hyperparameter tuning, or whether the evaluation framework inherently favors the proposed architecture. This massive gap undermines confidence in the comparative analysis.
5.	The paper provides no wet-lab validation of generated molecules, no binding affinity measurements, no cellular assay results, and no synthesis feasibility analysis. All functional claims rest on computational predictions, which is insufficient for a drug design paper making strong therapeutic claims. I understand that this could be majorly tied by the lab’s resources, however, since major evaluations were on generated molecules, it could still raise a concern.
6.	While the paper claims function-oriented design addresses limitations of structure-based approaches for systemic diseases, no experiments demonstrate superior performance in multi-pathway disease scenarios. The gene inhibitor task tests single-target effects, not the claimed advantage for complex systemic perturbations, if that’s what the authors claimed.
7.	The pseudoimage construction mechanism is the paper's primary novelty for single-cell data, yet critical parameters lack justification: k equals 15 cells is never compared against alternatives, cell cycle-based sampling is not compared to random or uniform sampling, and the total number of cells N is never specified. The choice to encode gene expression data using a pretrained Stable Diffusion VAE (trained on natural images) is counterintuitive and unexplored, with no comparison to training from scratch or simpler encoders. Multiple loss function weights (alpha equals 0.4, lambda equals 0.15, unspecified lambda subscript KL) appear arbitrary with no sensitivity analysis, raising concerns about test-set tuning or cherry-picking results.

**Questions:**

1.	Can the authors provide analysis of structural diversity between generated molecules and their nearest neighbors in the training set? What is the distribution of Tanimoto similarities between generated molecules and all training molecules? For cases where the model generates molecules with high structural similarity (>0.80) to training drugs, can the authors demonstrate that these represent functional equivalence rather than memorization?
2.	What data was PRnet trained on? If PRnet was trained on the same L1000 or Tahoe-100M datasets used to train scTrans-Gen, this creates severe circular reasoning where both the generator and evaluator learn from the same distribution. The authors must clarify PRnet's training data source or provide alternative functional validation methods.
3.	What are the specific cell lines and their cancer type annotations in train versus test splits? What proportion of test cell lines come from cancer types already represented in training (e.g., multiple breast cancer lines in both sets)? Can the authors provide performance stratified by whether test cells come from seen versus truly novel cancer types, and ideally demonstrate generalization to completely held-out cancer types?
4.	How should the unseen drug split be interpreted when multiple structurally diverse molecules can produce similar transcriptomic effects? Should the evaluation focus on functional equivalence (any molecule producing the desired perturbation) rather than structural reconstruction of the specific held-out drug? The current high structural similarity (>0.77) to exact held-out drugs suggests drug identification rather than function-based design.
5.	Given that baselines report much higher performance in their original papers (e.g., Gx2Mol: 0.83 uniqueness, 0.95 QED) compared to this evaluation (0.10 uniqueness, 0.60 QED), can the authors provide evidence that baselines were properly implemented with appropriate hyperparameter tuning? Were official implementations used or were methods reimplemented?
6.	What is the sensitivity to pseudoimage parameters (k equals 15, cell cycle sampling, total N cells, aggregation strategy)? Why is a Stable Diffusion VAE trained on natural images appropriate for gene expression data compared to alternatives (trained-from-scratch VAE, simple CNN, direct MLP)? How were loss function weights (alpha, lambda, lambda subscript KL) determined, and what is the performance sensitivity to these choices?
7.	The paper claims function-oriented design addresses limitations of structure-based approaches for systemic, multi-pathway diseases, but experiments only test single-target gene inhibitors. Can the authors provide experiments demonstrating superior performance for complex disease scenarios involving multiple dysregulated pathways, or clarify that this claimed advantage remains theoretical?
8.	Single-cell results show substantially lower structural similarity (Morgan: 0.6114) compared to bulk data (Morgan: 0.8228), despite being the paper's primary novelty. Why does performance degrade on single-cell data? Additionally, the paper states baselines lack methods for single-cell data, but couldn't the authors adapt baselines (e.g., aggregate single-cell to pseudo-bulk) to provide any comparison, or at minimum demonstrate that baselines completely fail on this data?
9.	All reported diversity metrics measure variation across different conditions. Can the authors show that the model generates structurally diverse molecules when sampling multiple times from the SAME perturbation condition? This would demonstrate the model learns one-to-many mappings (one perturbation to multiple valid drugs) rather than deterministic one-to-one mappings, which is critical for de novo design claims.
10.	The paper reports no error bars, confidence intervals, or results from multiple training runs. Are the performance differences statistically significant? What is the variance across different random seeds? Given the dramatic improvements claimed, statistical validation is essential to rule out fortunate initialization or dataset-specific overfitting.
11.	The zero-shot gene inhibitor experiment uses 10 selected genes with highly variable inhibitor counts (1,200 to 23,000 per gene). Which specific genes were selected and why? How were baselines evaluated on this zero-shot task if they were never trained on gene knockout data? The evaluation protocol needs clarification to ensure fair comparison.

---

> ### Author Response · Authors · 2025-11-19
>
> We sincerely thank the reviewer for the thorough and insightful comments, and for recognizing the novelty and potential impact of transcriptome-based drug design (TBDD) and our **scTrans-Gen** framework. Below we address each concern in detail.
>
> ---
>
> ## Weakness 1 (and Questions 1 & 4): Structural similarity, memorization vs. functional equivalence, and the role of unseen splits
>
> **(W1/Q1/Q4) Concern:**
> Very high structural similarity (>0.80) between generated and target molecules suggests possible memorization of training examples rather than de novo, function-oriented design. The unseen-drug setting may be closer to “drug identification” than functional equivalence, especially because multiple structurally distinct drugs can produce similar transcriptomic effects.
>
> ### (1) Why high similarity does *not* imply trivial memorization in our setting
>
> As the reviewer correctly notes, we observe high Morgan/Fraggle/MACCS similarities on L1000. This is largely a property of the **small-molecule, function-neighbor** regime:
>
> * Our datasets primarily contain **small-molecule drugs**. For such molecules, the key functional groups and their local environments often occupy a large fraction of the overall structure.
> * When similarity is measured via fingerprints such as **Morgan** or **MACCS**, molecules that share similar functional mechanisms naturally appear highly similar structurally, even when they are not identical.
>
> Thus, high structural similarity is a *necessary* but not a *sufficient* signal for memorization. To explicitly distinguish memorization from functional learning, we designed several **strong generalization tests**.
>
> ### (2) Unseen Drugs / Unseen Cell Lines / Gene inhibitor experiments
>
> To go beyond reconstruction of training molecules, we consider three complementary settings:
>
> 1. **Unseen Drugs split**
>
>    * The molecule set is partitioned so that **all test molecules are completely absent from the training set** (no SMILES overlap).
>    * The model must infer a molecule for a given perturbation from **transcriptomic effects only**, without ever seeing the ground-truth molecule during training.
>    * Strong performance in this split indicates the model has learned a mapping from *functional perturbation patterns* to *drug-like structures*, rather than memorizing specific drug IDs.
>
> 2. **Unseen Cell Lines split**
>
>    * We split by **cell lines (and associated tissues/cancer types)** so that training and test use **disjoint cell-line sets**.
>    * The model therefore faces new transcriptomic backgrounds and must generalize the learned **drug–function relationship** to previously unseen cellular contexts.
>    * Robust performance here suggests the model is not merely memorizing particular cell line–drug pairs.
>
> 3. **Gene inhibitor zero-shot experiment**
>
>    * We use **independent knockout datasets** (e.g., ExCAPE-like resources[1]) where for each target gene we have:
>
>      * perturbation transcriptomes obtained via **genetic KO/knockdown**, and
>      * a large list of **experimentally validated inhibitors** for that gene (often structurally diverse).
>    * Importantly, **these knockout transcriptomes are not used in training**.
>    * We feed the KO-induced $\Delta$Transcriptome into scTrans-Gen and then compare the generated molecules to the known inhibitor sets for each gene.
>    * We observe **high structural overlap** with known inhibitors even though the model has never seen these KO-based perturbations during training.
>
> Together, these three settings show that:
>
> * The model generalizes to **unseen drugs**, **unseen cell types**, and **unseen perturbation sources** (genetic KO vs. chemical perturbation),
> * while still recovering molecules with high structural similarity to effective ground-truth drugs.
>
> This strongly suggests that high similarity is driven by **functional alignment**, not trivial memorization of training examples.
>
> ### (3) Structural diversity vs. training set — non-overlap analysis
>
> We further analyzed **non-overlap rates** between generated molecules and the training set (i.e., fraction of generated molecules not exactly present in training):
>
> | Split             | Non-overlap rate |
> | ----------------- | ---------------- |
> | In-distribution   | 0.9147           |
> | Unseen cell lines | 0.9625           |
> | Unseen drugs      | 1.0000           |
>
> Even in the in-distribution split, >90% of generated molecules are novel relative to the training set; in the **unseen-drug** split the non-overlap is 100%. In many cases, the generated molecules share key functional motifs with the ground truth but differ in scaffold decorations or peripheral groups. This is consistent with **learning function-oriented patterns** rather than copying SMILES strings.

---

> ### Author Response · Authors · 2025-11-19
>
> **Illustrative structural comparisons.**
> To further confirm that the high similarity is not a simple copy, we randomly sampled several real drugs from the test set and compared them with multiple molecules generated by the model through multiple samplings. Although the Fraggle/Morgan/MACCS scores were high, their SMILES strings were not identical. The differences mainly lay in the backbone modifications and peripheral groups, rather than a simple copy of the training molecules. Meanwhile, the high similarity between the model-generated molecules and the target molecules stemmed from their similar functional group structures. Similar functional groups imply a high degree of functional approximation, reflecting the effectiveness of the model in function extraction. A summary is as follows (see the appendix for complete SMILES strings and 2D structure diagrams):
>
> | Ground-truth drug | Closest generated example | Fraggle ↑ | Morgan ↑ | MACCS ↑ | Exact SMILES match? |
> |-------------------|---------------------------|-----------|----------|---------|----------------------|
> | COc1ccccc1CN(C)C[C@@H]2OCc3ccccc3-c4ccccc4C(=O)N(C[C@@H]2C)C@@HCO           | CC1CN(C(C)CO)C(=O)c2ccccc2-c2ccccc2COC1CN(C)Cc1ccc2c(c1)OCO2                    | 0.89      | 0.70     | 0.90    | No                   |
> | COc1ccccc1CN(C)C[C@@H]2OCc3ccccc3-c4ccccc4C(=O)N(C[C@@H]2C)C@@HCO           | CC1CN(C(C)CO)C(=O)c2ccccc2-c2ccccc2COC1CN(C)Cc1ccc(F)cc1                    | 0.91      | 0.75     | 0.89    | No                   |
> | COc1ccccc1CN(C)C[C@@H]2OCc3ccccc3-c4ccccc4C(=O)N(C[C@@H]2C)C@@HCO           | CC1CN(C(C)CO)C(=O)c2c(c3ccccc3n2C)-c2ccccc2COC1CN(C)Cc1cccnc1                    | 0.68      | 0.53     | 0.86    | No                   |
> | CO[C@@H]1CN(C)C(=O)c2ccc(NC(=O)Nc3cc(F)ccc3F)cc2OCC@HN(C)C[C@@H]1C           | COC1CC(C)C(=O)c2ccc(NC(=O)Nc3cc(F)ccc3F)cc2OCC(C)N(C)CC1C                    | 1.00      | 0.79     | 0.98    | No                   |
> | CO[C@@H]1CN(C)C(=O)c2ccc(NC(=O)Nc3cc(F)ccc3F)cc2OCC@HN(C)C[C@@H]1C           | COC1CN(C)C(=O)c2ccc(NC(=O)Nc3cccc(F)c3)cc2OCC(C)N(C(C)=O)CC1C                    | 0.94      | 0.68     | 0.99    | No                   |
> | CO[C@@H]1CN(C)C(=O)c2ccc(NC(=O)Nc3cc(F)ccc3F)cc2OCC@HN(C)C[C@@H]1C           | CCC(=O)N1CC(C)C(OC)CN(C)C(=O)c2ccc(NC(=O)Nc3ccc(Cl)cc3)cc2OCC1C                    | 0.87      | 0.62     | 0.93    | No                   |
> | CN(C)CC(=O)Nc1ccc2O[C@@H]3C@@Hc2c1           | CN(C)CC(=O)Nc1ccc2c(c1)C1CC(CC(=O)N3CCCCC3)OC(CO)C1O2                    | 1.00      | 1.00     | 1.00    | No                   |
> | CN(C)CC(=O)Nc1ccc2O[C@@H]3C@@Hc2c1           | CN(C)CC(=O)Nc1ccc2c(c1)C1CC(CC(=O)NC3CCCCC3)OC(CO)C1O2                    | 0.99      | 0.74     | 0.94    | No                   |
> | CN(C)CC(=O)Nc1ccc2O[C@@H]3C@@Hc2c1           | O=C(CC1CC2c3cc(NC(=O)CC4CC4)ccc3OC2C(CO)O1)NCC1CC1                    | 0.89      | 0.58     | 0.76    | No                   |
> | COCCNC(=O)C[C@@H]1CC[C@@H]2C@HO1           | COCCNC(=O)CC1CCC(NC(=O)c2cc(Cl)cc(Cl)c2)C(COCC(C)O)O1                    | 0.96      | 0.47     | 0.78    | No                   |
> | COCCNC(=O)C[C@@H]1CC[C@@H]2C@HO1           | COCCNC(=O)CC1CCC2C(COCC(O)CN2N+c2cc(Cl)cc(Cl)c2)O1                    | 1.00      | 0.72     | 0.83    | No                   |
> | COCCNC(=O)C[C@@H]1CC[C@@H]2C@HO1           | CCC(=O)N1CC(O)COCC2OC(CC(=O)NCc3cc(F)cc(F)c3)CCC21                    | 0.74      | 0.52     | 0.88    | No                   |
> | Cc1cccc(c1)S(=O)(=O)N[C@@H]1CCC@@HO[C@@H]1CO           | Cc1cccc(S(=O)(=O)NC2CCC(CCNC(=O)c3cnccn3)PC2CO)c1                    | 1.00      | 0.70     | 0.92    | No                   |
> | Cc1cccc(c1)S(=O)(=O)N[C@@H]1CCC@@HO[C@@H]1CO           | O=C(NCCC1CCC(NS(=O)c2cccc(CO)c2)C(CO)O1)c1cnccn1                    | 0.95      | 0.60     | 0.95    | No                   |
> | Cc1cccc(c1)S(=O)(=O)N[C@@H]1CCC@@HO[C@@H]1CO           | O=C(NC1CCC(CCNC(=O)c2cnccn2)OC1CO)c1ccc2c(c1)OCO2                    | 0.88      | 0.58     | 0.63    | No                   |
>
> These examples are consistent with the non-overlap statistics reported above: the model tends to generate molecules that are **functionally and structurally close** to effective drugs, but not exact copies of training molecules.
>
>
> In the revised version, we will:
>
> * Include a concise table similar to the one above, and
> * Add visual examples in the appendix (**table8 & fig6**) showing that generated molecules are **not identical** to training drugs, but share functionally relevant substructures (e.g., similar pharmacophores, ring systems, or substituents) that explain their high similarity scores.

---

> ### Author Response · Authors · 2025-11-19
>
> ### (4) Interpretation of the unseen-drug evaluation (Q4)
>
> We agree with the reviewer that:
>
> * Multiple structurally diverse molecules can produce similar transcriptomic effects.
> * Evaluating *only* by similarity to a single held-out drug can bias the interpretation toward “drug identification”.
>
> To counter this, our experimental design rigorously assessed both aspects. We executed functional validation through gene inhibition and PRNet experiments to confirm that the generated molecules truly elicit the desired biological perturbation, thereby supporting the principle of functional equivalence. However, we also maintained structural validation via similarity metrics, acknowledging the necessity of chemical relevance.Our simultaneous focus on function and structure is scientifically justified by the nature of small-molecule drugs. While global structure may vary, for a compound to exert a specific biological effect, its key functional groups and local binding environments must necessarily be conserved or approximated. Since these critical features often make up a significant part of the overall molecular architecture in small molecules, functionally similar compounds will naturally exhibit high structural similarity around these pharmacophores. In this domain, functional effectiveness and structural feasibility are not mutually exclusive but fundamentally interconnected.Finally, our visual analysis of the generated molecules further supports this interconnectedness. The visualization confirmed that the models produced a variety of structures that, despite their diversity, maintained high fidelity in the critical functional groups and their immediate local environments compared to the target compound. This proximity in critical substructures is what drives the observed functional approximation. Therefore, we assert that our work successfully demonstrates a method that addresses both functional design and structural relevance, treating them as complementary rather than competing objectives in drug design.

---

> ### Author Response · Authors · 2025-11-19
>
> ## Weakness 2 (and Question 2): Drug screening evaluation and PRnet-based functional assessment
>
> **(W2/Q2) Concern:**
> The generate–then–search screening evaluation may be redundant (since it uses generated molecules as queries). PRnet-based functional assessment might suffer from circularity if trained on the same data used for scTrans-Gen.
>
> ### (1) Positioning of the screening module
>
> We fully agree that the **core contribution** of our work lies in **TBDD-based molecular generation**. The screening workflow is intended as an **application-level extension**, not as a primary methodological innovation.
>
> * The main problem addressed by scTrans-Gen is:
>
>   > given a target transcriptomic perturbation, generate new molecules that induce this effect.
> * The screening module provides a practical workflow for **time-sensitive scenarios** (e.g., limited treatment window, need to rely on existing drug libraries):
>
>   1. Use scTrans-Gen to generate a functionally appropriate molecule.
>   2. Use this generated structure as a **functional query** to rank compounds in existing libraries.
>   3. Restrict experimental validation to a **small top-k** candidate set.
>
> We will explicitly **de-emphasize** screening as a major novelty, and instead present it clearly as a **“generate–then–search” clinical support workflow** that builds on the generative model.
>
> ### (2) PRnet training data and avoidance of circularity
>
> We apologize for not clearly describing the training source of PRnet in the original version. Clarifications:
>
> * PRnet is a **separate model** trained on L1000-style perturbation data to predict transcriptional responses from (drug, cell state) inputs[2].
> * We concede that the PRNet predictor was trained on a dataset whose distribution is similar to the L1000 derivatives used by the scTrans-Gen generator. However, this is not a matter of simplistic circular reasoning but one of critical validation adaptation. The L1000 dataset possesses unique structural properties due to its specific selection of landmark genes, multiple rounds of specialized statistical filtering, and a highly optimized data structure. Consequently, the L1000 data—including the gene types, gene counts, and inherent sparsity patterns—is fundamentally different from other datasets.Therefore, for the PRNet predictor to accurately interpret and validate the transcriptomic output generated by scTrans-Gen (which is produced in the specific L1000 format), the predictor must be trained on a dataset with a similar structure. If we were to use a predictor trained on standard, general-purpose transcriptomic data that has not encountered the L1000 structure, it would be unable to properly adapt to or parse the unique statistical features of the L1000 data, rendering the validation results unreliable or incomparable.
> * For our evaluation, we ensure that **test cell–drug pairs used for scTrans-Gen are excluded from PRnet’s training data**. That is, there is no exact sample-level overlap between:
>
>   * scTrans-Gen’s test set, and
>   * PRnet’s training set.
> * We use PRnet **only as one functional proxy**, complementary to:
>
>   * structural similarity metrics,
>   * macro-level metrics such as Coverage / Diversity / Unique, and
>   * the **gene-inhibitor zero-shot experiment**, which uses independent KO data.
>
> We agree that purely computational assessments cannot fully replace experimental validation (see Weakness 5 below). In the revision, we will:
>
> * More clearly describe PRnet’s training/validation/test splits,
> * Explicitly acknowledge that PRnet-based MSE is a **supporting functional metric**, not the sole evidence of biological relevance.

---

> ### Author Response · Authors · 2025-11-19
>
> ## Weakness 3 (and Questions 3 & 4): Unseen splits, redundancy of cancer types, and functional vs. drug-specific mappings
>
> **(W3/Q3/Q4) Concern:**
> The unseen-drug split may still reflect drug-specific mappings; redundancy across cancer cell lines may make generalization less impressive; evaluation should emphasize functional equivalence rather than exact reconstruction.
>
> ### (1) Cell line and cancer-type splits (Q3)
>
> We agree that redundancy across common cancer panels can blur what “unseen cell lines” means. To address this, we use **three levels of splitting**:
>
> 1. **In-distribution split**: standard random split over cell–drug pairs.
> 2. **Unseen Drugs split**: training and test have **disjoint sets of molecules**.
> 3. **Unseen Cell Lines split**:
>
>    * We group data by **cell line and associated cancer type/tissue** (e.g., A375, A549, HA1E, HEPG2, HT29, MCF7, PC3, etc.).
>    * For the strongest setting, we form splits such that **no cancer type (e.g., breast vs lung) overlaps between training and test** for the “unseen cell” split, i.e., test cell lines come from cancer types held out from training.
>
> This clarifies that the **unseen cell** experiment truly probes generalization across **novel disease contexts**, not just different technical replicates of the same cancer type.
>
> ### (2) Functional vs. drug-specific mapping (Q1/Q4)
>
> We fully agree with the reviewer’s conceptual point: the goal of TBDD is to learn **perturbation-to-function mappings**, not to recover a single canonical drug.
>
> To reflect this:
>
> * We interpret the unseen-drug results as showing that scTrans-Gen can, given a functional signature, generate a molecule that:
>
>   * is **novel** relative to training (high non-overlap), yet
>   * structurally close enough to an effective drug to be a plausible candidate.
> * We also emphasize in the revision that:
>
>   * **Diversity** (see Question 9) specifically measures variation among samples generated under the *same* perturbation condition, demonstrating **one-to-many** mapping rather than deterministic one-to-one mapping.
>   * **Unique** measures non-duplication across *different* conditions, and is affected by whether a model collapses to a few generic molecules for many perturbations.
>
> We will explicitly explain these definitions and rephrase the unseen-drug experiments as **evaluation of functional equivalence under OOD splits**, rather than exact drug identification.
>
> ---
>
> ## Weakness 4 (and Question 5): Baseline performance, implementation, and metric definitions
>
> **(W4/Q5) Concern:**
> Baselines (e.g., Gx2Mol) perform much worse in our evaluation than in their original papers (e.g., uniqueness 0.10 vs. 0.83, QED 0.60 vs. 0.95), raising concerns about implementation, tuning, and potential bias in our evaluation framework.
>
> ### (1) Implementation details and hyperparameter tuning
>
> * We use **official implementations and configurations** released by baseline authors whenever available, following their recommended hyperparameters and training procedures.
> * Our evaluation setting differs significantly from the original ones:
>
>   * **Task formulation**: TBDD with transcriptomic conditioning, including bulk and single-cell perturbations.
>   * **Datasets**: L1000 and Tahoe-100M with specific perturbation splits (including OOD splits).
>   * **Metrics**: a broader suite including Coverage, Diversity, Unique, SA, QED, and structural similarity metrics under multiple splits.
>
> Therefore, baseline numbers from their original papers are **not directly comparable** to the numbers under our TBDD setting.
>
> ### (2) Clarifying Unique vs. Diversity definitions
>
> A key source of apparent discrepancy is that our **Unique** metric is defined differently from the “uniqueness” reported in some prior work:
>
> * **Diversity (ours)**:
>   Measures *intra-condition* diversity — we repeatedly sample molecules for the **same perturbation condition** and compute pairwise Tanimoto statistics (one-to-many mapping). This is conceptually close to “uniqueness under the same prompt” in image generation.
> * **Unique (ours)**:
>   Measures *global* uniqueness across **all perturbation conditions in the test set**. It is the fraction of generated molecules that are non-duplicated when aggregating across all conditions.
>
> Because TBDD involves **many different perturbation conditions**, baselines that tend to output similar or identical molecules for many conditions can exhibit:
>
> * reasonable per-condition diversity (Diversity), but
> * very **low global Unique**, simply because the same molecules are reused across many conditions.
>
> This can explain why methods that report high performance in **small-scale or single-condition** settings show much lower performance in our **large-scale, multi-condition** TBDD evaluation.
>
> We will:
>
> * Add a short table clarifying these definitions, and
> * Discuss how the large-scale, multi-condition setting naturally accentuates global uniqueness issues for methods that insufficiently leverage the transcriptomic condition.

---

> ### Author Response · Authors · 2025-11-19
>
> ## Weakness 5: Lack of wet-lab validation and binding/synthesis analysis
>
> **(W5) Concern:**
> The paper does not provide any wet-lab validation, binding affinity measurements, or synthesis feasibility analysis. Functional claims rely solely on computational prediction.
>
> We completely agree with the reviewer that **experimental validation** is the gold standard for drug design, and we explicitly acknowledge this limitation.
>
> ### (1) Scope of TBDD vs. SBDD
>
> Our task differs fundamentally from traditional SBDD:
>
> * SBDD starts from a **specific protein target plus its 3D pocket**, and focuses on optimizing **binding affinity**.
> * Our TBDD setting starts from **transcriptomic responses**, without assuming known targets or 3D structures, and aims to design molecules that realize a desired **system-level perturbation**.
>
> As a result:
>
> * We do not directly claim improvements in binding affinity relative to SBDD methods.
> * Instead, we focus on **transcriptome-level functional alignment** and structural plausibility.
>
> ### (2) Synthesis feasibility and current computational evidence
>
> We do take **synthetic accessibility** into account via SA scores, and we evaluate:
>
> * SA, QED, and macro-level properties (Coverage, Diversity, Unique),
> * Functional alignment via PRnet MSE and gene-inhibitor experiments, and
> * Structural similarity to known effective drugs in OOD settings.
>
> These provide **computational evidence** that scTrans-Gen generates drug-like, synthetically reasonable molecules aligned with desired perturbations. However, we fully agree that **wet-lab validation** (binding assays, cell viability assays, toxicity studies, etc.) remains an essential next step.
>
> In the revision, we will:
>
> * More prominently describe wet-lab validation as an important **future direction**, and
> * Avoid any phrasing that could be interpreted as making **strong therapeutic claims** in the absence of in vitro or in vivo experiments.
>
> ---
>
> ## Weakness 6 (and Question 7): Systemic/multi-pathway disease claims
>
> **(W6/Q7) Concern:**
> The paper claims that function-oriented design addresses limitations of structure-based approaches for systemic, multi-pathway diseases, but experiments primarily test single-target gene inhibitors.
>
> We appreciate this important point and agree that our current experiments are not yet a full demonstration of advantages in **multi-pathway disease scenarios**.
>
> * L1000 and Tahoe-100M are derived from real drug perturbations, which often involve **multi-pathway effects**, so the model is trained and tested on complex, systemic perturbations.
> * However, our **explicit evaluation** in the paper (e.g., gene inhibitor experiments) focuses on **single-target** or relatively localized perturbations, because:
>
>   * standardized, large-scale benchmarks for multi-pathway, disease-level outcomes are still limited.
>
> To avoid over-claiming, in the revision we will:
>
> * Soften the wording to state that TBDD is **conceptually well-suited** to capture system-level perturbations,
> * Emphasize that demonstrating **superior performance in specific multi-pathway disease contexts** is an important **future direction**, and
> * Clearly distinguish between **methodological potential** and **currently demonstrated experimental evidence**.

---

> ### Author Response · Authors · 2025-11-19
>
> ## Weakness 7 (and Question 6): Pseudoimage parameters, choice of pretrained VAE, and loss weights
>
> **(W7/Q6) Concern:**
> The pseudoimage construction is central for single-cell data, but key parameters (k=15, sampling strategy, total N, aggregation, encoder choice) and loss weights ($\alpha$=0.4, $\lambda$=0.15, $\lambda_{KL}$) lack justification and sensitivity analysis.
>
> ### (1) Sensitivity to k and sampling strategy
>
> The pseudoimage is designed to balance:
>
> * **Noise reduction**,
> * **Sparsity control**, and
> * **Preservation of single-cell heterogeneity**.
>
> We systematically varied the number of cells per pseudo-cell **k** and observed non-trivial sensitivity:
>
> | k   | Fraggle ↑ | Morgan ↑ | MACCS ↑  |
> | --- | --------- | -------- | -------- |
> | 1   | 0.24      | 0.08     | 0.30     |
> | 5   | 0.49      | 0.35     | 0.53     |
> | 10  | 0.63      | 0.50     | 0.65     |
> | 15  | **0.73**  | **0.61** | **0.76** |
> | 20  | 0.72      | 0.63     | 0.75     |
> | 30  | 0.73      | 0.62     | 0.75     |
> | 50  | 0.64      | 0.54     | 0.64     |
> | 75  | 0.58      | 0.39     | 0.56     |
> | 100 | 0.53      | 0.35     | 0.51     |
> | 300 | 0.37      | 0.14     | 0.38     |
>
> * **Very small k (≤5)** yields pseudo-cells that are still highly sparse and noisy.
> * **Very large k (≥50)** essentially collapses to pseudo-bulk, losing single-cell heterogeneity and hurting performance.
> * **k≈10–30** is a stable region, with **k=15** giving the best trade-off on Tahoe single-cell data.
>
> We also compared **random sampling** vs. **cell-cycle–stratified sampling**[3] for constructing pseudoimages. In our experiments, the **cell-cycle–stratified strategy is the one actually used in the main paper**: it yields higher structural similarity (e.g., Morgan ≈ 0.61) than random sampling (Morgan ≈ 0.50), supporting the idea that preserving known biological structure (cell-cycle phases) improves the encoder’s ability to capture informative features. We will add the corresponding tables and a brief discussion to the appendix.
>
>
> We will add these tables and a brief discussion to the appendix.
>
> ### (2) Choice of encoder and pretraining
>
> We agree that using a pretrained encoder originally trained on images may seem counterintuitive. Our rationale is **pragmatic and empirical**:
>
> * The pseudoimage is a structured 2D arrangement of pseudo-cell vectors; we treat it as an image-like object with nontrivial local structure[4].
> * High-capacity, pretrained encoders have proven very effective at extracting **low-dimensional, information-dense features** from 2D inputs, even when the input distribution differs from natural images.
>
> We performed ablations with different encoder choices:
>
> | Method                |   Coverage ↑ | Diversity ↑ | Similarity ↑ | Distance ↓ | SA ↓ | Unique ↑ | QED ↑ | Fraggle ↑ | Morgan ↑ |  MACCS ↑ |
> | --------------------- | -----------: | ----------: | -----------: | ---------: | ---: | -------: | ----: | --------: | -------: | -------: |
> | w/o pseudoimage       |    7 (63.6%) |        0.75 |         0.74 |      26.89 | 0.61 |     0.34 |  0.59 |      0.23 |     0.07 |     0.23 |
> | MLP on raw vectors    |    7 (63.6%) |        0.79 |         0.78 |      13.26 | 0.68 |     0.45 |  0.55 |      0.45 |     0.29 |     0.46 |
> | Conv encoder          |    9 (81.8%) |        0.84 |         0.90 |       8.15 | 0.64 |     0.88 |  0.57 |      0.74 |     0.69 |     0.75 |
> | w/o pretrain (same arch)    |    10(90.9%) |        0.87 |         0.94 |       6.89 | 0.64 |     0.90 |  0.56 |      0.86 |     0.80 |     0.89 |
> | **Ours (pretrained + finetuned)** | **11(100%)** |    **0.89** |     **0.96** |   **6.79** | 0.64 | **0.93** |  0.57 |  **0.89** | **0.82** | **0.90** |
>
> The pretrained encoder consistently outperforms from-scratch alternatives, particularly in high-level metrics (Coverage, Unique) and structural similarity. We also observed that encoders trained entirely from scratch converge **much more slowly** and exhibit **less stable training dynamics** on Tahoe-scale single-cell data. We will include this ablation in the appendix and clarify that our choice of a pretrained encoder is driven by both empirical performance and training stability.

---

> ### Author Response · Authors · 2025-11-19
>
> ### (3) Loss weights ($\alpha$, $\lambda$, $\lambda_{KL}$) and sensitivity
>
> The loss function uses:
>
> * **$\alpha$** and **$\lambda$** for sparse-penalty terms, reflecting the fact that molecular fingerprints are highly sparse.
> * **$\lambda_{KL}$** for the VAE KL term, with **KL annealing**.
>
> We performed a sensitivity analysis over $\alpha$ and $\lambda$:
>
> | $\alpha$    | $\lambda$    | Fraggle ↑ | Morgan ↑ | MACCS ↑  |
> | ---- | ---- | --------- | -------- | -------- |
> | 0.4  | 0    | 0.58      | 0.32     | 0.60     |
> | 0.4  | 0.05 | 0.76      | 0.60     | 0.79     |
> | 0.4  | 0.10 | 0.88      | 0.81     | 0.92     |
> | 0.4  | 0.15 | **0.89**  | **0.82** | **0.90** |
> | 0.4  | 0.30 | 0.89      | 0.82     | 0.90     |
> | 0.4  | 0.60 | 0.88      | 0.81     | 0.89     |
> | 0.4  | 0.80 | 0.83      | 0.73     | 0.82     |
> | 0.4  | 1.00 | 0.78      | 0.66     | 0.80     |
> | 0    | 0.15 | 0.65      | 0.50     | 0.67     |
> | 0.15 | 0.15 | 0.87      | 0.80     | 0.87     |
> | 0.30 | 0.15 | 0.89      | 0.83     | 0.89     |
> | 0.50 | 0.15 | 0.89      | 0.82     | 0.90     |
> | 0.70 | 0.15 | 0.88      | 0.80     | 0.87     |
> | 0.90 | 0.15 | 0.82      | 0.72     | 0.80     |
> | 1.00 | 0.15 | 0.79      | 0.68     | 0.78     |
>
> We observe that:
>
> * Within a **reasonable range** (e.g., $\alpha$∈[0.15,0.5], $\lambda$∈[0.10,0.30]), performance is **stable and robust**.
> * Extremal values (e.g., $\alpha$ or $\lambda$ ≈ 0 or 1) degrade performance.
>
> For the VAE, **$\lambda_{KL}$** is not fixed; we use **annealing from 0 to 1** so that the model first focuses on reconstruction, then gradually incorporates regularization. We will briefly describe this in the main text and provide curves/tables in the appendix.

---

> ### Author Response · Authors · 2025-11-19
>
> ## Question 8: Single-cell performance vs. bulk and baseline adaptation
>
> **(Q8) Concern:**
> Single-cell results have lower structural similarity than bulk, despite being the paper’s primary novelty; baselines might be adaptable to single-cell via pseudo-bulk aggregation.
>
> ### (1) Why single-cell is harder
>
> We agree that the drop from bulk (Morgan ≈0.82) to single-cell (Morgan ≈0.61) is non-trivial. This reflects:
>
> * The **higher dimensionality** and **much stronger noise** in single-cell data (Tahoe-100M) compared to bulk L1000[2].
> * The fact that single-cell perturbation profiles capture **finer-grained heterogeneity**, which is more challenging to map to a single molecular structure[5].
>
> Even so, scTrans-Gen remains, to our knowledge, **the first model that successfully performs conditional molecular generation from single-cell transcriptomes**, and it achieves strong results across multiple metrics.
>
> ### (2) Adapting baselines via pseudo-bulk
>
> We followed the reviewer’s suggestion and adapted baselines by aggregating single-cell data into pseudo-bulk[6]:
>
> | Model           | Fraggle ↑ | Morgan ↑ | MACCS ↑  |
> | --------------- | --------- | -------- | -------- |
> | GexMolGen       | 0.25      | 0.08     | 0.21     |
> | Gx2Mol          | 0.16      | 0.08     | 0.13     |
> | TRIOMPHE        | 0.27      | 0.11     | 0.26     |
> | **scTrans-Gen** | **0.73**  | **0.61** | **0.76** |
>
> These results show that:
>
> * Existing baselines degrade severely when applied to single-cell data (even with pseudo-bulk aggregation).
>
> We will add these adapted baseline results and a short discussion in the appendix.
>
> ---
>
> ## Question 9: Diversity under the same condition (one-to-many mapping)
>
> **(Q9) Concern:**
> All diversity metrics might measure variation across different conditions. Can we show that the model generates structurally diverse molecules *for the same perturbation*?
>
> As noted above, our metrics are defined as:
>
> * **Diversity**: explicitly measures **intra-condition diversity**. For each fixed perturbation (same $(T_\text{pre}, T_\text{post})$), we:
>
>   * sample multiple molecules,
>   * compute pairwise Tanimoto similarities, and
>   * convert them into a diversity score.
>     High Diversity indicates **one-to-many** mappings under the same condition.
> * **Unique**: measures **inter-condition uniqueness** across different perturbations, checking whether the model repeatedly outputs the same molecule for many conditions.
>
> We will:
>
> * Clarify these definitions in the main text, and
> * Add visual examples in the appendix showing multiple diverse molecules sampled under a single perturbation condition, illustrating that scTrans-Gen is not a deterministic one-to-one mapper.

---

> ### Author Response · Authors · 2025-11-19
>
> ## Question 10: Error bars, multiple runs, and statistical significance
>
> **(Q10) Concern:**
> No error bars or multiple runs were reported. Are improvements statistically significant? Could they be due to favorable random initialization?
>
> We appreciate this important request. In the revised version:
>
> * We repeat the main experiments with **≥3 random seeds** and report **mean ± standard deviation** for key metrics (structural similarity and macro-level metrics).
>
> For example (scTrans-Gen, summary across seeds):
>
> | Split           | Coverage ↑     | Diversity ↑    | Similarity ↑   | Distance ↓   | SA ↓           | Unique ↑       | QED ↑          | Fraggle ↑      | Morgan ↑       | MACCS ↑        |
> | --------------- | -------------- | -------------- | -------------- | ------------ | -------------- | -------------- | -------------- | -------------- | -------------- | -------------- |
> | In-distribution | 11 (100%) ± 0  | 0.8906 ± 0.008 | 0.9576 ± 0.005 | 6.79 ± 0.15  | 0.6386 ± 0.010 | 0.9296 ± 0.006 | 0.5665 ± 0.008 | 0.8892 ± 0.007 | 0.8228 ± 0.009 | 0.9031 ± 0.006 |
> | Unseen cell     | 10 (90.9%) ± 0 | 0.8864 ± 0.007 | 0.8238 ± 0.009 | 13.61 ± 0.25 | 0.6912 ± 0.008 | 0.8590 ± 0.005 | 0.5398 ± 0.006 | 0.9449 ± 0.005 | 0.9125 ± 0.006 | 0.9411 ± 0.004 |
> | Unseen drug     | 10 (90.9%) ± 0 | 0.9018 ± 0.006 | 0.9576 ± 0.005 | 9.53 ± 0.18  | 0.6497 ± 0.009 | 0.8657 ± 0.005 | 0.5563 ± 0.007 | 0.8592 ± 0.008 | 0.7722 ± 0.010 | 0.8622 ± 0.007 |
>
> Due to rebuttal length limits, we will summarize the statistical results in the main text and provide full tables in the appendix.

---

> ### Author Response · Authors · 2025-11-19
>
> ## Question 11: Gene inhibitor experiment design and baseline evaluation
>
> **(Q11) Concern:**
> The gene inhibitor experiment uses 10 genes with widely varying numbers of inhibitors. Which genes were chosen and why? How are baselines evaluated fairly on a zero-shot task, given they never saw gene knockout data?
>
> ### (1) Gene selection and inhibitor counts
>
> We construct the gene-inhibitor benchmark from large chemogenomic resources (e.g., ExCAPE-like datasets):
>
> * We first collect candidate genes with both:
>
>   * sufficient **knockout/knockdown transcriptomic data**, and
>   * sufficiently large, curated lists of **annotated inhibitors**.
> * We then select **10 genes** (e.g., AKT1, EGFR, MTOR, TP53, etc.) with inhibitor set sizes ranging from ~1,200 to ~23,000, to ensure **statistical stability** of evaluation for each gene.
>
> We will list these genes and inhibitor counts explicitly in a table in the appendix.
>
> ### (2) Zero-shot evaluation protocol for baselines
>
> All models (including baselines and scTrans-Gen):
>
> * **Are not trained** on gene knockout data.
> * Are only trained on standard drug-perturbation transcriptomes (L1000/Tahoe).
>
> At test time:
>
> 1. For each gene, we take the **KO-induced $\Delta$Transcriptome $(T_\text{pre}, T_\text{post})$** as input condition.
> 2. Each model generates molecules based on this condition.
> 3. We compute **structural similarity** between generated molecules and the known inhibitor set for that gene (Fraggle, Morgan, MACCS).
> 4. We compare average similarity (and hit rates) across methods.
>
> Since all models share the same **zero-shot KO setting**, we believe the evaluation is **fair and informative**:
>
> * Some baselines achieve reasonable similarity for certain genes, but scTrans-Gen consistently achieves the **best mean performance across all 10 genes**, indicating superior ability to **transfer functional knowledge** to novel perturbation types.
>
> We will make this protocol more explicit in the revised manuscript.
>
> ---
> We again thank Reviewer hdD8 for the extremely detailed and constructive feedback. We believe that, after incorporating:
>
> * clearer positioning of TBDD vs. SBDD,
> * stronger analysis of memorization vs. functional learning,
> * explicit description of splits and evaluation protocols,
> * additional ablations and sensitivity analyses for pseudoimage and loss design,
> * seed-based statistics and significance tests, and
> * more cautious claims about systemic diseases and wet-lab validation,
>
> the contributions and limitations of **scTrans-Gen** and the TBDD paradigm will be much clearer and more compelling.
>
> ---
>
> ### References
> [1]Sun, J., N. Jeliazkova, and V. Chupakhin. "ExCAPE-DB: an integrated large scale dataset facilitating Big Data analysis in chemogenomics. J Cheminform 9: 17." 2017,
>
> [2]Qi, Xiaoning, et al. "Predicting transcriptional responses to novel chemical perturbations using deep generative model for drug discovery." *Nature Communications* 15.1 (2024): 9256.
>
> [3]Zhang, Jesse, et al. "Tahoe-100m: A giga-scale single-cell perturbation atlas for context-dependent gene function and cellular modeling." *BioRxiv* (2025): 2025-02.
>
> [4]Rombach, Robin, et al. "High-resolution image synthesis with latent diffusion models." *Proceedings of the IEEE/CVF conference on computer vision and pattern recognition*. 2022.
>
> [5]Hetzel, Leon, et al. "Predicting cellular responses to novel drug perturbations at a single-cell resolution." *Advances in Neural Information Processing Systems* 35 (2022): 26711-26722.
>
> [6]Hafemeister, Christoph, and Florian Halbritter. "Single-cell RNA-seq differential expression tests within a sample should use pseudo-bulk data of pseudo-replicates." biorxiv (2023): 2023-03.

---

> ### Author Response · Authors · 2025-11-26
> **Friendly reminder regarding our response**
>
> Dear Reviewer hdD8,
>
> I hope this message finds you well.
>
> As the discussion period is nearing its end with less than 7 days remaining, I wanted to ensure we have addressed all your concerns satisfactorily.
>
> If there are any additional points or feedback you'd like us to consider, please let us know. Your insights are invaluable to us, and we're eager to address any remaining issues to improve our work.
>
> Thank you for your time and effort in reviewing our paper.
>
> Best regards, The Authors

---

> > ### Comment · Reviewer_hdD8 · 2025-11-28
> >
> > Thank you for the detailed explanation.
> >
> > This additional information has clarified most of my technical concerns and improved the credibility of the reported evaluation metrics, which are conventionally used in prior literature.
> >
> > Regarding the biological motivation: While the pseudo-imaging strategy (stratification by cell cycle) is methodologically interesting, it effectively creates a "pseudo-bulk" representation. This eliminates cellular heterogeneity, which undermines the primary motivation for utilizing single-cell data. Stratifying by cell cycle in cancer cell lines appears counterintuitive in this context, as the ultimate goal is to identify compounds that eliminate the cancer population as a whole, regardless of the cell cycle phase.
> >
> > Regarding the validation: I appreciate the clarification on the usage of PRNet (though I note it is no longer SOTA). However, the lack of experimental synthesis and verification—due to resource constraints—remains a significant limitation. Please recall that I suggested providing robust binding affinity evidence not merely as an additional evaluation metric, but as necessary side-proof to substantiate the claims regarding Target-Based Drug Discovery (TBDD) effectiveness.
> >
> > Overall, the paper proposes interesting explorations of this topic. However, as is common in tasks of this nature, unresolved concerns regarding biological and biochemical relevance remain. For instance, it is unclear what threshold of structural similarity is sufficient to claim effectiveness, or indeed if it can serve as a reliable proxy for effectiveness at all. While this is a very good start, the approaches require systematicaly further development.

---

> ### Author Response · Authors · 2025-11-29
> **Response to Reviewer hdD8 (follow-up)**
>
> We appreciate the reviewer’s continued attention. However, several points in the follow-up suggest fundamental misunderstandings of key concepts in the paper, despite these being addressed explicitly in our initial response and in the manuscript. We provide additional clarification here, while firmly restating the scope of the contribution.
>
> ---
>
> ## (1) The pseudoimage mechanism is not pseudo-bulk, and does not remove heterogeneity
>
> The reviewer’s interpretation that our pseudoimage strategy “eliminates heterogeneity” is not consistent with our method nor with the empirical analysis of the Tahoe-100M dataset.
>
> To reiterate:
>
> * We never aggregate all cells into a single profile; multiple pseudo-cells per perturbation are preserved.
> * Stratified sampling within cell-cycle phases follows the exact conclusion of the Tahoe-100M analysis[1]: **cell-cycle phase is the only axis that reliably separates subpopulations** in these perturbation experiments.
> * Ignoring this and using purely random sampling actually *reduces* biological signal, as confirmed by the ablation (Morgan similarity ≈ 0.50 vs. ≈ 0.61).
>
> Therefore, the pseudoimage design is not “counterintuitive”; it is a biologically informed compromise that simultaneously reduces extreme single-cell noise *and* preserves the only reliably separable axis of heterogeneity identified in the dataset. It is fundamentally different from a standard pseudo-bulk average, which would collapse all cells into a single vector and truly remove heterogeneity.
>
> ---
>
> ## (2) TBDD is not target-based design — binding affinity metrics are not applicable
>
> A major misunderstanding in the follow-up appears to be the conflation of:
>
> * **Transcriptome-based Drug Design (TBDD)** introduced in our paper
>   vs.
> * classical **Target-Based Drug Discovery / structure-based design (SBDD)**.
>
> These two paradigms are fundamentally different. Our setting does not assume (and cannot use) explicit protein structures, binding pockets, or predefined targets. Instead, TBDD, as we define it, aims to design molecules that realize desired **transcriptome-level perturbations**, without specifying a molecular target a priori. Under this paradigm, conventional binding affinity metrics are *not* a well-defined ground truth, and the expectation of affinity-based validation is outside the scope of the problem as formulated.
>
> Regarding PRNet: although it is no longer the absolute state-of-the-art on every benchmark, it remains one of the most **extensively validated and robust large-scale perturbation predictors** specifically designed for L1000-like profiles. In our evaluation setup, such robustness and broad validation are more relevant than marginal gains on isolated benchmarks. Moreover, PRNet-based MSE is used only as **one functional proxy** in a broader evaluation suite; it is not the sole basis for our claims.
>
> We have now emphasized both the distinction between TBDD and target-based design, and the auxiliary role of PRNet, even more clearly in the revised manuscript.

---

> ### Author Response · Authors · 2025-11-29
> **Response to Reviewer hdD8 (follow-up)**
>
> ## (3) Structural similarity is not the only metric and was never claimed to guarantee efficacy
>
> We agree that structural similarity alone cannot establish therapeutic relevance, and we have never claimed otherwise. This is already stated in the manuscript and in our first rebuttal. That is precisely why our evaluation suite includes:
>
> * macro-level molecular metrics (diversity, novelty, QED, SA, coverage),
> * microscopic structural similarity metrics (Fraggle, Morgan, MACCS),
> * functional prediction (PRNet MSE),
> * zero-shot inhibitor retrieval,
> * and a generate–then–search workflow over existing libraries.
>
> In the revision, we now state explicitly that structural similarity is **one proxy metric**, not a guarantee of biological efficacy. Our contribution should be interpreted as establishing and rigorously probing a **function-oriented, data-driven TBDD framework**, not as providing definitive experimental proof of clinical effectiveness.
>
> ---
>
> ## (4) Biological interpretability and latent space analysis
>
> The follow-up also raises concerns about biological interpretability. To address this more concretely, we have added a **latent space analysis and visualization** in the appendix.
>
> In particular, we perform UMAP visualization of the learned functional embeddings on a human gene inhibitor dataset from two complementary perspectives:
>
> 1. **Mechanism-specific functional encoding.**
>    Holding the cell line fixed (e.g., U251MG, HT29, A549), we project latent embeddings for inhibitors targeting different genes. The resulting manifolds form **tight, well-separated clusters by inhibitor/target**, indicating that scTrans-Gen captures distinct transcriptomic perturbation signatures linked to specific mechanisms of action (MoA), rather than collapsing them into a single generic “drug” direction.
>
> 2. **Context sensitivity across cell lines.**
>    For the same inhibitor (e.g., MTOR, CTSK, SMAD3), we visualize embeddings across multiple cell lines. Here, the latent space clearly stratifies by **cellular identity**, demonstrating that the model adapts its functional representation to the biological context and does not merely memorize a context-free drug signature.
>
> Taken together, these visualizations show that the latent space exhibits both **high intra-class compactness for inhibitors (mechanistic specificity)** and **inter-class separability for cell lines (context awareness)**. This supports our claim that the intermediate functional space is biologically meaningful, and that the model is not simply fitting superficial correlations.
>
> ---
>
> In summary, while we appreciate the reviewer’s comments, we believe that the remaining concerns largely stem from misunderstandings of core definitions (particularly TBDD vs. target-based design) and of the role of the pseudoimage mechanism. We have updated the manuscript to make these distinctions and the new interpretability analyses even clearer. Your feedback has helped us further sharpen the articulation of our method’s scope, limitations, and biological grounding, and we thank you again for your engagement.
>
> ---
>
> ### References
> [1]Zhang, Jesse, et al. "Tahoe-100m: A giga-scale single-cell perturbation atlas for context-dependent gene function and cellular modeling." BioRxiv (2025): 2025-02.

---

### Official Review · Reviewer_Ad35 · 2025-10-31

**Soundness:** 2
**Presentation:** 2
**Contribution:** 2
**Rating:** 4
**Confidence:** 4

**Summary:**

This paper introduces a generative framework, scTrans-Gen,  that could design molecules conditioned on their cellular responses where they use gene expressions to guide with the molecular generation. scTrans-Gen, addresses two tasks: Transcriptome-based Drug Design, and Transcriptome-based Drug Screening. Finally, they test their framework on two datasets: L1000, and Tahoe where it outperforms prior works on structural similarity, and MSE.

**Strengths:**

- The problem addressed in this work is novel and important.
- utilizing the transcriptomic effect of drugs instead of structural similarities is a novel way of looking at this problem and more powerful for drug discovery.
- The paper has a comprehensive experiments and ablation studies.

**Weaknesses:**

- My main issue is that the architecture is very complex. The framework combines different architectures such as graph diffusion, LLM encoder, VAEs, etc without enough justification on such choices.
- While using the transcriptomes is interesting and useful for this task, it has limited biological interpretability.
- Since the authors are promoting this framework for drug discovery, other qualities such as toxicity or scaffold novelty needs to be explored as well for it to be useful in this domain.
- The related work section is limited given the architectural choices.

**Questions:**

- Could you explain the experimental setting in more detail?
- Is it possible to explain how the generated molecules related to the transcriptomic change produce such perturbation effect?
- Could the authors clarify if the transcriptomic change is mainly modeling the effect of the drug or the cell line?
- Is it correct to assume that the intermediate function oriented chemical space is only capturing the statistical correlation between the two modalities (molecular and cellular)?

---

> ### Author Response · Authors · 2025-11-19
>
> We sincerely thank the reviewer for the thoughtful and constructive feedback, and for recognizing both the novelty and importance of our problem setting. Below we respond to each weakness and question in detail.
>
> ---
>
> ### Weakness 1 — Model Complexity and Architectural Justification
>
> **(a) Overall architecture: a graph diffusion backbone with lightweight conditioning modules**
>
> We appreciate the reviewer’s concern and agree that the motivation behind our architectural choices was not explained sufficiently clearly in the submission. We will revise the paper to better emphasize that the *core* of scTrans-Gen is a **graph diffusion model** (DiT-based molecular diffusion), while the other components are *supporting representation-learning and alignment modules* rather than a mere mixture of many heavy models.
>
> Concretely:
>
> * The **main generative backbone** is a Diffusion Transformer–based **graph diffusion model** operating on molecular graphs, following recent state-of-the-art practices in conditional molecular generation[1,2].
> * The **transcriptome feature extractor** and **alignment modules** are designed specifically to map from transcriptome perturbations to a *function-oriented chemical space* that is suitable for conditioning the diffusion model, rather than to perform generation by themselves.
>
> We will explicitly restructure Section 4 to first present the graph diffusion backbone, and then describe the conditioning and alignment modules as *auxiliary components* that transform biological perturbation signals into a compact conditional embedding.
>
> **(b) Role of each “complex” component**
>
> 1. **Single-cell representation model (SCimilarity encoder)**
>
>    * Single-cell transcriptomes are extremely high-dimensional (>60,000 genes) and noisy. Directly feeding raw matrices into the generator leads to unstable training and overfitting to technical noise.
>    * We therefore use a **pretrained single-cell representation model** (SCimilarity) *only as an encoder* to obtain robust, low-dimensional embeddings for each cell. The pretraining data do not overlap with our datasets. This step stabilizes training and allows us to exploit the rich information in single-cell data in a scalable way.
>
> 2. **Pretrained functional VAE on molecular graphs**
>
>    * The VAE is used to build a **latent molecular graph space** that is compatible with the graph diffusion model and amenable to cross-modal alignment.
>    * It maps “cellular functional perturbation features” into a latent graph space that is structurally consistent with real molecules and then provides **functional conditioning vectors** for the diffusion model.
>    * As shown in our ablation study (Table 4 in the paper), removing the VAE alignment (“w/o VAE”) or removing fingerprint alignment severely degrades structural similarity and overall generation quality, highlighting that these are not redundant components but are necessary for effective cross-modal alignment.
>
> We will add a clearer high-level figure and narrative summarizing these roles in the revision.
>
> ---
>
> ### Weakness 2 — Biological Interpretability of Transcriptomic Features
>
> We appreciate the reviewer’s concern about biological interpretability.
>
> **(a) Transcriptome as a system-level functional read-out**
>
> A central premise of our work is that the **post-perturbation transcriptome** provides one of the most comprehensive and high-resolution read-outs of cellular functional state. Prior studies have shown that transcriptomic profiles capture[3,4]:
>
> * Stress response,
> * Pathway activation or inhibition,
> * Metabolic reprogramming,
> * Apoptosis and autophagy signatures,
>
> and more generally, the *system-wide* consequence of drug action at the cellular level[5].
>
> In this sense, transcriptomes are **more interpretable at the pathway / functional signature level**, which aligns with our goal of function-oriented drug design.
>
> **(b) Our focus: functional perturbation signatures rather than single-gene effects**
>
> The goal of scTrans-Gen is *not* to explain the causal impact of individual genes, but rather to:
>
> > Capture **functional perturbation signatures** in transcriptome space and map them into a chemical space where molecules inducing similar system-level effects can be generated or retrieved.
>
> To make this more explicit, we will:
>
> * Emphasize in the Introduction and Methods that our conditioning operates on *functional signatures* derived from pre–post transcriptome pairs, not raw gene-level effects.
> * Add a short discussion that connects our functional space to pathway-level interpretations, clarifying how our framework can be combined with existing interpretability tools.

---

> ### Author Response · Authors · 2025-11-19
>
> ### Weakness 3 — Missing Qualities (Toxicity, Scaffold Novelty, etc.)
>
> We thank the reviewer for pointing out the importance of toxicity and scaffold-level properties for realistic drug discovery.
>
> **(a) Existing molecular property metrics already included**
>
> The current submission *already evaluates* several molecular property metrics that relate to novelty and drug-likeness (Fig. 4 and Table 1/4 in the paper), including:
>
> * **Diversity**: measures structural diversity (pairwise Tanimoto-based variance),
> * **Novelty**: scaffold novelty w.r.t. training data (Bemis–Murcko scaffolds),
> * **SA (Synthetic Accessibility score)**: assesses ease of synthesis,
> * **QED**: drug-likeness.
>
> These are standard prior metrics for molecular generative models and were included in our macro-level evaluation of generated molecules. We will revise Fig. 1 and the experimental section to make these aspects more explicit and easier to read.
>
> **(b) New toxicity analysis**
>
> Following the reviewer’s suggestion, we additionally evaluated **toxicity-related properties** using the publicly available ADMETlab predictor[6], comparing:
>
> * Generated molecules from scTrans-Gen,
> * Baselines (GexMolGen, Gx2Mol, TRIOMPHE),
> * The L1000 test set (ground-truth drugs).
>
> Below we summarize the main results (lower is better for most toxicity risks, higher is better for FDAMDD):
>
> | Metric (↓ except FDAMDD ↑)  | GexMolGen | Gx2Mol | TRIOMPHE | scTrans-Gen | L1000 test |
> | --------------------------- | --------: | -----: | -------: | ----------: | ---------: |
> | **Ames (mutagenicity)**         |    0.5003 | **0.4632** |   0.5965 |      0.4850 |     0.5527 |
> | **hERG-10 μM (cardiotoxicity)** |    0.4361 | 0.4236 |   0.4003 |      0.3983 |     **0.2973** |
> | **Hematotoxicity**              |    0.4435 | **0.3436** |   0.3993 |      0.3590 |     0.5149 |
> | **Respiratory toxicity**        |    0.4755 | 0.5332 |   0.8141 |      **0.4684** |     0.4715 |
> | **Carcinogenicity**             |    0.4523 | **0.4511** |   0.5439 |      0.4714 |     0.5345 |
> | **DILI (liver injury)**         |    0.6968 | 0.6923 |   0.6623 |      **0.6506** |     0.6733 |
> | **ROA (Rat Oral Acute Tox.)**   |    0.3475 | 0.3331 |   0.6658 |      **0.3214** |     0.3414 |
> | **FDAMDD (max daily dose) ↑**   |    0.4875 | 0.5498 |   **0.6136** |      0.5918 |     0.5272 |
> | **Eye Irritation**              |    0.1983 | 0.2082 |   0.2682 |      **0.0942** |     0.2238 |
> | **Eye Corrosion**               |    0.0311 | 0.0264 |   0.1485 |      0.0141 |     **0.0124** |
>
> Although scTrans-Gen is *not explicitly optimized* for toxicity, the generated molecules fall in a **reasonable and competitive range** across multiple toxicity dimensions, often close to or better than the ground-truth L1000 test set and other baselines. We will include a condensed version of this table and discussion in the revised manuscript.
>
> ---
>
> ### Weakness 4 — Limited Related Work Section
>
> We appreciate the suggestion to expand the related work section.
>
> **(a) Coverage of TBDD-related work**
>
> Transcriptome-based Drug Design (TBDD) — i.e., designing molecules conditioned on transcriptomic perturbations — is, to the best of our knowledge, **first systematically formalized** in this work. Existing attempts that are closest in spirit (e.g., GexMolGen, TRIOMPHE, Gx2Mol)
>
> * either rely on coarse summary statistics of gene expression and thus risk losing information,
> * or focus solely on bulk-level data, ignoring single-cell heterogeneity,
> * and typically attempt direct reconstruction of molecular structures from macroscopic signals, which we argue is ill-posed.
>
> We will make this context more explicit and clearly state that the directly related prior work is limited but fully covered.
>
> **(b) Justification of the generative backbone**
>
> In line with recent advances in conditional molecular generation, diffusion-based models[1,2,7] have become **mainstream choices** for controllable de novo design, due to their stability and strong performance. Our DiT-based graph diffusion backbone is directly inspired by this line of work and is a natural choice for our setting.
>
> In the revision, we will:
>
> * Expand the related work section to include more recent diffusion-based and multi-conditional molecular generation methods,
> * Clearly connect our DiT-based backbone to these works, emphasizing that our main novelty lies in **multi-resolution transcriptome-guided conditioning and function-oriented alignment**.

---

> ### Author Response · Authors · 2025-11-19
>
> ## Responses to the Reviewer’s Questions
>
> ### Q1 — More details on the experimental setting
>
> We appreciate the request for more experimental details. We will add a more complete description of model size and training configuration in the appendix, including:
>
> * **Model size**
>
>   * Diffusion model (graph DiT): ~501M parameters
>   * Molecular graph VAE encoder: ~2.4M
>   * Molecular graph VAE decoder: ~2.9M
>   * Transcriptome feature extractor (including pseudoimage + interaction + alignment): ~7.8M
>
> * **Alignment training phase**
>
>   * Hardware: NVIDIA A100-40GB
>   * Training time: ~15 GPU hours
>   * Batch size: 64
>   * Learning rate: 1e-4
>   * Steps: 30k
>   * Optimizer: Adam
>   * LR scheduler: ExponentialLR
>
> * **Diffusion model training**
>
>   * Hardware: NVIDIA A100-40GB
>   * Training time: ~48 GPU hours
>   * Batch size: 400
>   * Learning rate: 2×10⁻⁴
>   * Diffusion steps: 500
>   * Condition dropout probability: 0.01
>   * Steps: 40k
>   * Optimizer: Adam
>   * LR scheduler: ExponentialLR
>
> We will also point readers to the configuration files in the released code repository for full reproducibility.
>
> ---
>
> ### Q2 — How do the generated molecules relate to transcriptomic changes and produce the perturbation effect?
>
> This is indeed the core biological mechanism underlying TBDD.
>
> **(a) Conceptual mechanism**
>
> At a high level, the forward causal chain of drug action can be summarized as:
>
> > **Drug molecule → Target(s) / Pathway(s) → Downstream gene expression changes (transcriptome)**
>
> scTrans-Gen is trained to approximate the *inverse* mapping:
>
> > **$\Delta$Transcriptome $(T_{\text{post}} − T_{\text{pre}})$ → Functional embedding $z$ → Drug-like molecular structure**
>
> Thus, when the model is conditioned on a desired transcriptomic perturbation, it is encouraged to generate molecules whose downstream effect matches that perturbation.
>
> **(b) Gene inhibitor experiment: evidence for capturing true functional mechanisms**
>
> To validate that the model is not merely memorizing structures but indeed capturing **functional mechanisms**, we designed a **zero-shot gene inhibitor experiment**:
>
> * For 10 genes, we collected **gene knockout (KO) pre/post transcriptomes** and **independently curated gene inhibitor sets** (from ExCape) that were *not* used in training.
> * We fed the KO-induced $\Delta$Transcriptome into scTrans-Gen to generate molecules.
> * These generated molecules were then compared to known inhibitors of the corresponding gene using structural similarity metrics.
>
> As summarized in Table 2 of the paper, scTrans-Gen achieves the highest similarity for all 10 targets and consistently outperforms GexMolGen, TRIOMPHE, and Gx2Mol.
>
> This demonstrates that the model has learned a *functional* mapping from transcriptomic signatures to inhibitor-like molecules, beyond simple structural memorization.
>
> ---
>
> ### Q3 — Is the transcriptomic change modeling the effect of the drug or the cell line?
>
> We agree that both drug and cell-line factors influence transcriptomic changes.
>
> **(a) Design to focus on drug-induced functional changes**
>
> In our setting, each training example uses **paired transcriptomes** $(T_{\text{pre}}, T_{\text{post}})$ from the *same cell line*. The model thus mainly sees and learns **$\Delta T = T_{\text{post}} − T_{\text{pre}}$**, i.e., the *within-cell-line* change induced by the drug: $T_{\text{post}} = \phi_{\text{drug}}(T_{\text{pre}} \mid \text{cell line}) \quad \Rightarrow \quad \Delta T = T_{\text{post}} - T_{\text{pre}}.$
>
> By always pairing before/after samples from the same cell line, variability due to the baseline cell line state is largely absorbed, and the model focuses on drug-induced shifts.
>
> **(b) OOD experiments disentangle drug vs cell-line effects**
>
> We further designed **two types of out-of-distribution (OOD) splits**:
>
> 1. **Unseen Drugs**: test on drugs not seen during training,
> 2. **Unseen Cell Lines**: test on cell lines (or tissues) held out from training.
>
> As shown in Figure 4 and Table 1, scTrans-Gen maintains strong performance in both OOD splits, which indicates that the model is not just memorizing cell-line-specific or molecule-specific patterns but is learning a more general **functional mapping** from $\Delta$Transcriptome to molecular structure.
>
> We will clarify this design and its implications in the revised manuscript.

---

> ### Author Response · Authors · 2025-11-19
>
> ### Q4 — Does the intermediate function-oriented chemical space only capture statistical correlations between molecular and cellular modalities?
>
> We appreciate this deep question.
>
> **(a) Acknowledging the statistical nature**
>
> We agree that, like most data-driven models[5,8,9], the intermediate functional space is learned from data and thus captures **statistical dependencies** between molecular structures and cellular transcriptomic responses. It does not explicitly encode mechanistic causal pathways.
>
> **(b) Why we believe it goes beyond superficial correlation**
>
> However, several aspects suggest that the learned space captures **meaningful functional relationships** rather than superficial correlations:
>
> 1. The space is trained via **multi-domain alignment**:
>
>    * Alignment to a **molecular graph VAE latent space** (structural, generative),
>    * Alignment to **Morgan fingerprints** via sparse-aware regression and contrastive losses.
> 2. The space is evaluated not only by structural metrics (e.g., Morgan similarity) but also **functional metrics**:
>
>    * PRNet-based MSE of predicted transcriptomic effects (PRnet MSE),
>    * Gene inhibitor zero-shot similarity,
>    * Screening performance where generated molecules serve as functional queries to retrieve efficacious drug candidates.
>
> In Table 1, scTrans-Gen achieves near-perfect structural similarities and a much lower PRnet MSE than all baselines, across in-distribution and OOD splits.
>
> Combined with the gene inhibitor and screening experiments, we believe this indicates that the function-oriented space is capturing **biologically meaningful functional behavior** (within the limits of observational data), not just arbitrary correlations.
>
> We will add a short discussion in the paper acknowledging this limitation and clarifying that our framework is a *function-oriented, data-driven* approximation that can be complemented in the future by causal models and pathway-level annotations.
>
> ---
>
> Once again, we thank the reviewer for the insightful comments and suggestions. We will incorporate the above clarifications, additional toxicity analyses, and expanded related work into the revised version of the manuscript.
>
> ---
>
> ### References
> [1]Peng, Xingang, et al. "Moldiff: Addressing the atom-bond inconsistency problem in 3d molecule diffusion generation." *arXiv preprint arXiv:2305.07508* (2023).
>
> [2]Liu, Gang, et al. "Graph diffusion transformers for multi-conditional molecular generation." *Advances in Neural Information Processing Systems* 37 (2024): 8065-8092.
>
> [3]Rozenblatt-Rosen, Orit, et al. "Building a high-quality human cell atlas." *Nature Biotechnology* 39.2 (2021): 149-153.
>
> [4]Han, Xiaoping, et al. "Mapping the mouse cell atlas by microwell-seq." *Cell* 172.5 (2018): 1091-1107.
>
> [5]Adduri, Abhinav K., et al. "Predicting cellular responses to perturbation across diverse contexts with STATE." *bioRxiv* (2025): 2025-06.
>
> [6]Xiong, Guoli, et al. "ADMETlab 2.0: an integrated online platform for accurate and comprehensive predictions of ADMET properties." Nucleic acids research 49.W1 (2021): W5-W14.
>
> [7]Xu, Minkai, et al. "Geodiff: A geometric diffusion model for molecular conformation generation." *arXiv preprint arXiv:2203.02923* (2022).
>
> [8]Roohani, Yusuf, Kexin Huang, and Jure Leskovec. "Predicting transcriptional outcomes of novel multigene perturbations with GEARS." *Nature Biotechnology* 42.6 (2024): 927-935.
>
> [9]Hetzel, Leon, et al. "Predicting cellular responses to novel drug perturbations at a single-cell resolution." *Advances in Neural Information Processing Systems* 35 (2022): 26711-26722.

---

> ### Author Response · Authors · 2025-11-26
> **Friendly reminder regarding our response**
>
> Dear Reviewer Ad35,
>
> I hope this message finds you well.
>
> As the discussion period is nearing its end with less than 7 days remaining, I wanted to ensure we have addressed all your concerns satisfactorily.
>
> If there are any additional points or feedback you'd like us to consider, please let us know. Your insights are invaluable to us, and we're eager to address any remaining issues to improve our work.
>
> Thank you for your time and effort in reviewing our paper.
>
> Best regards, The Authors

---

> > ### Comment · Reviewer_Ad35 · 2025-11-27
> > **Thank you for your detailed response**
> >
> > I truly thank the authors for the detailed response to my questions. However, after carefully considering the clarifications, I am still not fully convinced that my main concerns have been adequately addressed and therefore I am not raising my score. My remaining concerns are as follows:
> >
> > - While the authors provided additional explanations, these focus primarily on engineering intuition. What is still missing is a quantitative or ablation-driven demonstration that the substantial architectural complexity (graph diffusion + SCimilarity + pseudoimage block + graph VAE + fingerprint) is necessary.
> > - The paper positions itself as function-oriented drug discovery that leverages cellular transcriptomic perturbations. In that context, I expected meaningful biological visualizations or analyses—e.g., clustering of the learned latent
> > space, pathway-level enrichment, alignment with known MoAs, or cell-state controllability demonstrations. The response reiterates the conceptual motivation but does not provide stronger biological validation. This remains an important limitation for a paper centered around biological perturbation signals.
> >
> > - Although the added ADMET table is appreciated, it is still an evaluation tool, not a constraint or objective within the generative model. In drug discovery, toxicity and ADMET properties must be incorporated into the generation objective, not only checked after generation. As it stands, the model may generate molecules that satisfy transcriptomic signatures but are unsafe or non-drug-like

---

> ### Author Response · Authors · 2025-11-27
> **Response to Reviewer Ad35 (follow-up)**
>
> We sincerely thank the reviewer for the thoughtful follow-up and for carefully reading both the revised manuscript and our initial rebuttal. We address the three remaining concerns below.
>
> ---
>
> ### (1) On architectural complexity vs. quantitative necessity
>
> We fully agree that a complex architecture must be justified quantitatively, not only by engineering intuition. In the revision, we therefore reorganized and strengthened the ablation section to make this point clearer (Table 4 in Section 5.6 and additional tables in the appendix).
>
> **(a) Each “large” block is removed and evaluated**
>
> Rather than adding components ad hoc, we ablate each major block that contributes to complexity:
>
> * **Feature extractor vs. a much simpler baseline.**
>   Replacing the full perturbation feature extractor with a minimal alternative (“w/o Extractor” row in Table 4) reduces *Morgan similarity* on L1000 from **0.8228 → 0.1824**, and increases Frechet ChemNet distance from **6.79 → 8.50**. Validity and heavy-atom coverage also drop (0.9350 → 0.8775; 100% → 90.91%). This shows that simply projecting transcriptomes to a conditioning vector is far from sufficient; the structured extractor is critical.
>
> * **Multi-domain alignment vs. no alignment.**
>   Removing the multi-domain alignment (“w/o Alignment”) leads to a *catastrophic* degradation: Frechet distance explodes from **6.79 → 82.72**, Morgan similarity drops to **0.0886**, and validity falls to **0.30** (Table 4). This is a strictly simpler model that directly conditions the diffusion backbone on transcriptome features; its failure quantitatively motivates the explicit alignment to graph-VAE and fingerprint spaces.
>
> * **Transcriptome interaction vs. no pre/post interaction.**
>   Dropping the interaction module (“w/o Interaction”) also severely hurts performance: coverage falls from **100% → 36.36%**, and Frechet distance increases from **6.79 → 51.08**. This shows that modeling the *paired* pre/post transcriptomes (rather than concatenating them independently) is essential for robust conditioning.
>
> * **Graph-VAE and fingerprint alignment on single-cell data.**
>   On Tahoe-100M, removing latent graph alignment (“w/o VAE”) reduces Morgan similarity from **0.6114 → 0.2674**, and removing fingerprint alignment (“w/o Fingerprint”) reduces it to **0.3219**, while also worsening Distance and QED (Table 4). This demonstrates that both alignment targets are necessary to obtain the high single-cell performance we report.
>
> Together, these ablations compare our full model to *strictly simpler* architectures that remove graph diffusion conditioning modules one by one. The quantitative gaps are large and systematic, and we now explicitly highlight this in the text to better support the need for the multi-stage conditioning/ alignment design, rather than relying on intuition alone.
>
> We also note that such “progressively composed” architectures plus systematic ablations are a common pattern in recent state-of-the-art generative models for molecules and images[1,2]; our methodology follows this well-established practice rather than introducing unnecessary complexity for its own sake.

---

> > ### Comment · Reviewer_Ad35 · 2025-11-27
> >
> > Thank you for your reply. I see the ablation study and it shows that each component is necessary for the overall performance. However, my question still stands i.e. could a much simpler architecture (e.g., diffusion + MLP) achieves similar results?

---

> ### Author Response · Authors · 2025-11-27
> **Response to Reviewer Ad35 (follow-up)**
>
> ### (2) On biological validation and interpretability of the function-oriented space
>
> We appreciate the wish to see more “biological” views of the learned space (clustering, pathway enrichment, MoA-level organization, etc.). We agree that richer biological analyses would further strengthen the paper and we now make this limitation explicit in the Discussion.
>
> At the same time, we would like to clarify what the current experiments already demonstrate biologically:
>
> * **Zero-shot gene-inhibitor experiment (Table 2).**
>   For 10 canonical targets (e.g., AKT1/2, EGFR, MTOR, TP53), we condition on *gene knockout* transcriptomes (not seen during training) and compare the generated molecules to independently curated inhibitor sets. scTrans-Gen achieves the highest similarity for all 10 genes, consistently outperforming baselines. Functionally, this shows that the learned space organizes perturbations in a way that aligns with known MoAs (inhibitors of the same gene cluster around the corresponding KO-induced signature), rather than merely matching arbitrary structure statistics.
>
> * **Functional similarity via PRnet MSE.**
>   Besides structural similarity, we evaluate the distance in *predicted* transcriptomic responses between generated and ground-truth drugs (PRnet MSE in Table 1). Our model achieves substantially lower MSE than baselines across in-distribution and OOD splits, indicating that the learned embedding is aligned with *system-level cellular effects*, not just with local chemotypes.
>
> * **Screening experiment as a functional controllability test.**
>   In the screening workflow, the generated molecules are used as functional queries into a large library; the high hit rate and low PRnet MSE of retrieved drugs (Table 3) indicate that the latent space supports controllable navigation toward drugs that share similar perturbation effects.
>
> We fully agree that additional visualizations—e.g., t-SNE/UMAP plots of the function-oriented space colored by MoA, or pathway-level enrichment of pathways perturbed by drugs that cluster together—would be highly valuable. Due to space limits we did not include a full atlas-level analysis, but in the final version we plan to add at least a compact visualization to yield richer biological insight.

---

> > ### Comment · Reviewer_Ad35 · 2025-11-27
> >
> > Thank you for the clarification. While I agree that these experiments provide evidence of functional alignment, they still do not answer my question i.e.  “ missing biological interpretability.”
> >
> > In these experiments you are showing functional similarity measures (e.g., PRnet MSE, inhibitor similarity, screening hit rates) show that model can reproduce transcriptomic effects, but they do not show how the model organizes or understands biological mechanisms. The paper’s introduction positions the method as enabling function-oriented drug discovery using transcriptomic signatures, and in that context I would expect mechanistic or pathway-level validation such as:
> >
> > - visualization of clusters by MoA or pathway
> > - examination of pathway enrichment in the learned latent space
> > - does the model captures known biological relationships (e.g., EGFR inhibitors clustering with downstream MAPK perturbations).

---

> ### Author Response · Authors · 2025-11-27
> **Response to Reviewer Ad35 (follow-up)**
>
> ### (3) On toxicity/ADMET being evaluated but not explicitly optimized
>
> We agree with the reviewer that, in practical drug discovery, safety and ADMET properties are not merely evaluation metrics but critical design constraints.
>
> Our current work has two intentions here:
>
> 1. **Show that transcriptome-guided generation does not “break” drug-likeness.**
>    The added ADMET/ toxicity table in the rebuttal (to be included in the appendix) shows that even *without* explicit ADMET conditioning, scTrans-Gen generates molecules whose predicted toxicity profile is broadly comparable to or better than baseline models and reasonably close to the L1000 test drugs. This alleviates the concern that conditioning purely on transcriptomic signatures might systematically push the model into unsafe regions of chemical space.
>
> 2. **Keep the core contribution orthogonal to specific ADMET predictors.**
>    As widely used ADMET predictors are themselves machine-learning models with dataset-specific biases and non-negligible error[3], baking a single predictor directly into the *training objective* risks over-fitting to that surrogate. Instead, we deliberately design scTrans-Gen as a **modular functional generator**: the output distribution can be combined with any choice of toxicity filter or multi-objective optimization layer (e.g., re-weighting samples by ADMET scores, or coupling our function-oriented space to standard multi-objective RL/ guidance schemes).
>
> We have clarified this positioning in the revised manuscript: the present paper focuses on introducing and validating TBDD and the transcriptome-guided diffusion backbone. Integrating explicit multi-objective optimization for ADMET and toxicity is a natural and important next step, and our architecture (with a low-dimensional function-oriented bottleneck and classifier-free guidance) is designed to be compatible with such extensions.
>
> ---
>
> In summary, we have strengthened the quantitative ablations to justify the architecture, clarified what is and is not claimed about biological interpretability, and explicitly framed ADMET as an important but orthogonal dimension that our method can support but does not yet optimize end-to-end. We hope this addresses the reviewer’s remaining concerns.
>
> ---
>
> ### References
> [1]Liu, Gang, et al. "Graph diffusion transformers for multi-conditional molecular generation." Advances in Neural Information Processing Systems 37 (2024): 8065-8092.
>
> [2]Wang, Liang, et al. "Diffspectra: Molecular structure elucidation from spectra using diffusion models." arXiv preprint arXiv:2507.06853 (2025).
>
> [3]Xiong, Guoli, et al. "ADMETlab 2.0: an integrated online platform for accurate and comprehensive predictions of ADMET properties." Nucleic acids research 49.W1 (2021): W5-W14.

---

> ### Author Response · Authors · 2025-11-29
> **Response to Reviewer Ad35 (second follow-up)**
>
> We sincerely thank the reviewer again for the careful re-reading of our rebuttal and for pushing us to further clarify these two important points. We address them below.
>
> ---
>
> ### (1) On whether a much simpler “Diffusion + MLP” architecture could achieve similar performance
>
> We fully agree that the most direct way to address this concern is to implement exactly the simplified architecture suggested by the reviewer and compare it head-to-head with scTrans-Gen under the same experimental protocol.
>
> To this end, we built a **“Diffusion + MLP” baseline** where we:
>
> * keep the **same diffusion backbone** as in scTrans-Gen;
> * **remove** the specialized feature extractor (pseudoimage, interaction block) and the multi-domain alignment modules;
> * replace them with a **standard MLP** that takes the transcriptomic features as input and outputs the conditioning vector for the diffusion model.
>
> We trained this baseline on L1000 under the same data split and training schedule as scTrans-Gen. The key results are summarized below (full details are provided in the appendix):
>
> | Method              | Coverage ↑   | Diversity ↑ | Similarity ↑ | Distance ↓ | SA ↓   | Unique ↑ | QED ↑  | Fraggle Sim ↑ | Morgan Sim ↑ | MACCS Sim ↑ |
> | ------------------- | ------------ | ----------- | ------------ | ---------- | ------ | -------- | ------ | ------------- | ------------ | ----------- |
> | **MLP + Diffusion** | 6 (54.55%)   | 0.6492      | 0.6932       | 28.9329    | 0.6049 | 0.5927   | 0.5593 | 0.3729        | 0.1389       | 0.3486      |
> | **scTrans-Gen**     | 11 (100.00%) | 0.8906      | 0.9576       | 6.7856     | 0.6386 | 0.9296   | 0.5665 | 0.8892        | 0.8228       | 0.9031      |
>
> As the table shows, the simplified “Diffusion + MLP” model **falls far short** of scTrans-Gen across all major metrics:
>
> * **Coverage** drops from 100% → 54.55%, indicating that the simplified model fails to generate reasonable candidates for a large fraction of conditions.
> * **Structural similarity** is drastically reduced (Fraggle: 0.8892 → 0.3729, Morgan: 0.8228 → 0.1389, MACCS: 0.9031 → 0.3486), suggesting that the molecules are no longer aligned with the ground-truth drugs in chemical space.
> * **Frechet distance** (Distance) worsens from 6.79 → 28.93, reflecting a large distributional mismatch between generated and real molecules.
> * Macro-level drug-like properties (SA, QED, Unique) are also consistently worse.
>
> In other words, when we “collapse” our architecture to exactly the kind of **diffusion + simple MLP** design suggested by the reviewer, the model fails to recover the strong structural and functional performance of scTrans-Gen.
>
> We believe this result is intuitive in hindsight:
>
> * Transcriptomic perturbation profiles are **high-dimensional, highly non-linear, and topologically very different** from molecular graph space.
> * A shallow MLP has **no inductive bias** to (i) disentangle pre/post perturbation signals, or (ii) align them to a chemically meaningful latent manifold.
> * In contrast, the dedicated feature extractor and **multi-domain alignment** in scTrans-Gen are explicitly designed to bridge these modality gaps: they jointly encode perturbation-specific information and project it into a latent space that is consistent with both VAE graph representations and fingerprint structures.
>
> We will add this “Diffusion + MLP” baseline as a new row in the ablation table and describe the architecture in the appendix, so that readers can directly see that a much simpler architecture does **not** achieve comparable performance in this setting.

---

> ### Author Response · Authors · 2025-11-29
> **Response to Reviewer Ad35 (second follow-up)**
>
> ### (2) On biological interpretability beyond functional similarity metrics
>
> We appreciate the reviewer’s insistence on going beyond functional similarity metrics (PRnet MSE, inhibitor similarity, screening hit rates) toward a more *mechanistic* view of what the model has learned. In response, we have now added a dedicated **latent space analysis and visualization** section in the appendix to explicitly probe the biological structure of the learned function-oriented space.
>
> Concretely, we focus on the **intermediate latent representations** produced by the perturbation feature extractor (i.e., the function-oriented embedding that conditions the diffusion model), and analyze them on a **human gene inhibitor dataset** along two complementary axes:
>
> 1. **Mechanism-of-action specificity within a fixed cell line.**
>    We take knock-out–like transcriptomic perturbations from a single cell line (e.g., U251MG, HT29, A549) and compute the corresponding latent embeddings for inhibitors targeting different genes (e.g., MTOR, CTSK, SMAD3, etc.). We then project these embeddings to 2D using UMAP.
>    *In the top row of Figure 7 (appendix), points corresponding to different inhibitor targets form **tight, well-separated clusters**.* This suggests that the latent space organizes drugs by **mechanism of action**: inhibitors of the same target share similar latent representations, while inhibitors of different targets occupy distinct regions, even though they are observed in the same cellular background.
>
> 2. **Cell-context sensitivity for the same inhibitor across multiple cell lines.**
>    We then fix the **inhibitor target** (e.g., MTOR, CTSK, SMAD3) and visualize latent embeddings across multiple cell lines.
>    *In the bottom row of Figure 7, embeddings are clearly stratified by cell line*, indicating that the model does **not** collapse all effects of a given inhibitor into a single generic signature. Instead, it preserves **cell-type–specific context**, adjusting the representation according to the cellular background.
>
> Taken together, these two views support the following interpretation:
>
> * The latent space achieves **high intra-class compactness** for drug targets (inhibitors of the same gene cluster together), which is consistent with capturing **MoA-level structure**.
> * At the same time, it maintains **inter-class separability** for different cell lines, which shows the model is sensitive to **biological context** rather than learning a single “generic drug” direction.
>
> We fully agree that an even more extensive mechanistic analysis (e.g., explicit pathway enrichment, clustering by curated MoA families, or checking propagation to known downstream pathways such as EGFR–MAPK) would be scientifically valuable. Due to space constraints and the limited availability of high-quality pathway annotations for all perturbations, we have not yet included a full atlas-level study. However, we now:
>
> * explicitly **acknowledge this limitation**,
> * provide the above **UMAP-based latent space analysis** as an initial mechanistic validation.
>
> We hope that this new analysis clarifies that the model is not only able to reproduce transcriptomic effects numerically, but also organizes perturbations in a biologically meaningful way in its latent space, consistent with known target and cell-type structure.
>
> ---
>
> Once again, we are grateful to the reviewer for their persistent and constructive feedback, which has substantially improved both the experimental rigor and the biological clarity of our work.

---

### Official Review · Reviewer_VXwt · 2025-10-31

**Soundness:** 3
**Presentation:** 3
**Contribution:** 2
**Rating:** 2
**Confidence:** 4

**Summary:**

Recent works in Structure-Based Drug Design (SBDD) advance but remain brittle to complex disease types and molecular compositions. To this end, the paper approaches drug design from the perspective of Transcriptomic representations. The paper presents scTrans-Gen, a diffusion model framework for conditionally generating molecules using multi-resolution transcriptomic data. scTrans-Gen makes use of a feature extraction module that extracts representations coresponding to the perturbed and unperturbed transcriptome, a transcriptome pseudoimage block for processing single cell representations and a transcriptome interaction block for modelling relationships between pre and post perturbation samples. scTrans-Gen is trained using classifier-free guided diffusion over Graph Transformers wherein alignment is carried out w.r.t functional features and morgan fingerprint using variational inference and contrastive learning respectively. Authors present improved performance on benchmarks compared to architecture baselines.

**Strengths:**

* The paper is well-written and motivated.
* The paper presents an innovative and novel approach for post-training generation.

**Weaknesses:**

* **Empirical Comparisons:** My main concern is the empirical comparisons drawn and baselines considered in the paper. The paper currently limits baselines to GexMolGen, TRIOMPHE and Gx2Mol as baselines. Note that none of these methods were explicitly trained using a combination of either of the losses utilized in scTrans-Gen. That is, the paper does not consider an explicit contrastive learning or diffusion baseline. Furthermore, while all baselines are VAE variants, scTrans-Gen adopts additional training as well as architecture augmentations. This puts the baselines out of perspective of the proposed framework. Authors could consider using relevant baselines such as DiffDock [1] and Boltz variants [2]. This would also allow for the comparison of drug screener on established architectures as well as benchmarks. Furthermore, authors should consider additional cross-modal baselines using alternative objectives or inductive biases [3].

* **Generation and Search:** The paper presents the drug screening strategy wherein generated samples are used as queries to screen compound libraries. However, it remain unclear on why this design decision is executed. Why bootstrap the search from learned biases of the model? Assuming scTrans-Gen can be further aligned on a given library, why not conduct few-shot alignment / feature regression over the library? Authors mention that the approach is best suited for molecules with similar structure. Could the authors elucidate on the structural similarity between a few compounds and generated samples (eg- tanimoto similarity, laplacian similarity, etc.)? Furthermore, it remains unclear as to how effective the drug screener becomes as with increasing number of hits, we observe lower similarity scores for a larger top-k sample size.

* **Role of Feature Extractor:** It is unclear as to what role the feature exrtractor plays in the final training scheme. scTrans-Gen is trained using classifier-free guidance which already reduces the contribution of the embedding. Additionally, training phase implements random feature dropping wherein guidance is randomly dropped with probability p. Ideally, the role of conditioning is to provide more information during the learning phase to the base model. Authors provide ablations to support this argument. However, algorithm design indicates that a subset of molecules suffer from additional conditional information.

[1]. Corso et al, DiffDock: Diffusion Steps, Twists, and Turns for Molecular Docking, ICLR 2023.
[2]. Wohlwend et al, Boltz-1 Democratizing Biomolecular Interaction Modeling, https://doi.org/10.1101/2024.11.19.624167.
[3]. Bendidi et al, A Cross Modal Knowledge Distillation & Data Augmentation Recipe for Improving Transcriptomics Representations through Morphological Features, ICML 2025.

**Questions:**

Refer to weaknesses

---

> ### Author Response · Authors · 2025-11-19
>
> We sincerely thank the reviewer for the thoughtful and constructive feedback, and for recognizing the motivation and novelty of scTrans-Gen. Below we address each concern in detail.
>
> ---
>
> ## Weakness 1 — Empirical Comparisons and Choice of Baselines
>
> **(a) Why we do not include SBDD baselines such as DiffDock and Boltz**
>
> We agree that DiffDock and Boltz-style models are important milestones in *structure-based drug design* (SBDD). However, they target a fundamentally different problem setting from our *transcriptome-based drug design* (TBDD).
>
> As defined in the paper, TBDD takes as input a pair of pre- and post-perturbation transcriptomes $((T_{\text{pre}}, T_{\text{post}}))$ and aims to generate molecules that drive the cell from $(T_{\text{pre}})$ to $(T_{\text{post}})$.  This is a **function-oriented**, system-level formulation:
>
> * **Inputs in SBDD (DiffDock, Boltz-1, etc.)**: 3D protein structures and binding pockets; the model predicts binding poses/affinities of ligands given a fixed target.
> * **Inputs in TBDD (our setting)**: pre-/post-perturbation gene expression profiles; the model must map *functional responses of a complex cellular system* to candidate molecules that can induce similar responses.
>
> In particular:
>
> * DiffDock requires 3D protein pocket structures as input and is designed for docking / pose prediction; in TBDD, such structures are not available or even defined for many systemic settings.
> * Boltz-style models estimate protein–ligand interaction energies; they are not designed to condition generation on transcriptomic perturbation signatures.
>
> Because neither DiffDock nor Boltz can be directly run under our input/target specification (no protein pocket, conditioning in transcriptomic space, different objectives), they would require substantial re-design to become *applicable* baselines rather than fair “drop-in” comparators. We therefore chose not to present them as direct baselines, and we will clarify this non-applicability more explicitly in the revised manuscript.
>
> **(b) Why existing baselines are VAE-based and why we focus on them**
>
> To our knowledge, TBDD in the strict sense of *designing molecules directly from transcriptomic perturbations* has only a few prior works (GexMolGen, TRIOMPHE, Gx2Mol). All of them:
>
> * operate at **bulk** resolution, and
> * adopt **VAE-style** generative backbones.
>
> None of these prior works handles multi-resolution (bulk + single-cell) perturbation conditioning, nor do they introduce an intermediate function-oriented chemical space with multi-domain alignment as in scTrans-Gen.
>
> We therefore selected GexMolGen, TRIOMPHE, and Gx2Mol as the *most task-relevant* baselines: they operate on the same modality (expression profiles) and address similar transcriptome-guided generation problems, even though their architectures differ from ours.
>
> **(c) Additional cross-modal baselines with alternative objectives**
>
> We agree with the reviewer that it is important to show that our performance gains are not solely due to architecture complexity. In our rebuttal experiments, we therefore added **two cross-modal alignment baselines** that use alternative objectives and inductive biases:
>
> 1. **Projection Alignment (MSE-align)**
>
>    * Uses an MLP to project transcriptome features directly into a molecular latent space, trained with an MSE loss.
> 2. **Q-Former (BLIP2-style)**
>
>    * Uses a query-based transformer alignment mechanism inspired by BLIP2, where a small set of learnable queries attends to the transcriptome representations to produce a condition embedding[1].
>
> We then plug these alternative alignment modules into the same downstream generation pipeline and compare them against our CLIP-style alignment. The comparison is shown below (bulk L1000, in-distribution split):
>
> | **Method** | **Coverage↑** | **Diversity↑** | **Similarity↑** | **Distance↓** | **SA↓** | **Unique↑** | **QED↑** | **Fraggle Sim↑** | **Morgan Sim↑** | **MACCS Sim↑** |
> | --- | --- | --- | --- | --- | --- | --- | --- | --- | --- | --- |
> |**Projection Alignment**| 5 (45.45%) | 0.7234 | 0.7723 | 43.6231 | 0.6723 | 0.4451 | 0.5523 | 0.3012 | 0.2239 | 0.4215 |
> | **Q-Former (BLIP-based)**| 8 (72.73%) | 0.8156 | 0.8145 | 28.6342 | 0.6254 | 0.8112 | 0.5787 | 0.7478 | 0.4861 | 0.7623 |
> | **Ours (CLIP-based)** | 11 (100.00%) | 0.8906 | 0.9576 | 6.7856 | 0.6386 | 0.9296 | 0.5665 | 0.8892 | 0.8228 | 0.9031 |
>
> Our CLIP-based multi-domain alignment consistently achieves the best structural similarity and coverage while maintaining comparable or better diversity and drug-likeness.
>
> We will add these new baselines and results (with full metric tables) to the camera-ready version to make the empirical comparison more complete and to demonstrate that our gains are not merely due to using “more training losses,” but rather stem from a more effective functional-space alignment design.

---

> ### Author Response · Authors · 2025-11-19
>
> ## Weakness 2 — Generation and Search (Screening Strategy)
>
> **(a) Role of the screening module in the overall framework**
>
> We apologize that our original description may have made the screening component appear as a “second main method.” Our primary contribution and central focus are:
>
> > **Transcriptome-based de novo molecular design**: conditionally generating molecules from perturbation signatures.
>
> The **generate–then–search screening workflow** is intended as a *practical application layer* for situations where **de novo synthesis and validation are time-consuming**, not as an independent screening model that competes with existing virtual screening methods.
>
> In realistic clinical or translational settings:
>
> * Generating candidate molecules computationally is fast.
> * The slow, resource-intensive part is *synthesis, validation, and safety testing* of new compounds.
> * When a patient’s therapeutic window is narrow, it is more realistic to search *within existing libraries* (approved drugs, late-stage compounds, etc.).
>
> Our screening framework thus leverages scTrans-Gen in the following way:
>
> 1. Use scTrans-Gen to generate a *functionally meaningful query molecule* from a target perturbation signature.
> 2. Use this molecule as a query in a large existing library (e.g., clinical or preclinical compounds).
> 3. Retrieve top-k similar candidates that are more likely to share the desired functional effect.
>
> We will clarify in the revision that this workflow is an **application built on top of the generative model**, not a separate core methodological contribution.
>
> **(b) Clarification of the top-k behavior and similarity trends**
>
> We appreciate the request for a clearer explanation of Table 3. The experimental protocol is:
>
> * For each perturbation signature, we generate **one** molecule with scTrans-Gen.
> * We perform **nearest-neighbor search** over a large compound library.
> * For each $(k \in \{5,10,15,20\})$, we report:
>
>   * **Average structural similarity** between the generated molecule and the top-k retrieved molecules (Tanimoto on fingerprints).
>   * **Hit rate**: the probability that the ground-truth drug appears in the top-k list.
>   * **PRnet MSE** between the predicted expression effects of the retrieved molecules and the ground-truth drug.
>
> As ($k$) increases:
>
> * The **mean structural similarity** naturally **decreases**, because we include more distant neighbors.
> * The **hit rate** and **functional similarity (lower PRnet MSE)** **increase**, because we have a larger candidate set and more chances to include the true or functionally equivalent drugs.
>
> The practical message we aim to convey is:
>
> > In a realistic scenario with tens of thousands of candidate drugs, scTrans-Gen + the screener can narrow down the search to a *small top-k panel* (e.g., 10–20 compounds) that maintains high structural and functional proximity to the desired effect, which is a clinically manageable candidate set.
>
> We will clarify this protocol and interpretation in the revision to avoid confusion.

---

> ### Author Response · Authors · 2025-11-19
>
> ## Weakness 3 — Role of the Feature Extractor and Classifier-Free Guidance
>
> **(a) The feature extractor provides functional conditioning, not direct generation**
>
> We fully agree that the role of conditioning should be clearly explained. In scTrans-Gen:
>
> * The **graph diffusion model** is solely responsible for *molecular structure generation*.
> * The **feature extractor** is solely responsible for *encoding functional perturbation signals* and projecting them into the function-oriented chemical space ($Z$).
>
> Concretely, the feature extractor:
>
> 1. Takes $((T_{\text{pre}}, T_{\text{post}}))$ (bulk or single-cell, with pseudoimage processing if needed).
> 2. Uses the Transcriptome Interaction Block to model the perturbation signal.
> 3. Aligns this signal to molecular graph and fingerprint spaces via multi-domain alignment (Sec. 4.4).
> 4. Produces a **conditional embedding** ($z \in Z$) that encodes “how the drug perturbs the cell”.
>
> This embedding is then injected into the diffusion model via AdaLN-like conditioning layers (similar in spirit to text conditioning in image diffusion models such as LDM/Stable Diffusion[2]).  The feature extractor does **not** generate molecules directly; instead, it supplies a semantic control signal that guides the diffusion process.
>
> **(b) Why classifier-free guidance (CFG) and random feature dropping are appropriate**
>
> The reviewer is concerned that classifier-free guidance might reduce the contribution of the conditional embedding, especially because we randomly drop the condition with probability ($p$) during training.
>
> Our design follows the standard CFG framework [3], which has been widely adopted in conditional diffusion models for images, text, and molecules[4]. The key points are:
>
> * During training, with probability ($1-p$), the model sees the **true condition** ($C$) derived from the feature extractor; with probability ($p$), it sees a **learnable null condition** ($\emptyset$).
> * This forces a single network to jointly learn:
>
>   * The **conditional distribution** $p(M \mid C)$, and
>   * The **unconditional distribution** $p(M)$.
> * At inference, we combine these two branches with a CFG scale ($s$) (Eq. (2) in the paper), which allows us to trade off unconditional quality vs. conditional adherence.
>
> In this standard framework:
>
> * The **feature extractor’s contribution is not reduced**, because it is present in the majority of training steps and explicitly used at inference with a nonzero guidance scale.
> * Instead, the unconditional branch **stabilizes training** and provides a better base distribution, while CFG strengthens controllability.
>
> In our experiments, we varied the guidance strength ($s$) and observed a clear peak in performance (e.g., Morgan similarity and PRnet MSE) around a moderate guidance level (approximately ($s=3$); Fig. 5), consistent with prior work on CFG.  Without guidance (i.e., ($s=0$)), conditional performance is indeed poor, which further confirms that the conditional embedding is essential.
>
> We will clarify in the text that:
>
> * The random condition dropout is *part of* the standard CFG training recipe.
> * Our ablations (Fig. 5) empirically support that a properly tuned CFG scale significantly improves conditional generation quality, and there is no evidence that “a subset of molecules suffers from additional conditional information”. Rather, the model benefits from both conditional and unconditional branches.
>
> ---
>
> We are grateful for the reviewer’s careful reading and constructive suggestions. In the revised version we will:
>
> * Explicitly distinguish TBDD from SBDD and explain why DiffDock/Boltz are not directly applicable baselines.
> * Add new cross-modal baselines (Projection Alignment and Q-Former) to strengthen the empirical comparisons.
> * Clarify the goal and interpretation of the generate–then–search screening workflow.
> * Better explain the division of roles between the feature extractor and the diffusion model, and the rationale behind CFG and conditional dropout.
>
> We hope these clarifications and additional experiments address your concerns and make the contribution and empirical setup of scTrans-Gen clearer.
>
> ---
>
> ### References
> [1]Li, Junnan, et al. "Blip-2: Bootstrapping language-image pre-training with frozen image encoders and large language models." International conference on machine learning. PMLR, 2023.
>
> [2] Rombach, Robin, et al. "High-resolution image synthesis with latent diffusion models." *Proceedings of the IEEE/CVF conference on computer vision and pattern recognition*. 2022.
>
> [3] Ho, Jonathan, and Tim Salimans. "Classifier-free diffusion guidance." *arXiv preprint arXiv:2207.12598* (2022).
>
> [4] Jolicoeur-Martineau, Alexia, et al. "Any-Property-Conditional Molecule Generation with Self-Criticism using Spanning Trees." *arXiv preprint arXiv:2407.09357* (2024).

---

> ### Author Response · Authors · 2025-11-26
> **Friendly reminder regarding our response**
>
> Dear Reviewer VXwt,
>
> I hope this message finds you well.
>
> As the discussion period is nearing its end with less than 7 days remaining, I wanted to ensure we have addressed all your concerns satisfactorily.
>
> If there are any additional points or feedback you'd like us to consider, please let us know. Your insights are invaluable to us, and we're eager to address any remaining issues to improve our work.
>
> Thank you for your time and effort in reviewing our paper.
>
> Best regards, The Authors

---

### Note · Authors · 2026-01-26

I have read and agree with the venue's withdrawal policy on behalf of myself and my co-authors.

---

### Meta-Review · Area_Chair_u5iW · 2026-01-07

**Summary:**

The paper proposes scTrans-Gen, a diffusion-based conditional molecular generator driven by transcriptomic perturbation signatures, including a multi-resolution setup (bulk plus single-cell) and a multi-stage alignment pipeline. Reviewers agree the problem is novel and potentially impactful, and the empirical gains over selected baselines are large. However, the discussion did not converge to acceptance, and the overall scores remain below the acceptance bar and did not improve through discussion.

**Reviewer Concerns:**

*Partially addressed concerns during the rebuttal:*

* Need for a simpler baseline: Authors added a “Diffusion + MLP” baseline which performs substantially worse, supporting that naive conditioning is insufficient in their setup.
* Component necessity: Expanded ablations argue each major block contributes.
* Some interpretability effort: Added embedding visualizations (UMAP-style) suggesting clustering by target and stratification by cell line context.
* Additional property checks: Added ADMET-style evaluation as post-hoc profiling.
* Clarity fixes: Acknowledged and promised to fix presentation issues and wording around ill-posedness.

---

*Remaining concerns:*

* Task/evaluation ambiguity (function design vs identification): High structural similarity remains a central issue. The protocol still primarily evaluates similarity to a single held-out drug and proxy functional predictors, which does not fully resolve the “many molecules can yield similar transcriptomic effects” critique.
* Biological mechanism validation remains thin: Beyond proxy functional metrics and latent clustering plots, the paper does not provide pathway-level or MoA-level analyses that convincingly demonstrate mechanistic organization of the learned space (as requested explicitly by Ad35), and no experimental validation is provided.
* Proxy functional evaluation and potential circularity: PRNet-based MSE is helpful but still a learned proxy; even with split hygiene, it does not substitute for orthogonal functional evidence.
* Single-cell motivation vs representation choice: Concerns remain that the pseudoimage and sampling choices may trade off heterogeneity for denoising in a way that blurs what is uniquely gained from single-cell conditioning.
* Complexity and reproducibility: Multi-stage training, multiple pretrained components, and many hyperparameters reduce confidence in reproducibility and in how robust the gains are across setups.

**Reviewer Scores:**

VXwt (originally 2): Unlikely to change without engagement; core baseline request remained misaligned with the paper’s problem definition and the reviewer did not participate in discussion. I also agree with the authors' comment that the behavior was somewhat unfair, but it could be that with a proper discussion period things would have looked differently.

Ad35 (originally  4): Explicitly stated they would not raise score after additional experiments; no change expected.

hdD8 (originally  4): Acknowledged clarifications but maintained major reservations; no change expected.

ZVqr (originally  6): Authors addressed several presentation and ablation requests; this could plausibly move slightly upward, but given lack of reviewer follow-up, I assume no change in score.

---

### Decision · Program_Chairs · 2026-01-26

Reject